# Comprehensive proteogenomic characterization of early duodenal cancer reveals the carcinogenesis tracks of different subtypes

Lingling Li[1,6], Dongxian Jiang [2,6], Hui Liu[3,6], Chunmei Guo[1,6], Rui Zhao[4,6], Qiao Zhang[1,6], Chen Xu [2,6], Zhaoyu Qin[1], Jinwen Feng[1], Yang Liu[1], Haixing Wang[2], Weijie Chen[2], Xue Zhang[2], Bin Li[2], Lin Bai[1], Sha Tian[1], Subei Tan[1], Zixiang Yu[2], Lingli Chen[2], Jie Huang[2], Jian-Yuan Zhao [4,5 ✉], Yingyong Hou [2 ✉] & Chen Ding[1 ✉]

The subtypes of duodenal cancer (DC) are complicated and the carcinogenesis process is not well characterized. We present comprehensive characterization of 438 samples from 156 DC patients, covering 2 major and 5 rare subtypes. Proteogenomics reveals *LYN* amplification at the chromosome 8q gain functioned in the transmit from intraepithelial neoplasia phase to infiltration tumor phase via MAPK signaling, and illustrates the *DST* mutation improves mTOR signaling in the duodenal adenocarcinoma stage. Proteome-based analysis elucidates stage-specific molecular characterizations and carcinogenesis tracks, and defines the cancer-driving waves of the adenocarcinoma and Brunner's gland subtypes. The drug-targetable alanyl-tRNA synthetase (AARS1) in the high tumor mutation burden/immune infiltration is significantly enhanced in DC progression, and catalyzes the lysine-alanylation of poly-ADP-ribose polymerases (PARP1), which decreases the apoptosis of cancer cells, eventually promoting cell proliferation and tumorigenesis. We assess the proteogenomic landscape of early DC, and provide insights into the molecular features corresponding therapeutic targets.

Small intestine cancer (SIC) is a malignantly rare cancer, of which the incidence rate was approximately 0.6%, and the mortality was 0.3% of cancer-related death worldwide (https://seer.cancer.gov/statfacts/html/smint.html)[1]. DC, the most common type of SIC (ranging from 30 to 50%)[2,3], mainly consists of four types, adenocarcinoma, neuroendocrine tumor, stromal tumor, and lymphoma[4]. Recent advances in imaging have resulted in dramatic improvements in diagnostic

capability and concluded typical subtypes of duodenal lesions, including two major subtypes (adenocarcinoma (named D subtype in this study) and Brunner's gland (named G subtype in this study)) and other subtypes, including heterotopic pancreas (named P subtype in this study), intermuscular gland (named N subtype in this study), lymphangioma (named L subtype in this study), cystic dystrophy (named C subtype in this study), well-differentiated neuroendocrine

[1]State Key Laboratory of Genetic Engineering and Collaborative Innovation Center for Genetics and Development, School of Life Sciences, Institute of Biomedical Sciences, Human Phenome Institute, Zhongshan Hospital, Fudan University, Shanghai 200433, China. [2]Department of Pathology, Zhongshan Hospital, Fudan University, Shanghai 200032, China. [3]State Key Laboratory Cell Differentiation and Regulation, Overseas Expertise Introduction Center for Discipline Innovation of Pulmonary Fibrosis, (111 Project), College of Life Science, Henan Normal University, Xinxiang 453007, China. [4]Institute for Development and Regenerative Cardiovascular Medicine, MOE-Shanghai Key Laboratory of Children's Environmental Health, Xinhua HospitalShanghai Jiao Tong University School of Medicine, Shanghai 200092, China. [5]Department of Anatomy and Neuroscience Research Institute, School of Basic Medical Sciences, Zhengzhou University, Zhengzhou 450001, China. [6]These authors contributed equally: Lingling Li, Dongxian Jiang, Hui Liu, Rui Zhao, Qiao Zhang, Chunmei Guo, Chen Xu. ✉e-mail: zhaojy@fudan.edu.cn; hou.yingyong@zs-hospital.sh.cn; chend@fudan.edu.cn

tumor (named NET subtype in this study)[5]. Whereas, the specific characterizations and associations of the complex subtypes have not been revealed. Interestingly, the D subtype usually co-occurred with the G subtype in one patient, while occasionally with rare NET/L/P/N/C subtypes. However, the origins and the driven events of different DC subtypes remain largely unknown. Tobacco usage has important implications regarding the gut environment and various small intestine disorders, including functional dyspepsia[6], irritable bowel syndrome[7], and duodenal ulcer disease[8]. Nevertheless, the contribution of tobacco usage in DC needs to be further explored.

The complexity, tumor heterogeneity, and low incidence of DC are challenging to collect the DC sample. Several genomic datasets have illustrated the genomic aberrations of small-bowel adenocarcinoma, and NET presented the frequent mutations, and proposed the effects of mTOR signaling in tumor carcinogenesis[9]. However, the impacts of the genomic aberrations on the proteomic alterations and phosphoproteomic actions remain unclear yet. In addition, the findings are focused on the advanced stages, the first occurrences of the mutations/key events, and the related effects during the carcinogenesis of DC are poor understood.

In DC progression, the morphological characteristics are complicated, especially for the two major subtypes (D and G subtypes). Due to its sporadic nature and locations on the posterior or lateral wall of the second part of the duodenum, the D subtype follows the transition pattern of colorectal cancer (CRC), from adenoma (T1 stage) to adenocarcinoma, including minimally invasion adenocarcinoma (named MIA stage in this study) and advanced-stage adenocarcinoma (e.g., T2–T4 stages, named duodenal adenocarcinoma (DAC) stage in this study)[10]. According to the differentiation degree, the epithelial adenoma can be divided into low-grade intraepithelial neoplasia stage (named LGIN in this study), and high-grade intraepithelial neoplasia stage (named HGIN stage in this study)[11]. The tubular adenoma and tubulovillous adenoma are usually present in the LGIN and HGIN[12,13].

The G subtype showed the features of high-grade dysplasia and transformed from normal (named N-G in this study) to hyperplasia (named H–G in this study) and to adenoma (named A–G in this study)[14–16]. The risks of patients at the different (sub)stages in DC progression are quite diverse. For the patients at the LGIN stage, the 5-year cumulative probability of eventual progression to the HGIN or to adenocarcinoma has been reported as no more than 50%[17,18]. Whereas, the HGIN is usually progressed to adenocarcinoma in pathology (>90%), and should be considered for surveillance[19]. The different stages during the carcinogenesis of DC are correlated with the lesion size and invasion to the corresponding layers[20] and require distinctive operation strategies. For example, for the lesions at the LGIN and HGIN stages (<20 mm in size), endoscopic resection (e.g., ESD and EMR) is suggested as an effective treatment approach[21]. The lesions invaded into the submucosal layer (≥20 mm in size) are assessed as carcinoma and are required for surgical resection[22]. For the lesions at the MIA stage, the clinic management is controversial, further implying the complexity and undefined characteristics of the (sub)stages in DC progression. Thus, it is vital for the revelation of the key events during the carcinogenesis process of DC to explore the pathological mechanism and the corresponding targeted therapy in DC progression. However, the molecular characterizations of the (sub) stages of the lesions in DC progression are yet unclear.

Generally, DC is clinically approached to CRC in a similar manner, such as panitumumab to anti-EGFR[23,24]. However, compared with CRC, DC patients have a worse outcome of 5-year survival. The increasing incidence and ambiguous features, and unclear malignancy potentials of the different subtypes require specific management of the DC clinic strategy. In addition, there is still a lack of targeted drugs for DC. Thus, precise strategies and specific druggable candidates for DC are in urgent need.

Here, we show comprehensive proteogenomic characterization of 438 samples dissected from early DC (EDC) of different stages in 156 patients. The sample pool covers nearly all stages of DC, including 2 major (D/G subtypes) and 5 rare (NET/L/P/N/C subtypes) pathological subtypes with 23 specific substages from early to advanced stages of DC. The comprehensive multi-omics analysis reveals the stage-specific molecular characterization and elucidates the key events in the transmit of the phases, and identifies four immune clusters. Furthermore, we propose the potential functions of AARS1 in DC progression, facilitating future basic and translational research of this rare digestive tract cancer.

## Results

### The proteogenomic landscape of early duodenal cancer cohort
To characterize the proteogenomic landscape of DC, whole-exome sequencing (WES), proteomic, and phosphoproteomic data, 438 samples were collected from 156 DC patients based on stringent criteria (Methods), who had not experienced prior chemotherapy or radiotherapy, covering 2 major (D/G subtypes) and 5 rare subtypes (NET/L/P/N/C subtypes) (Supplementary Fig. 1a). The clinicopathological characteristics of patients with different subtypes were summarized in Supplementary Data 1a (Table 1). To ensure the quality of the samples in our cohort, 438 samples spanning from 2012 to 2019, were separately dissected from the formalin-fixed, paraffin-embedded (FFPE) slides and were well preserved and systematically evaluated to confirm the histopathologic diagnosis and variant histology according to the World Health Organization (WHO) classification[11] by more than two expert gastrointestinal pathologists (Methods). Laser-capture microdissection (LCM)[25] was applied to dissect the sections of substage/subtype samples precisely (Supplementary Fig. 1b; Methods), and the tumor purity of all the tumor samples was over 95%, indicating the high quality of all samples in our cohort (Supplementary Fig. 1c, d; Supplementary Data 1b). As for the normal duodenal tissue, the proportion of normal cells is 100%, while the tumor cell is 0% in our cohort.

**Table 1 | Clinical and histopathologic characteristics of the Fudan DC cohort (n = 156)**

| Characteristic | n (%) | Parameter | n (%) |
|---|---|---|---|
| Age | | Survival status | |
| <50 yr | 38 (24.4) | Alive | 156 (100) |
| 50–70 yr | 101 (64.7) | Death | 0 (0) |
| >70 yr | 17 (10.9) | Histopathologic (438 samples) | 325 (74.2) |
| Gender | | Epithelial components | |
| Male | 55 (35.3) | Yes | 325 (74.2) |
| Female | 101 (64.7) | No | 113 (25.8) |
| Smoking | | Histology subtype | |
| Yes | 37 (23.7) | Adenocarcinoma (D) | 325 (74.2) |
| No | 92 (59.0) | Brunner's gland (G) | 97 (22.1) |
| NA | 27 (17.3) | Heterotopic pancreas (P) | 1 (0.2) |
| Drinking | | Intermuscular gland (N) | 9 (2.1) |
| Yes | 30 (19.2) | Lymphangioma (L) | 2 (0.5) |
| No | 93 (59.6) | Cystic dystrophy (C) | 3 (0.7) |
| NA | 33 (21.2) | Neuroendocrine tumor (NET) | 1 (0.2) |
| FAP | | | |
| Yes | 2 (1.3) | | |
| No | 116 (74.4) | | |
| NA | 38 (24.4) | | |
| HNPCC | | | |
| Yes | 4 (2.6) | | |
| No | 83 (53.2) | | |
| NA | 69 (44.2) | | |

**Table 2 | Subclassification information and the number of substage samples in the D subtype**

| Stages | Substages (n = sample number) |
|---|---|
| Non-tumor stage (NT stage) | 1 (n = 150), 1_1 (n = 1), 1_2 (n = 3), 1_3 (n = 7), 1_4 (n = 4), 1_5 (n = 1), 1_6 (n = 1) |
| Low-grade intraepithelial neoplasia stage (LGIN stage) | 2_1 (n = 22), 2_2 (n = 11), 2_3 (n = 21), 2_4 (n = 8) |
| High-grade intraepithelial neoplasia stage (HGIN stage) | 3_1 (n = 13), 3_2 (n = 9), 3_3 (n = 24), 3_4 (n = 25) |
| Minimally invasion adenocarcinoma stage (MIA stage) | 4 (n = 5) |
| Duodenal adenocarcinoma stage (DAC stage) | 5 (n = 20) |

**Table 3 | Subclassification information and the number of substage samples in the G subtype**

| Stages | Substages (n = sample number) |
|---|---|
| Normal stage (N–G stage) | G_1 (n = 47) |
| Hyperplasia Brunner's stage (H–G stage) | G_2 (n = 19), G_2_1 (n = 7) |
| Adenoma Brunner's stage (A–G stage) | G_3 (n = 22), G_3_1 (n = 1), G_3_2 (n = 1) |

Briefly, five stages (non-tumor (NT), LGIN, HGIN, MIA, and DAC stages from the D subtype, three stages (N–G, H–G, and A–G Brunner's gland stages) from the G subtype, were collected for multi-omics analysis (Tables 2 and 3). In total, 438 samples were collected for proteomic profiling, in which 120 samples (47 DC cases) and 111 samples (49 DC cases) were conducted on whole-exome sequencing (WES) (15.5 G raw data, 135× depth-coverage) and phosphoproteomic profiling, respectively (Fig. 1a and Supplementary Fig. 2a; Supplementary Data 1c and 2a). The stage-based classification with high tumor purity in our cohort, represented the advantages of our samples in a pathological region-resolved mode, providing the chance to portray molecular profiles of DC in a time-resolved mode.

At the genomic level, a total of 11,504 mutations were identified in DC, of which 10,454 and 2810 mutations were detected in the D and G subtypes, respectively (Supplementary Data 2b). The top-ranked mutations in the carcinogenesis of the D subtype were *DST*, *APC*, *MUC16*, *BIRC6*, *SYNE1*, etc. (Fig. 1b). In the G subtype, *KMT2B*, *MAP7D3*, and *NRXN2* were highly mutated (Fig. 1c). According to the pathological features and lesion invasion, the D subtype was divided into the NT phase (including NT stage), the intraepithelial neoplasia (IEN) phase (including LGIN and HGIN stages), and the infiltration tumor (IFT) phase (including MIA and DAC stages), and the G subtype was separated into normal Brunner's gland (N–G phase) and abnormal Brunner's gland (AN–G phase) (Supplementary Fig. 2b). Low mutation loads were observed in the IEN phase of D subtype and G subtype, as well as the N and C subtypes (Fig. 1d), indicating lowly mutation loads in the earlier phase in DC progression. During the carcinogenesis of DC, neo-mutations were found in different stages. Specifically, the number of the neo-mutations was increasingly increased in the D subtype, while the number peaked at the H–G stage in the G subtype (Supplementary Fig. 2b).

At the protein level, protein abundance was calculated by intensity-based absolute quantification (iBAQ)[26,27] and then normalized as a fraction of the total (FOT), allowing for comparisons between experiments. Whole-cell extract of HEK293T cells was used as the quality control (QC), which showed the MS was robust and consistent based on the large Spearman's correlation coefficients (mean = 0.93) between the QC samples (Supplementary Fig. 2c; Supplementary Data

3a). Label-free quantification measurement of 438 samples resulted in a total of 14,426 protein groups with a 1% false discovery rate (FDR) at the protein and peptide levels (Fig. 1e). The ascending numbers of protein identifications were also shown in the G subtype (Supplementary Fig. 2d; Supplementary Data 3b). Integration of the FFPE procedures, high tumor quality, and systematical evaluation, ensured the stability of the DC tissues/samples spanning from 2012 to 2019, evidenced by the no impacts of the different storage times on the protein identification of the samples at the same pathological stage (Supplementary Fig. 2e). During the carcinogenesis of the D and G subtypes, gradually decreased (Spearman's) correlation coefficient of the histopathological stages reflected the higher tumor heterogeneity during the carcinogenesis of DC (Supplementary Fig. 2f), highlighting the importance of exploring molecular characteristics in DC progression. At the phosphoprotein level, 38,696 phosphosites, corresponding to 9,716 proteins, were quantified in the DC progression (Fig. 1f and Supplementary Fig. 2g; Supplementary Data 3c). Taken together, we established a comprehensive landscape of DC progression at the gene, protein, and phosphoprotein levels, portraying two major (D/G subtypes) and five rare subtypes (NET/L/P/N/C subtypes) of DC.

### Enrichment of tobacco signature and DNA repair panel in DC

To investigate the specific etiological factors that may contribute to the mutagenesis of DC, we used non-negative matrix factorization (NMF)[28], and then performed cosine similarity analysis against mutational signatures in human cancer according to the Catalog of Somatic Mutations in Cancer (COSMIC) (https://cancer.sanger.ac.uk/cosmic/signatures)[29,30] (Supplementary Fig. 3a). As well as in the small intestinal NET[9], cytosine to thymine (C > T) substitutions were the predominant SNV type in the D and G subtypes (Supplementary Fig. 3b). The SBS29 signature, also known as Tobacco signature, exhibited transcriptional strand bias for C > A substitutions, and is associated with tobacco chewing[31]. In our cohort, we found that the SBS29 signature (Tobacco signature) was overrepresented in the D subtype (20 of 68 samples), and was detected in the NT stage last till to DAC stage of the D subtype (Fig. 2a and Supplementary Fig. 3c; Supplementary Data 4a). In addition, integration of the clinical information of 156 DC patients revealed that more patients (60% vs. 18.8%, Fisher's exact test, $p = 1.4E-3$) with smoking habit were observed in the Tobacco signature group with higher SBS29 (Tobacco signature) score (Wilcoxon signed-rank test, $p = 0.024$) (Fig. 2b and Supplementary Fig. 3d). These findings reflected that tobacco might be a risk factor for DC, especially for the D subtype.

Generally, cytochromes P450 families (CYPs) play a key role in the metabolic process of nicotine, major content in tobacco, and it is nitrosated carcinogenic derivative 4-(methylnitrosamino)−1-(3-pyridyl)−1-butanone (NNK)[32,33]. In our cohort, overrepresentation of CYPs (e.g., CYP2W1, CYP7B1, etc.) was observed in the Tobacco signature (Supplementary Fig. 3e), indicating the formation of NNK. Glycosaminoglycans (GAGs) enhanced the metabolic process from NNK to NNK-O-glucuronic acid, which can be released in small intestinal and kidney[34–36]. To further investigate the impacts of the Tobacco signature, we performed GO/KEGG enrichment analysis, and found protein O-linked glycosylation and aminoglycan catabolism were overrepresented in the Tobacco signature (Fig. 2c). In addition, positive association (Pearson's $R = 0.41$, $p = 2.2E-3$) between CYPs and aminoglycan catabolism was notably observed (Fig. 2d), implying the activation and cumulation of GAGs. Taken together, the Tobacco signature was prominent in the D subtype, which might induce NNK-O-glucuronic acid secretion via the overrepresentation of GAG (Fig. 2h).

Other signatures, such as signatures 6, 15, and 88, correlated with DNA repair[31], were detected in the D, G, P, and N subtypes, and named as 'DNA repair panel' in DC (Supplementary Fig. 3c). Microsatellite instability (MSI) occurs alterations in short-repetitive DNA sequences[37], as a consequence of DNA mismatch repair deficiency[38],

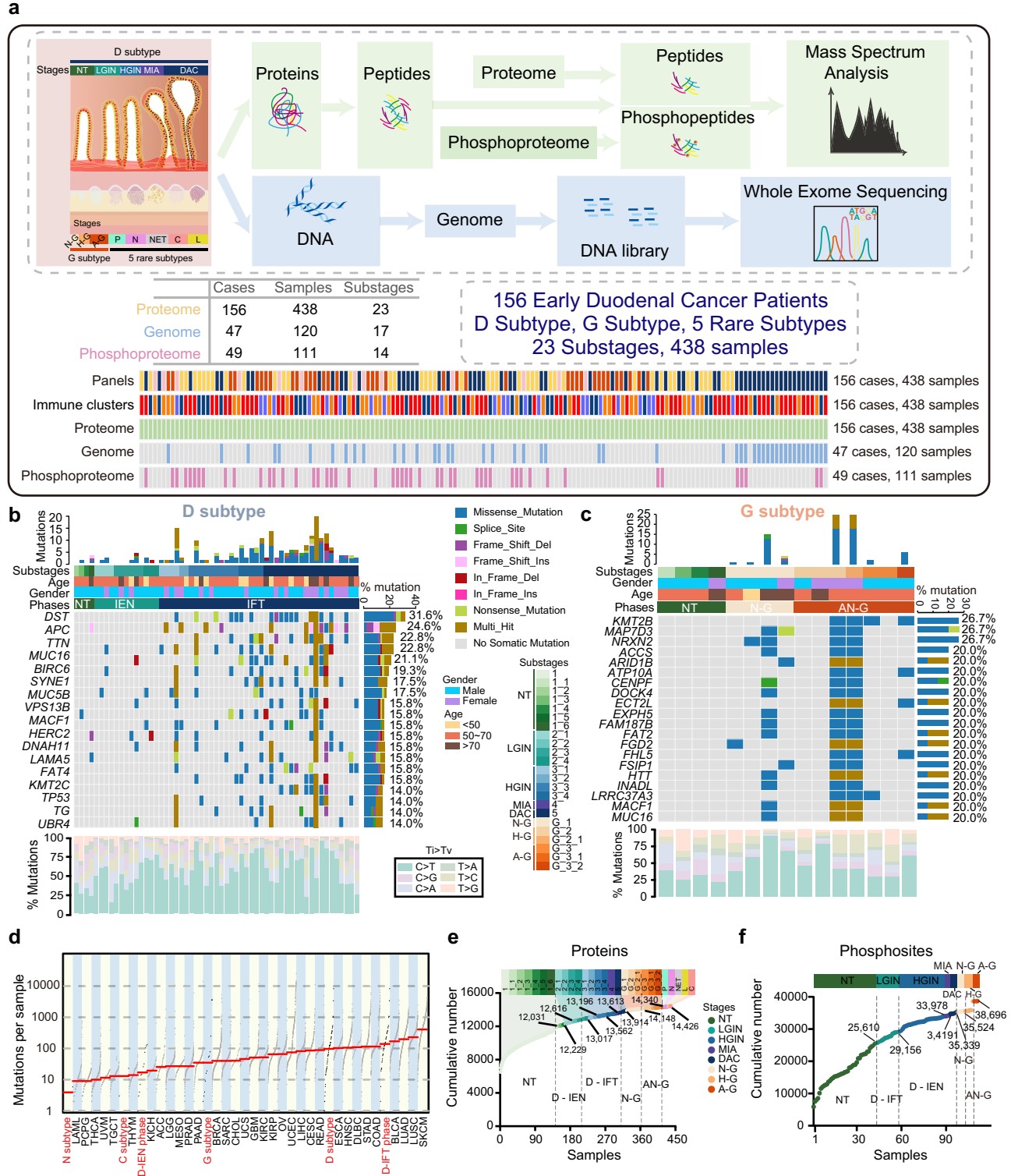

**Fig. 1 | Multi-omics landscape in DC progression. a** Overview of the experimental design and the number of samples for the genomic, proteomic, and phospho-proteomic analyses. The genomic profiles in the D subtype (**b**) and the G subtype (**c**) progression. Top: the mutation numbers and types of all the samples during carcinogenesis of DC. DC: duodenal cancer. Bottom: the percentages of somatic SNVs of all samples. The mutation frequencies are shown by a bar plot on the right panel. The square directs to a subset of patient samples used for WES (*n* = 120).

WES whole-exome sequencing, SNVs single nucleotide variants. **d** The mutation loads in the diverse phases (red) of D subtypes and other subtypes in DC. The cumulative number of protein identifications (**e**) and phosphosite identifications (**f**) in DC progression. A total of 438 and 111 samples for proteomic profiling and phosphoproteomics profiling, respectively, are used in the analysis. ****$p$ < 1.0E−4, ***$p$ < 1.0E−3, **$p$ < 1.0E−2, * $p$ < 0.05, ns. >0.05. Source data are provided as a Source data file.

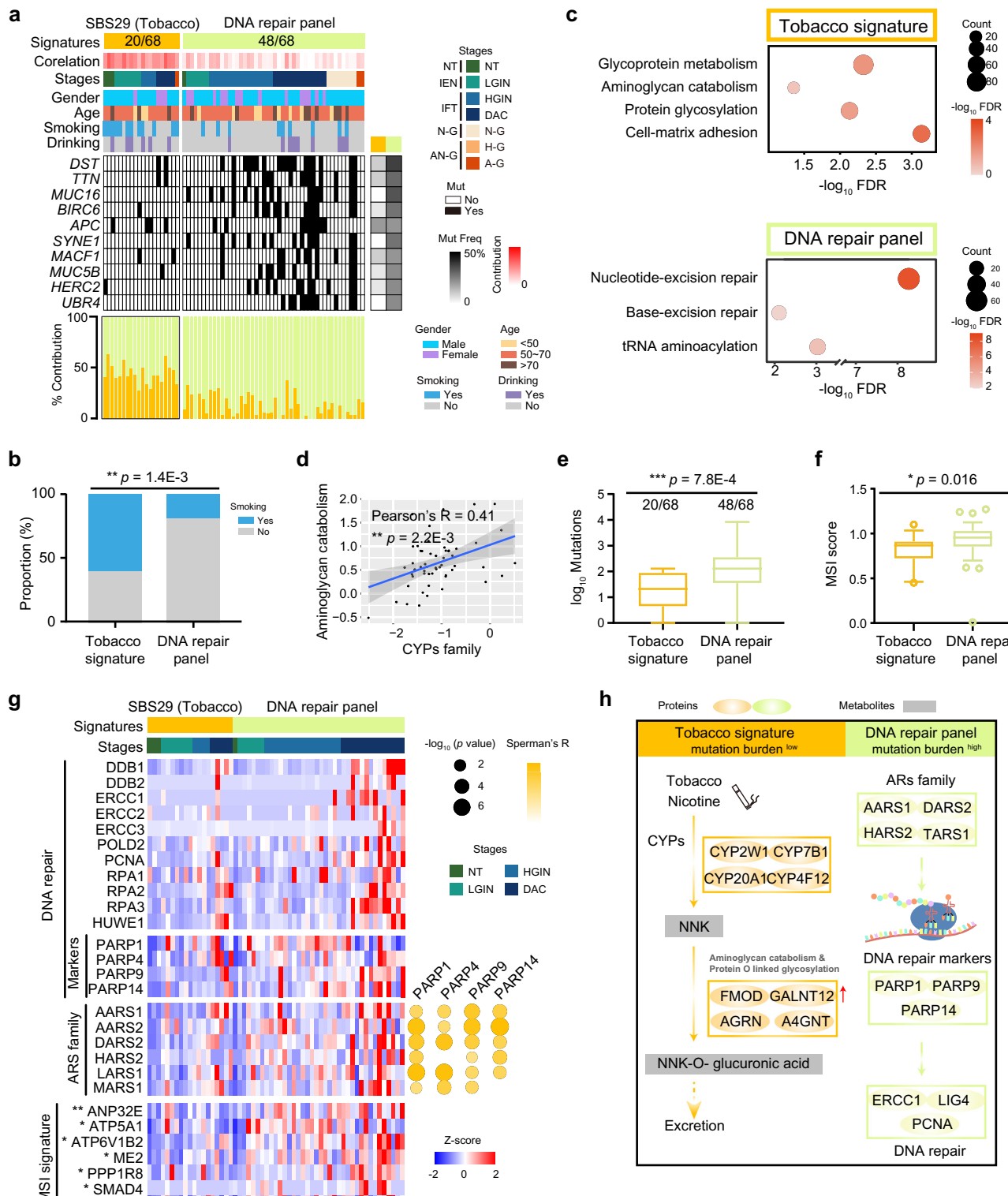

**Fig. 2 | The impacts of the Tobacco signature and DNA repair panel in DC.**
**a** Heatmap showing the clinic information (top), correlated mutations (middle), and the SBS29 (Tobacco) signature contribution (bottom) in the D and G subtypes. The square is directed to a subset of patient samples of the D and G subtypes used for WES (n = 120). **b** Histogram showing the proportion of the samples with the habit of smoking in the Tobacco signature and DNA repair panel (two-sided Fisher's exact test). **c** The dominant pathways in the Tobacco signature (top) and DNA repair panel (bottom). **d** Scatterplot showing the relationship between CYPs and aminoglycan catabolism at the protein level (two-sided Pearson's correlation test). CYPs cytochromes P450 families. **e** Boxplot showing the TMB in the Tobacco signature and DNA repair panel (two-sided Wilcoxon signed-ranked test). Boxplot shows median (central line), upper and lower quartiles (box limits), and 1.5× interquartile range (whiskers). **f** Boxplot presenting the MSI score in the Tobacco signature and DNA repair panel (two-sided Wilcoxon signed-ranked test). MSI microsatellite instability. Boxplot shows median (central line), upper and lower quartiles (box limits), and 1.5× interquartile range (whiskers). **g** Heatmap depicting the represented proteins of the predominant pathways in the DNA repair panel. The square directs to a subset of patient samples in the D subtype used for WES. **h** A brief summary of the impacts of the Tobacco signature and DNA repair panel in DC. ****$p < 1.0E-4$, ***$p < 1.0E-3$, **$p < 1.0E-2$, * $p < 0.05$, ns. >0.05. Source data are provided as a Source data file.

and shows a positive association with the presence of hypermutation[39,40]. In our cohort, a significantly higher tumor mutation burden (TMB) was detected (Wilcoxon signed-rank test, $p < 1.0E{-}3$) in the DNA repair panel (Fig. 2e and Supplementary Fig. 3f), suggesting MSI might exist in the DNA repair panel of DC. In order to validate the assumptions, we performed MSI analysis on the basis of MSI score, which was then found to be higher in the DNA repair panel (Wilcoxon signed-rank test, $p = 0.016$), evidenced by the over-represented MSI signature-related proteins in the DNA repair panel (Fig. 2f, g, and Supplementary Fig. 3g).

Furthermore, DNA repair-related proteins, including PARPs (e.g., PARP1 and PARP9) (Wilcoxon signed-rank test, adjust. $p < 0.05$), were prominent in the DNA repair panel (Fig. 2g). Aminoacyl-tRNA synthetases (ARS) are essential enzymes for protein synthesis with evolutionarily conserved enzymatic mechanisms[41], and promote the genes expression of PARPs, which act in the DNA damage-induced ADP-ribosylation[42]. In our cohort, the aminoacyl-tRNA biosynthesis was predominant in the DNA repair panel, evidenced by the over-representation of ARS families (Supplementary Fig. 3h), indicating the translational functions in the DNA repair panel. We then integrated the ARS families and performed a correlation analysis between the ARS families and PARPs. As a result, we found AARS1 was positively correlated with the significant makers (PARP1, PARP9, and PARP14) in DNA repair signaling (Fig. 2g). Moreover, AARS1 showed a significantly positive correlation with MSI-related proteins[40] (Supplementary Fig. 3i). Collectively, DNA repair panel was featured with high TMB and MSI, in which ARS-PARPs signaling was overactivated (Fig. 2h).

## Involvement of the chr8q gain in the transmit process from the IEN to IFT phase

Somatic copy number alterations (SCNAs) affected proteins and phosphoproteins in either "cis" or "trans" modes, corresponding to the diagonal and vertical patterns. To explore the impacts of SCNAs, we performed whole exome-based SCNAs analyses and examined the regulatory effects of 23,109 SCNAs on protein expressions of 120 samples in DC progression (Supplementary Data 2c). To this end, we found the gain of chromosome 8q (chr8q) was notably observed in the IFT phase but not in the IEN phase (Supplementary Fig. 4a). To explore the biological functions of the gain of chr8q, we performed cis effects of the genes with CNA regions at the protein level, and found 29 genes at chr8q gain perturbation profiles had significantly positive cis effects on their cognate proteins (Fig. 3a, b, and Supplementary Fig. 4b). Interestingly, these cis-effect genes at chr8q gain were involved in MAPK signaling, which were also detected in other gastrointestinal cancers[43], such as esophageal adenocarcinoma (ESCA), gastric cancer (GC), and CRC (Fig. 3c and Supplementary Fig. 4c). To assess the effects of these amplified genes, we annotated outliers for the degree of which short hairpin RNA (RNAi)-mediated depletion reduced DC cell lines (AZ521 and HUTU80 cell lines)[44]. As a result, we found the amplifications of PTK2, PUF60, RAB2B, and LYN, had negative effects on the proliferation of DC cells, in which LYN was recorded in cancer-associated genes (CAGs)[45,46] (Fig. 3b; Supplementary Data 4b).

LYN, a member of Src kinases, regulates the activation of ERK1/2 signaling[47] and promotes cell proliferation and survival in many diseases, including CRCs[48] and chronic myelogenous leukemia[49]. In our cohort, the LYN amplification had positive impacts on the proteome-levels of MAPK signaling and cell proliferation (Fig. 3d), evidenced by the adapters (e.g., CRK and CRKL), and LYN-protein complex (e.g., CBL and PTPN6) (Fig. 3e), which activated the MAPK signaling[48]. In addition, LYN was positively associated with the adapters and LYN-protein complex at the protein level. Furthermore, the phosphoprotein-levels of MAPK signaling (e.g., MAP2K2 S26 and MAPK1 Y187) were overrepresented in the IFT phase (Fig. 3e). These findings indicated the impacts of LYN amplification on MAPK signaling based on phosphorylation in DC progression.

To further investigate how LYN amplification affected the activation of MAPK signaling in DC, we performed kinase-substrate enrichment analysis (KSEA) of phosphoproteome on the LYN amplification group. As a result, MAPK signaling (e.g., MAPK1/3/7 and RPS6KA3) related kinases were overrepresented in the LYN amplification group, in which MAPK1 showed a positive association with LYN (Pearson's $R = 0.28$, $p = 3.2E{-}7$) (Fig. 3f and Supplementary Fig. 4d). Based on kinase-substrate regulation network analysis, among of those associated substrates, we found that MAPK Y187 was the only substrate of LYN identified and supported by our data, and was overrepresented in the LYN amplification group (Wilcoxon signed-ranked test, adjust. $p = 3.3E{-}3$, LYN Amp. vs. WT ratio = 2.52) (Fig. 3g and Supplementary Fig. 4e). In addition, the enhanced phosphorylation of MAPK1 (Y187) could activate the substrates of RPS6KA3 (S369, S375, S715, and T365) and TOP2A (S1213) (Supplementary Fig. 4f), which is one of the markers in cell proliferation[50]. These results implied the LYN amplification had positive impacts on cell proliferation in DC progression. In addition, LYN showed a positive association with cell proliferation-related markers (e.g., MCM4 and MCM6), and was significantly correlated with the cell proliferation pathway (Pearson's correlation, $R = 0.35$, $p = 2.7E{-}10$) (Fig. 3h and Supplementary Fig. 4g). Together, the chr8q gain functioned in the transmit process from the IEN phase to the IFT phase, in which the amplification of LYN enhanced the protein-level of LYN, which activated MAPK signaling and promoted the DC cell proliferation, hinting the potential medical action of saracatinib[51] in the DC clinic (Fig. 3i).

## The diverse characteristic origins and carcinogenesis tracks of the D and G subtypes

To explore the association between pathological subtypes and proteomic subtyping, we performed consensus clustering analysis on 438 samples of DC, and identified two clusters (Supplementary Fig. 5a). Specifically, cluster 1 (CCP1) mainly consisted of the non-surface epithelial components, including G, NET, L, P, N, and C subtypes (Supplementary Fig. 5b). The most of the samples in the surface epithelial component (D subtype) were observed in the cluster 2 (CCP2). These results indicated that the proteomic methods could be applied to the distinction of the pathological layers.

To investigate the molecular characterizations of the two clusters (CCP1 and CCP2), we integrated the differential expressed proteins (DEPs) between CCP1 and CCP2 (Wilcoxon signed-rank test, adjust. $p < 0.05$, CCP2 vs. CCP1 ratio $\geq 2$ or $\leq 0.5$), and found 1110 and 1992 proteins were overrepresented in the CCP1 and CCP2, respectively. Specifically, the CCP1 proteins participated in ECM signaling (e.g., COL1A1 and THBS3) and complement cascade (e.g., CFD and C5). In CCP2, the dominant pathways were cell cycle (e.g., CDK4 and RB1), apoptosis (e.g., BAX and CASP3), and glycolysis (e.g., ENO1 and HK2) (Supplementary Fig. 5c).

The two major subtypes (D and G subtypes) account for nearly 90% of DC[5], and were prominent in the CCP2 and CCP1, respectively. However, the divergency of their origins remains yet unknown. In our cohort, 60 patients (38.5%) had the D subtype only, 32 patients (20.5%) had the G subtype only, 54 patients (34.6%) had both D and G subtypes, and 10 patients (6.4%) had other subtypes. To explore the difference between D and G subtypes, we divided 156 DC cases into 4 panels shown in the Sankey diagram: Panel 1 (P1, patients with D subtype only), Panel 2 (P2, patients with G subtype only), Panel 3 (P3, patients with both the D and G subtypes), and Panel 4 (P4, patients with other subtypes) (Fig. 4a). Notably, the P1 and P2 were prominent in the CCP2 and CCP1, respectively (Fisher's exact test, $p < 2.2 E{-}16$) (Supplementary Fig. 5e). In addition, in the P3, the proportions of CCP1 and CCP2 were comparable, and while the P4 was overrepresented in the CCP1. The findings indicated that the molecular characterization of subtype-based panels might be relevant to that of the proteomic clusters. In addition, the proteomic cluster presented that the featured protein

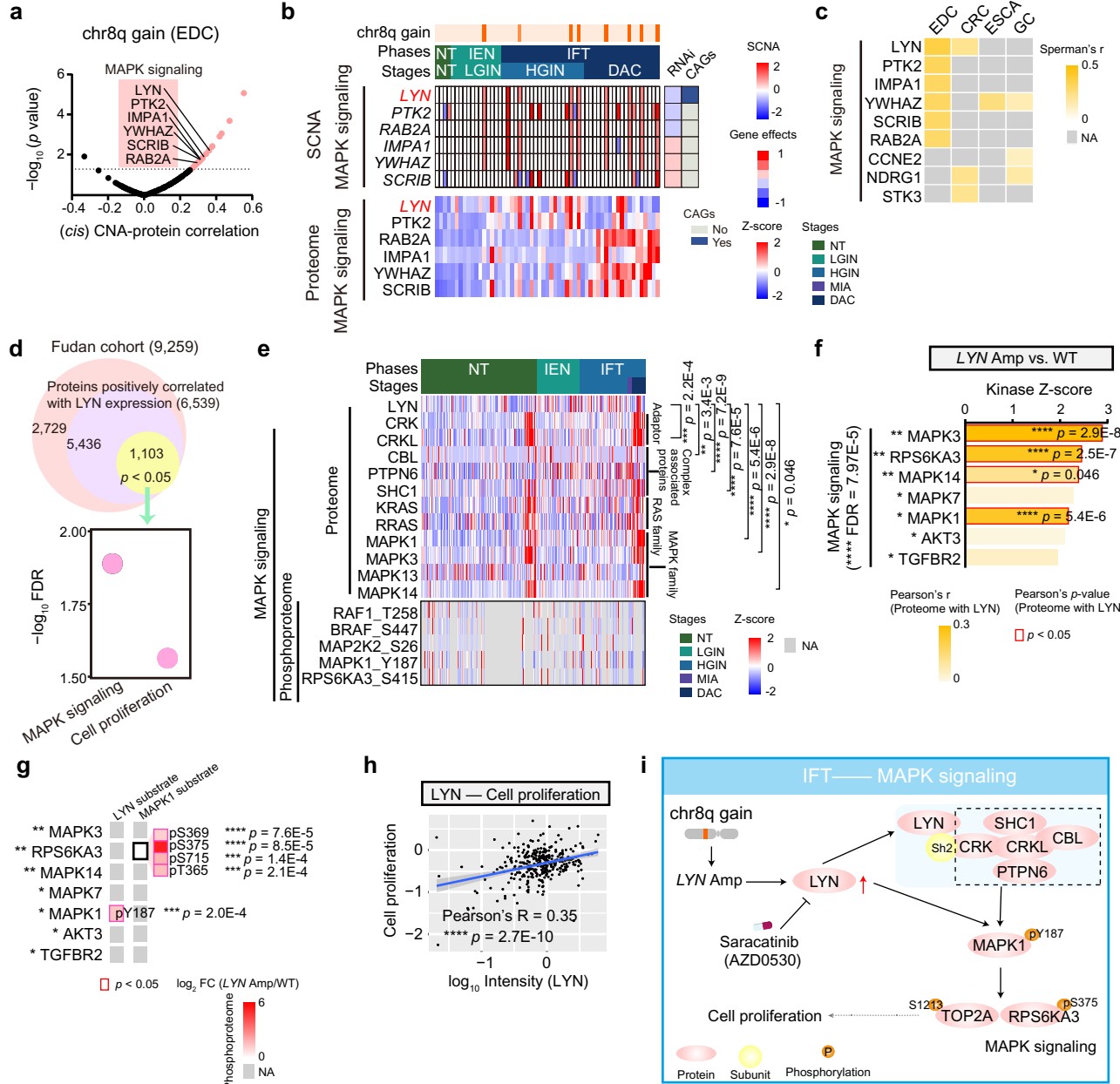

**Fig. 3 | The impacts of chr8q gain in the D subtype. a** Volcano plot showing the *cis*- correlation of the SCNA (*x*-axis) and the associated −log₁₀ (*p*-value) (*y*-axis) on the genes at chr8q gain (two-sided Spearman's correlation test). SCNA somatic copy number alterations. **b** The *cis* SCNA-protein regulations of significantly correlated genes (top) on their corresponding proteins expression (bottom). The genes are related to MAPK signaling. The right represents relative survival averaged across all available DC cell lines after depletion of the indicated genes by RNAi (Dependency map-supported (https://depmap.org) panels) and shows the correlation between the genes and CAGs. CAGs cancer-associated genes. The square directs to a subset of patient samples in the D subtype used for WES (*n* = 120). **c** The impacts of the MAPK signaling associated SCNAs on their corresponding protein expression in gastrointestinal cancers (two-sided Spearman's correlation test). **d** Venn diagram depicting the amplification of LYN positive associated proteins (top), and the represented signaling pathways (bottom). **e** Heatmap showing the

represented proteins in MAPK signaling at the protein (top) and phosphoprotein (bottom) levels (two-sided Pearson's correlation test). **f** The represented kinases in MAPK signaling were overrepresented in the *LYN* amplification group (two-sided Wilcoxon signed-ranked test, BH-adjusted *p* < 0.05). The red frame shows the significant correlation between LYN and the represented kinases (two-sided Pearson's correlation test). **g** Kinase-substrate regulation network showing the substrates of LYN and MAPK1. The asterisk shows the overrepresented expression of the substrates in the LYN amplification group (two-sided Wilcoxon signed-ranked test, BH-adjusted *p* < 0.05). **h** Scatterplot showing the relationship between LYN and cell proliferation at the protein level (two-sided Pearson's correlation test). **i** A brief summary of the impacts of the chr8q gain in the IFT phase of the D subtype. ****p* < 1.0E−4, ***p* < 1.0E−3, **p* < 1.0E−2, *p* < 0.05, ns. >0.05. Source data are provided as a Source data file.

patterns and dominant pathways were detected in the subtypes and stages in DC.

To further explore the molecular features of the subtype-based panels, we analyzed the DEPs of the panels. As a result, the P1 (D subtype panel) proteins were enriched in the cell cycle (e.g., CDK1/2/4/

6 and CDKN2A), apoptosis (e.g., CASP3/8 and PARP1/9), and glycolysis (e.g., HK2, PGK1, etc.), while the P2 (G subtype panel) proteins participated in ECM signaling (e.g., COL1A1, LAMA1, etc.) and complement cascade (e.g., CFH and C4A) (Fig. 4b). The P3, patients had both the D and G subtypes, were characterized by the cell cycle, ECM signaling,

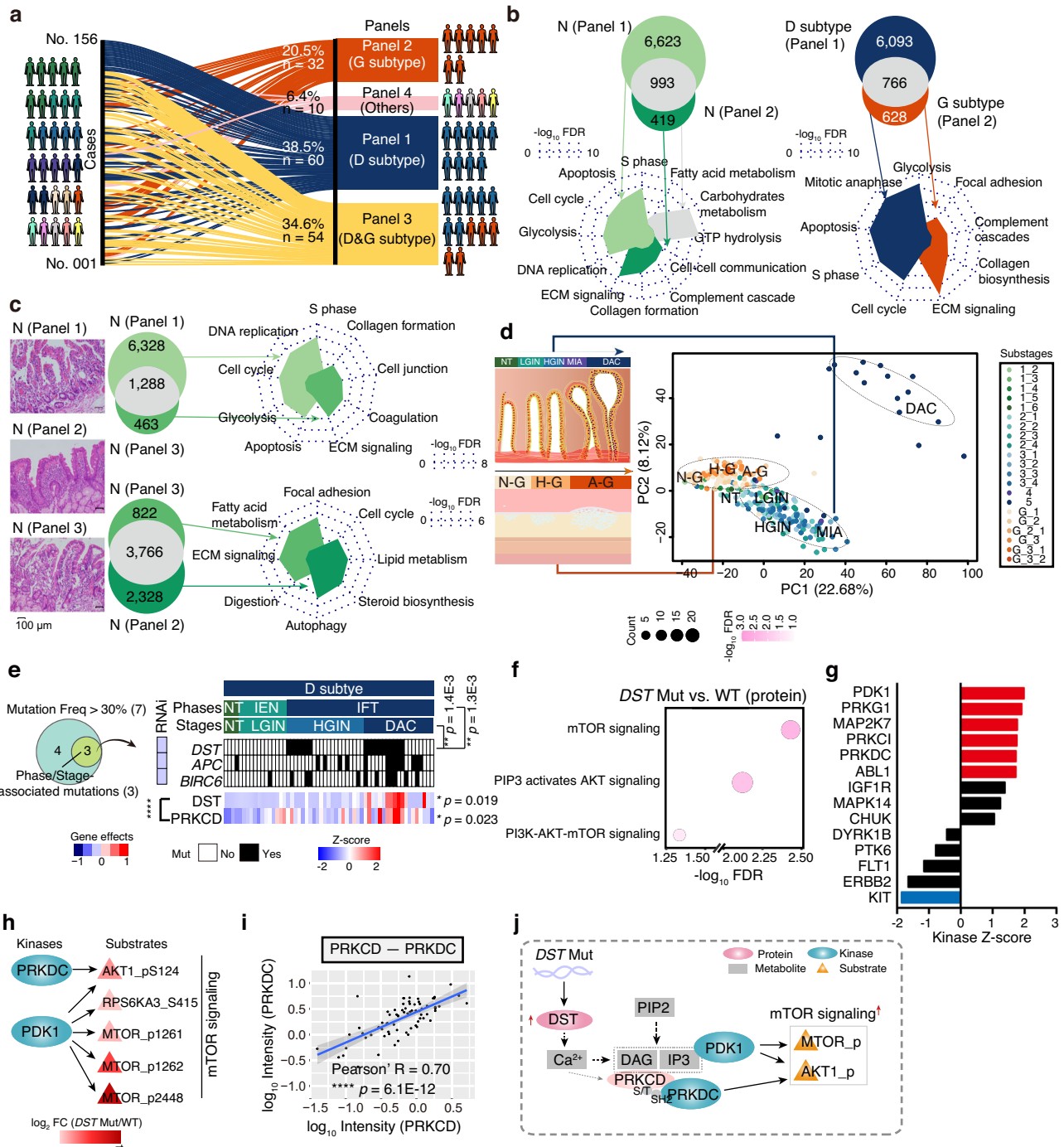

**Fig. 4 | The diverse characteristic origins and the staging waves in the progression of the D and G subtypes. a** Sankey diagram analysis of all the 156 cases classified into 4 panels. **b** Venn diagram depicting the specific driven pathways in the normal (left) and tumor (right) tissues of the P1 and P2. The green highlights the normal tissue. The blue and orange indicate the D and G subtypes, respectively. **c** The morphological feature (H&E staining) (left) and represented pathways (right) of normal tissue (stage 1) among the P1–P3. **d** PCA of the D and G subtypes samples. **e** Venn diagram (left) showing the phase/stage associated mutations in the D and G subtypes (two-sided Fisher's exact test) and heatmap (right) displaying the impacts of *DST* mutation on the protein-levels of DST and PRKCD (two-sided Wilcoxon signed-rank test). The square directs to a subset of patient samples in the D subtype used for WES (*n* = 120). **f** The represented pathways in the *DST* Mut group at the protein level. **g** KSEA of the kinase activity in the *DST* mutation group (*n* = 20) and WT group (*n* = 48). KSEA kinase-substrate enrichment analysis. **h** Kinase-substrate interaction network in mTOR signaling. **i** Scatterplot showing the relationship between $\log_{10}$ PRKCD and $\log_{10}$ PRKDC at the protein level (two-sided Pearson's correlation test). **j** A brief summary of the impacts of *DST* mutation in the DAC stage of the D subtype. DAC duodenal adenocarcinoma. ****$p < 1.0E-4$, ***$p < 1.0E-3$, **$p < 1.0E-2$, *$p < 0.05$, ns. >0.05. Source data are provided as a Source data file.

etc., which were observed in the P1 and P2, respectively (Supplementary Fig. 5f). The links of the findings in the proteomic clusters and subtype-based panels indicated the correlations between the proteomic clusters and the pathological stages/subtypes of DC. In addition, that consistency suggested the advances and applications of the proteome in stratifying patients and understanding cancer heterogeneity and phenotypes. Based on the proteomic methods and classification in the pathological layers, diverse subtypes/stages were

collected, promoting us to explore the origins and carcinogenesis of DC subtypes.

Interestingly, we found that the morphological features of the normal tissues were diverse among the P1–P3 (Fig. 4c), implying the different characterization of the normal tissues. We further analyzed the different molecular characterization of the P1–P3. As a result, the differences in the normal tissues were consistent with the corresponding tumor tissues of the P1–P3 at the protein level. For example, the highly expressed proteins of normal tissues in the P1 were enriched in the cell cycle, apoptosis, and glycolysis. In the P2, complement cascade and ECM signaling were dominant in the normal tissues (Fig. 4c). In addition, the P3 consisted of patients with both D and G subtypes, which shared the common normal tissues. Compared to the P1 (D subtype panel), ECM signaling (e.g., COL1A1 and LAMA1) was highlighted in the normal tissues of the P3 (Fig. 4c). While compared to the P2, cell cycle (e.g., CDK1 and CDK2) was observed in the normal tissues of the P3. Together, the different pathological features and molecular characterizations suggested the diversity in the D and G subtypes was associated with their origin in normal tissues.

## The staging and driver pathway waves in the progression of the D and G subtypes

In our cohort, stage-based classification during the D and G subtypes progression provided a chance to portray molecular profiles of D and G subtypes in a time-resolved mode at the multi-omics level. Principal component analysis (PCA) presented a clear boundary between the D and G subtypes, and among the diverse stages (Fig. 4d), further authenticating the specificity of the two major subtypes, and indicating dramatic alterations in the proteome patterns along with the carcinogenesis progression in both the D and G subtypes.

To further depict staging molecular models that drove carcinogenesis from early to progressive DC, we performed stage-based supervised clustering analysis and described the proposed driving pathways and characteristic molecules of different stages of the D and G subtypes. As a result, the D subtype was divided into five sequential waves (D1–D5): digestion pathway−AMPK signaling/ insulin resistance −cell cycle/DNA repair−apoptosis/aminoacyl-tRNA biosynthesis− mTOR signaling/EGFR signaling. In the D1 (normal tissue, NT stage), ACOX1 and APOA4 were generally overrepresented, indicating the remaining primary physiological functions of the duodenum in the early carcinogenesis stages. Enhanced proteome-levels of AMPK signaling (e.g., PRKAA1 and PPP2R5E) and insulin resistance (e.g., CAB39L and RAB13) were detected in the D2 (LGIN stage), and elevated cell cycle (e.g., CDK1 and RB1) and DNA repair (e.g., RFC1 and MSH3) in the D3 (HGIN stage), suggesting a response to exogenous stresses. Apoptosis (e.g., BID, CASP6, etc.) and aminoacyl-tRNA biosynthesis (e.g., CARS2 and AARS2) were predominant in the D4 (HGIN stage), suggesting cancer malignancy. In the D5 (DAC stage), mTOR signaling (e.g., AKT2 and RPTOR) and EGFR signaling (e.g., EGFR and GSK3B) were overrepresented (Supplementary Fig. 5g), implying the progression of cancer and tumor growth.

In the G subtype, we observed a different kinetic lineage with three sequential waves (M1–M3) in carcinogenetic progression. In the M1 (normal tissue, NT stage), the digestion pathway (e.g., LDLR and ACOX1) was dominant, which was similar to the D1 of the D subtype. ECM signaling (e.g., COL4A1 and LAMA1) and focal adhesion (e.g., ITGB1 and VWF) were prominent in the M2 (G1 and G2 stages). The M3 (G3 stage) was featured with complement cascade (e.g., C3, CFB, etc.) and platelet activation (e.g., ITPR1 and PLCB2) (Supplementary Fig. 5g), indicating the immune response was a dominant pathway during the progression of the G subtype.

Among the mutations in DC progression, we found that *DST* mutation, detected in the IFT phase (Fisher's exact test, $p = 1.3E−3$) and overrepresented in the DAC stage (Fisher's exact test, $p = 1.4E−3$), enhanced the protein-level of DST (Wilcoxon signed-ranked test, adjust. $p = 0.019$, *DST* Mut vs. WT ratio = 3.19), and had negative impacts on the proliferation of DC cell lines (Fig. 4e). Generally, DST plays a key role in calcium ($Ca^{2+}$) signal and increased $Ca^{2+}$ level[52], which is the activator in the transmit process from the phospholipid and diacylglycerol (DAG) to protein kinase C (PKC)[53]. In our study, we found that PRKCD, one of PKC[53], was overrepresented in the *DST* mutation group (Wilcoxon signed-ranked test, adjust. $p = 0.023$, *DST* Mut vs. WT ratio = 1.96), which exhibited a positive association with DST at the protein level (Pearson's $R = 0.60$, $p = 2.0E−6$) (Fig. 4e and Supplementary Fig. 5h).

To explore the impacts of *DST* mutation on DC progression, we performed GO/KEGG enrichment analysis on the overrepresented proteins in the *DST* mutation group, and found those proteins participated in mTOR signaling (e.g., MTOR, RPS6KA1/3) (Fig. 4f and Supplementary Fig. 5i). At the phosphoprotein level, 1608 phosphosites mapped to 885 phosphoproteins showed greater changes than corresponding protein abundance (Wilcoxon signed-ranked test, adjust. $p < 0.05$, *DST* Mut vs. WT ratio ≥2), and were significantly enriched in mTOR signaling (e.g., MTOR S1262 and DEPTOR S265) (Supplementary Fig. 5j, k). These findings indicated that *DST* mutation had positive impacts of mTOR signaling based on phosphorylation.

To investigate how *DST* mutation elevated phosphorylation and enhanced mTOR signaling, we performed KSEA of the phosphoproteome in the *DST* mutation group and WT group. As a result, the dominant kinases were found to be activated in the *DST* mutation group, including PDK1, PRKG1, MAP2K7, PRKCI, PRKDC, and ABL1, in which the protein-levels of PDK1 and PRKDC were overrepresented in the *DST* mutation group (Wilcoxon signed-ranked test, adjust. $p < 0.05$) (Fig. 4g and Supplementary Fig. 5l).

PDK1, a kinase upstream of mTOR, binds to inositol triphosphate (IP3) and activated PKB/AKT/mTOR signaling[54]. The kinase-substrate regulation network analysis revealed that the elevated substrates of PDK1 in our cohort, were involved in mTOR signaling, such as AKT1 S124, PRS6KA3 S415, MTOR S1261, etc. (Fig. 4h). These findings further implied the enhancement of mTOR signaling in the *DST* mutation group. PRKDC, one of the catalytic subunits of DNA-dependent protein kinase, of which the Sh2 domain can bind to the ser/thr sites and thus activates ser/thr kinases[55,56]. In our study, PRKDC showed significant association with PRKCD (Pearson's $R = 0.70$, $p = 6.1E−12$), and activated the substrate AKT1 (S124) (Fig. 4h, i). The integrated findings of the kinase-substrate analysis (PDK1 and PRKDC) suggested the overrepresentation of the mTOR signaling in the *DST* mutation group. Taken together, these results indicated that the overrepresented DST protein in the *DST* mutation group, enhanced the protein levels of the kinases (PDK1 and PRKDC), and thus activated mTOR signaling in the DAC stage (Fig. 4j), providing a reference database for the corresponding personalized medicine of DAC.

## Characterization of immune-based clusters in the DC subtypes

To evaluate the immune infiltration of the DC subtypes, we applied xCell (https://xcell.ucsf.edu)[57] to the 438 samples of the DC cohort, and deconvoluted immune, stromal, and microenvironmental cell signatures of DC. Consensus clustering identified four immune-based clusters as *Hot tumor* cluster, *Cold tumor* cluster, *Epithelial* cluster, and *B cells* cluster (Fig. 5a and Supplementary Fig. 6a; Supplementary Data 5a). Comparing with proteomic panels, we observed the P1 (D subtype panel) and P2 (G subtype panel) (Fisher's exact test, $p = 3.5E −6$) were prominent in the *Hot tumor/Epithelial* clusters and *Cold tumor/B cells* immune desert clusters, respectively (Supplementary Fig. 6b). Observation of the immune infiltration from xCell showed that higher stromal scores (Kruskal−Wallis test, adjust. $p = 9.6E−5$) in the non-epithelial components of DC, including G, N, and C subtypes (Supplementary Fig. 6c, d). During the carcinogenesis, stromal score was notably descended in the D subtype progression (Kruskal−Wallis test, adjust. $p = 2.0E−5$), which was not observed in the G subtype

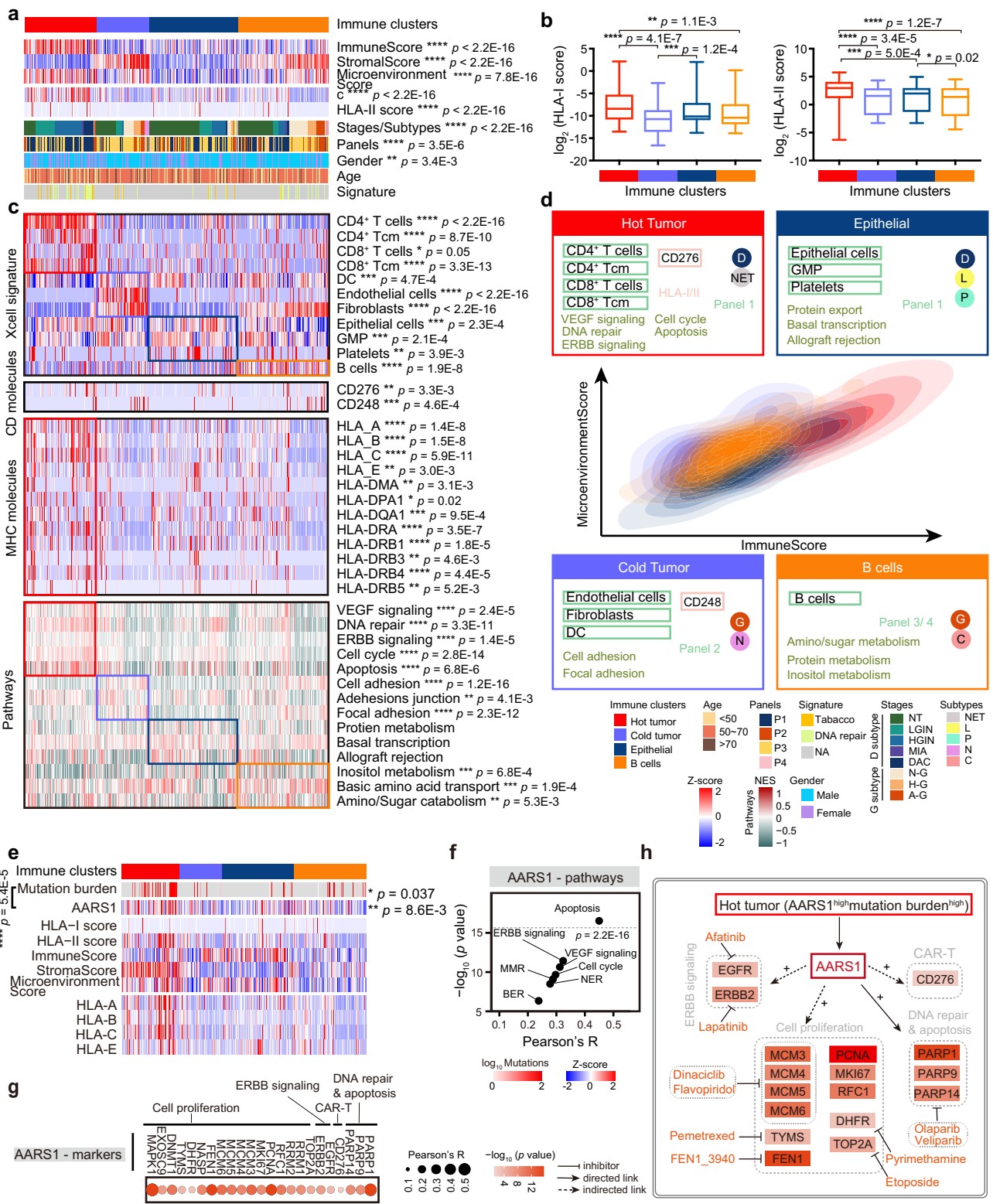

progression (Supplementary Fig. 6e). Of note, P1 was predominant in the *Hot tumor* cluster, and the P2 was overrepresented in the *Cold tumor* cluster and the *B cells* cluster. Consistently, pathway enrichment analysis revealed that the D subtype prominent pathways were dominant in the *Hot tumor* cluster, such as cell cycle, apoptosis, DNA repair, and so on (Fig. 5c, d). Moreover, the G subtype dominant pathways, such as focal adhesion, cell-cell adhesion, were overrepresented in the *Cold tumor* cluster. These results further illustrated the diverse

molecular characterization of the immune infiltration of DC was correlated with the subtypes.

In addition, the *Cold tumor* cluster, was characterized by the presence of dendritic cells (DCs), fibroblasts, and endothelial cells (Fig. 5c, d). CD248, tumor endothelial marker 1[58] and a modulator of fibrosis which featured by ECM deposition[59], was highly expressed in the *Cold tumor* cluster (Fig. 5c), implying the potential functions of anti-CD248 in the *Cold tumor* cluster.

**Fig. 5 | Characterization of immune-based clusters in DC subtypes. a** Heatmap illustrating cell type compositions and activities of the functional genes/proteins across the four immune clusters (Kruskal–Wallis test, BH-adjusted $p < 0.05$). The square directs to a subset of patient samples used for proteomic profiling ($n = 438$). **b** Boxplot showing the HLA-I score (left) and HLA-II score (right) in the four immune clusters (two-sided Wilcoxon signed-ranked test). $n$ (Hot immune cluster) = 105, $n$ (Cold tumor) = 75, $n$ (Epithelial cluster) = 129, $n$ (B cells cluster) = 129 biologically independent samples examined. Boxplot shows median (central line), upper and lower quartiles (box limits), 1.5× interquartile range (whiskers). **c** Proteome-based deconvolution of the immune-related cell signatures/proteins and dominant pathways in the four immune clusters (Kruskal–Wallis test, BH-adjusted $p < 0.05$). The GSVA score based on the global proteomic data for biological pathways overrepresented in different immune

clusters. The square directs to a subset of patient samples used for proteomic profiling ($n = 438$), as well as in Fig. 5a. GSVA gene set variation analysis. **d** Density contours of immune and microenvironment scores of four immune clusters. **e** Heatmap showing the correlation between AARS1 and TMB, immune infiltration, and HLA-I molecules (two-sided Pearson's correlation test). The square directs to a subset of patient samples used for WES ($n = 120$). TMB tumor mutation burden. **f** The (Pearson's) correlation between AARS1 and the predominant pathways in the *Hot tumor* cluster (two-sided Pearson's correlation test). **g** The correlation between AARS1 and makers of the predominant pathways in the *Hot tumor* cluster (two-sided Pearson's correlation test). **h** A brief summary of the impacts of AARS1 in the DC. ****$p < 1.0E-4$, ***$p < 1.0E-3$, **$p < 1.0E-2$, *$p < 0.05$, ns. >0.05. Source data are provided as a Source data file.

Comparably, the *Hot tumor* cluster with the highest immune score and microenvironment score, exhibited positive association with T cell activation, evidenced by the highest degree of CD4⁺ T cells, CD4⁺ Tcm, CD8⁺ T cells, and CD8⁺ Tcm (Fig. 5c). Besides, the highest HLA-I score and HLA-II score were detected in the *Hot tumor* cluster, evidenced by the expressions of the corresponding MHC molecules (e.g., HLA-B/C/E) (Fig. 5b, c). CD276 (also known as B7-H3), a member of B7 family[60], is broadly overexpressed by multiple tumor types on both cancer cells and tumor-infiltrating blood vessels[61], and was overrepresented in the *Hot tumor* cluster in the DC cohort. Recently, CD276 is identified as a compelling target and CAR-T targeting CD276 mediates significant anti-tumor activity in solid tumors[62]. The hyper-activation of T cell infiltration and overrepresentation of CD276 implicated the potential clinical strategies of CD276 and other CAR-T applications for the *Hot tumor* cluster. Taken together, we disclosed the different immune characterization in two major and five rare DC subtypes, and revealed the unique profiles of four immune clusters, providing the potential managements of CD248 in the *Cold tumor* cluster and CD276/CAR-T in the *Hot tumor* cluster.

Likely as the immune infiltration, the highest TMB was observed in the *Hot tumor* cluster (Kruskal-Wallis test, $p = 0.037$) (Supplementary Fig. 6f). Findings in many cancer types (e.g., bladder cancer and breast cancer) suggest the TMB may predict clinical response to immune checkpoint inhibitors[63]. In our study, the TMB exhibited positive association with the DNA repair signature (Pearson's $R = 0.44$, $p = 1.7E-4$), indicating MSI in the *Hot tumor* cluster (Supplementary Fig. 6g). The referred significant proteins in the DNA repair panel, especially AARS1 and TARS2, were significantly and highly expressed in the *Hot tumor* cluster, and were overrepresented in the tumor tissue both in the D and G subtypes, implying their potential value in the clinic strategy (Supplementary Fig. 6h). Furthermore, AARS1 was growingly enhanced in the D subtype progression (Kruskal–Wallis test, adjust. $p = 2.6E-5$) (Supplementary Fig. 6i), and showed significantly positive correlation with TMB (Pearson's $R = 0.45$, $p = 5.4E-5$) (Supplementary Fig. 6j), indicating the involvement of AARS1 in the immune infiltration of DC. To further investigate the functions of AARS1, we then performed correlation analysis between AARS1 and immune infiltration. As well as immune score, stroma score, and microenvironment score, we found that AARS1 represented positive correlation with HLA-I score and HLA-II score, which was correlated with TMB (Fig. 5e and Supplementary Fig. 6k, l). In addition, AARS1 showed significantly positive association with the dominant pathways of the *Hot tumor* cluster, including apoptosis, base excision repair, mismatched repair, VEGF signaling, ERBB signaling, etc. (Fig. 5f, g). These findings reflected that AARS1 was responsible for cancer-associated signaling pathways (Fig. 5h), implying the necessary of exploiting the utility of AARS1 in DC therapy.

## AARS1 promotes cancer cell proliferation and invasion through non-canonical function

DNA damage was proved to be the driving factor in DC progression, in which AARS1 displayed a key role, and could be perspective for DC

(Fig. 5h). In addition, we found the expression level of AARS1 increased significantly in tumor tissues, and growingly enhanced during DC progression (Fig. 6a).

Using western blot and immunohistochemistry (IHC), we confirmed that AARS1 protein levels were enhanced in tumor tissues during DC carcinogenesis ($t$-test, $p = 6.7E-8$) (Fig. 6b and Supplementary Fig. 7a; Supplementary Data 6a). As controls, the overall expression levels of TARS1 and SARS1 were not altered (Fig. 6b and Supplementary Fig. 7a). Overexpressing AARS1 promoted, while knocking down AARS1 inhibited cell proliferation (Fig. 6c, d; Supplementary Data 6b, c) and cell invasion capacities (Fig. 6e; Supplementary Data 6d) in both Hutu80 cells and WDC-1 cells. However, overexpression of other aminoacyl-tRNA synthetases, including SARS1 and TARS1, did not affect cell proliferation and invasion (Supplementary Fig. 7b, c; Supplementary Data 6e, f). These results suggested that the oncogenic role of AARS1 was not due to protein translation. To investigate the potential non-translational role of AARS1, we used tandem affinity purification (TAP) to identify AARS1-interacting proteins in Hutu80 cells. As a result, a total of 652 differential proteins were detected in cells, which were highly expressed in the D subtype ($p < 0.05$) (Supplementary Fig. 7d). Consistently, GO/KEGG enrichment analysis revealed that those 652 proteins participated in DNA damage repair-related pathways, which were the dominant pathways in the D subtype (Fig. 6f and Supplementary Fig. 7e), further indicating the roles of AARS1 in DC progression.

Among the 652 proteins, PARP1, one of poly-ADP-ribosyltransferases that mediates poly-ADP-ribosylation of proteins and plays a key role in DNA repair[64], was the prior preferential target that was interacted with AARS1 (Supplementary Fig. 7f, g). Accordingly, the interaction between AARS1 and PARP1 was confirmed by the results of co-immunoprecipitation assays using either exogenous AARS1 and PARP1 in cultured Hutu80 cells and WDC-1 cells (Fig. 6g) or endogenous AARS1 and PARP1 in DC tissues (Supplementary Fig. 7h). Besides, we validated that AARS1 was localized in both the cytosol and nucleus, supporting our observation that AARS1 interacted with PARP1 (Supplementary Fig. 7i). Furthermore, overexpression of AARS1 led to notably decreased cellular poly ADP ribosylation (PARylation) (Fig. 6h), and decreased PARP1 specific activity, which was measured using a PARP1 enzymatic activity assay (Fig. 6i; Supplementary Data 6g). In addition, the overexpression of other ARS families (e.g., TARS1 and SARS1) would not decrease PARP1 activity (Fig. 6i). Therefore, among of ARS families, PARP1 interacted with AARS1, reducing the reduction of PARP1 activity. Moreover, we performed PARP1 activity colorimetric assay in vitro, by adding purified AARS1 into the reaction mix. As a result, we found that overexpression of AARS1 significantly decreased the activity of PARP1 ($t$-test, $p = 2.6E-3$) (Supplementary Fig. 7j; Supplementary Data 6h). These results indicated that increased AARS1 correlated with decreased PARP1 activity in cells.

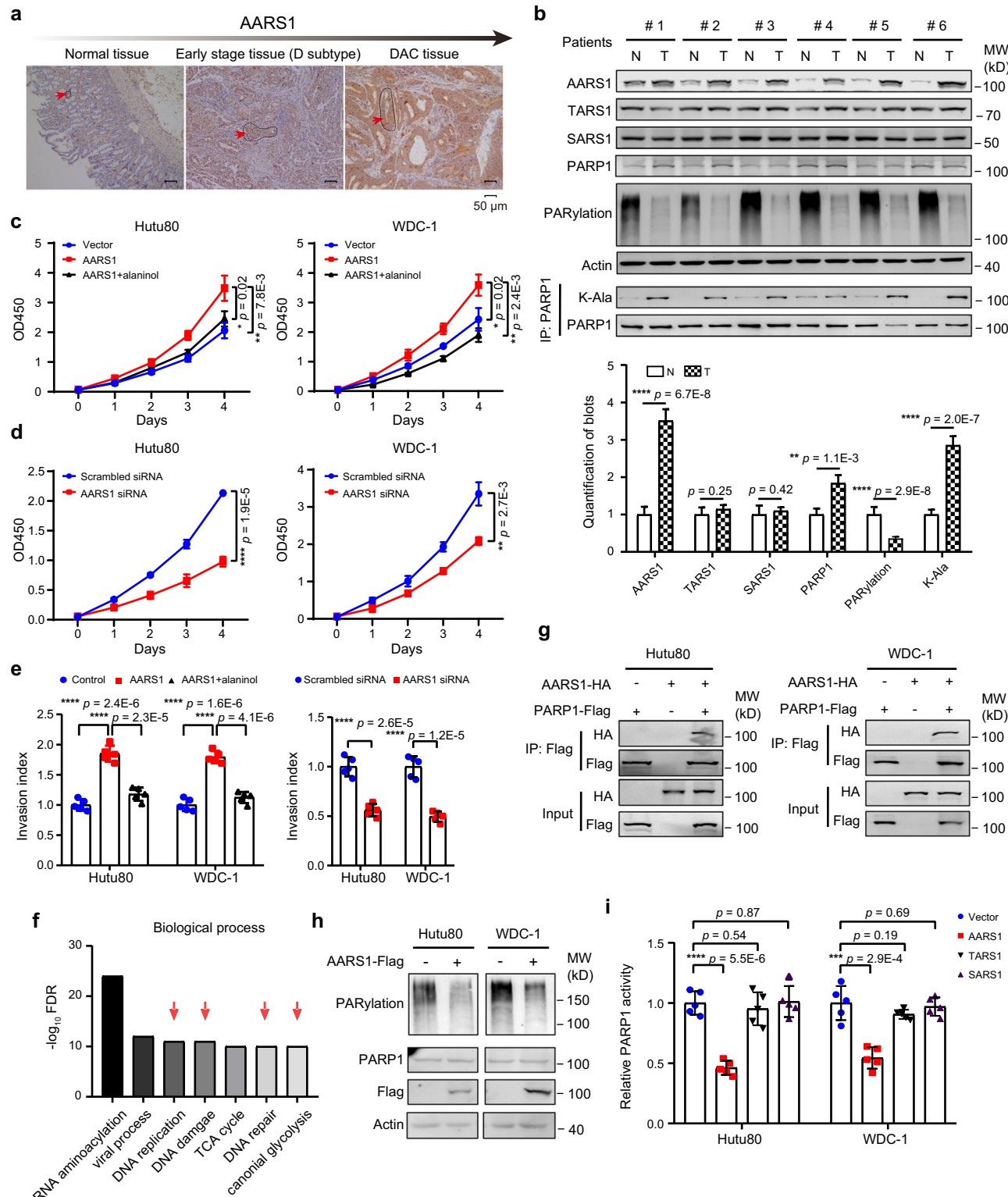

### AARS1 inhibits PARP1 and cell apoptosis and blocking K-Ala activated PARP1 and suppressed tumor growth

It has been well established that PARP1 plays an important role in the cell response to DNA damage[65] and the inductive effect on cell apoptosis[66]. We confirmed that cell apoptosis decreased significantly in AARS1-overexpressing cells (Fig. 7a; Supplementary Data 6i). The genotoxic reagent camptothecins (CPT) was able to induce DNA damage and increase cell apoptosis[67]. Overexpression of AARS1 in CPT-treated cells led to a reduced cell apoptosis rate

(Fig. 7a; Supplementary Data 6i), although the DNA damage level increased, as evidenced by the levels of γ-H2AX detected using immunofluorescence staining (Fig. 7b) and western blot and based on DNA damage detected using a comet assay (Supplementary Fig. 8a, b; Supplementary Data 6j). Combined with the results that AARS1 expression correlated with decreased PARP1 activity, these results indicated that AARS1 overexpression may contribute to promoting cell proliferation and invasion through reducing cell apoptosis.

**Fig. 6 | AARS1 promotes cancer cell proliferation through non-canonical function. a** IHC staining of the expression level of AARS1 from normal to advanced stage tissue. **b** Overrepresentation of AARS1 and PARP1 in tumor tissue is detected by western blot analysis (*n* = 12 repeats per group) (two-sided Student's *t*-test, mean ± SD). N = tumor-adjacent tissue, T = tumor tissue. The left panel shows western blot analysis of AARS1, TARS1, SARS1, PARP1, PARylation, actin, and lysine alanylation of PARP1 in tumor tissues and tumor-adjacent normal tissues. The bottom panel shows the quantified western blot results of 12 pairs of samples. The results of the other 6 pairs of samples are provided in Extended Data Fig. 7a. **c** The impacts of AARS1 overexpression (OE) on Hutu80 cells and WDC-1 cells' proliferation (*n* = 5 repeats per group) (two-sided Student's *t*-test, mean ± SD): vector (control), OE-AARS1, and OE-AARS1-alaninol. The value of OD450 and days are shown on the *y*- and *x*-axes, respectively. **d** The impacts of AARS1 knock-down (KO)

on Hutu80 cells and WDC-1 cells' proliferation (*n* = 5 repeats per group) (two-sided Student's *t*-test, mean ± SD): scrambled siRNA and AARS1 KO. **e** The effects of AARS1 OE (left)/KO (right) Hutu80 cells and WDC-1 cells invasion (*n* = 5 repeats per group) (two-sided Student's *t*-test, mean ± SD). **f** Gene Ontology enrichment analysis of AARS1-interacting proteins in biological processes. These proteins are identified by a proteomics survey using tandem affinity purification. **g** Co-immunoprecipitation assay showing that exogenous AARS1 and exogenous PARP1 interacted in cultured Hutu80 cells and WDC-1 cells. **h** The impacts of AARS1 on cellular PARylation level. **i** Activities of PARP1 immunoprecipitated from cells transfected with different aminoacyl-tRNA synthetases (*n* = 5 repeats per group) (two-sided Student's *t*-test, mean ± SD). ****$p < 1.0E-4$, ***$p < 1.0E-3$, **$p < 1.0E-2$, *$p < 0.05$, ns. >0.05. Source data are provided as a Source data file.

We surveyed how AARS1 inhibited PARP1. In addition to mediating protein translation, AARS1 can catalyze the alanyl-modification in protein lysine residues and transferred its signal by altering the functions of modified proteins[68]. Elevated AARS1 expression or alanine levels led to dose-dependent increases in lysine-alanylation (K-Ala) of total protein (Supplementary Fig. 8c) and K-Ala of PARP1 (Supplementary Fig. 8d), as evaluated through western blotting performed by a custom anti-K-Ala antibody (Supplementary Fig. 8e). Knocking down AARS1 abrogated the effect of alanine in forming K-Ala on PARP1 (Fig. 7c). In tumor and adjacent normal tissues, we confirmed that AARS1-overexpressing tumors exhibited K-Ala modification on PARP1 (Fig. 6b).

We screened for K-Ala modification sites in PARP1 protein, employing liquid chromatography resolution followed by tandem mass spectrometry analysis. Four lysine sites were identified as being alanylated in PARP1 (Supplementary Fig. 8f, g), supporting the possibility that K-Ala affects PARP1 activity. Using an in vitro assay, we validated that AARS1 catalyzed the K-Ala modification in synthetic PARP1 K621 peptides (Supplementary Fig. 8h). However, an aminoacyl-AMP formation-defective AARS1$^{R751G}$ mutant failed to form K-Ala in synthetic peptides (Supplementary Fig. 8h). To identify the possible enzyme responsible for K-Ala removal, we screened the K-Ala-removing capacities of nuclear localized NAD$^+$-dependent sirtuins, including SIRT1, 2, 6, and 7, in cultured Hutu80 cells, as these deacetylases might have deaminoacylase capacities. We found that only overexpression of SIRT6, but not other sirtuins, significantly increased the cellular PARylation in cells, suggesting that SIRT6 could potentially remove the K-Ala on PARP1 (Supplementary Fig. 8i). In vitro, we validated that SIRT6 could catalyze the de-alanylation of synthetic K-Ala PARP1 peptides (Fig. 7d). Moreover, we confirmed that although overexpression of AARS1 led to increased K-Ala levels of PARP1, exogenous expression of SIRT6 in cells could decrease the K-Ala levels of PARP1 (Fig. 7e). These results demonstrated that PARP1 is subjected to AARS1-mediated K-Ala modification.

To further investigate the function of K-Ala modification on PARP1, we constructed an alanylation memetic lysine (23, 414, 418, 621)-to-alanine switch mutant PARP1 (KA mutant) and a non-alanylation memetic lysine (23, 414, 418, 621)-to-arginine switch mutant PARP1 (KR mutant). We found that the PARP1 KA mutant exhibited decreased binding affinity to DNA, while the PARP1 KR mutant had similar binding affinity to DNA compared to the wild-type PARP1 (Fig. 7f; Supplementary Data 6k). Through an in vitro assay, we observed that the KA mutant had decreased PARylation activity, while the KR mutant did not influence the PARylation activity compared to wild-type PARP1 (Fig. 7g). To inhibit the AARS1-induced oncogenic effect on cells, we identified the alanine structural analog alaninol as an AARS1 inhibitor. Alaninol inhibited cellular protein K-Ala and increased the PARylation level (Fig. 7h). When cultured media were supplemented with alaninol, the effects of AARS1 overexpression, including decreased cell apoptosis, enhanced cell proliferation and invasion were reversed. In addition, we noted that increased AARS1 expression

promoted the xenograft growth of tumor cells, especially in HU-treated cell xenografts in nude mice, whereas inhibition of K-Ala with alaninol delayed the xenograft growth of tumor cells (Fig. 7i).

## Discussion

Duodenum is a relatively privileged organ that rarely develops malignant tumors. The rarity and complicated subtypes of DC cause unique diagnostic and therapeutic challenges. The landscape on the molecular level, such as the genome, proteome, and phosphoproteome, could serve as the basis for understanding the mechanism of the carcinogenesis and as a resource for seeking potential diagnostic and therapeutic targets. Generally, the (sub)stage of lesions during the carcinogenesis of DC have complicated histopathological and morphological characteristics, including the benign polyp (NT stage), adenoma (e.g., LGIN and HGIN stage) to adenocarcinoma (e.g., MIA and DAC stage)[10]. Considering the different operation strategies to meet the risks of the patients at the different (sub)stages in the clinic, the revelation of the dominant mutations and key events may be conductive to exploring the molecular mechanism during DC carcinogenesis and providing precise medication in the clinic of DC. However, the pathological mechanism of the (sub)stage of the lesions in DC progression are yet unclear. In this study, we employed 438 samples from 156 DC cases covering 2 major and 5 rare subtypes, in which the D and G subtypes (2 major subtypes) were further classified into 17 and 6 substages, respectively, providing a chance for exploring the origins and proteogenomic characterization of DC subtypes and (sub)stages, and tracking the key events during DC carcinogenesis in a time resolved mode.

To achieve deep proteome coverage of such a tiny FFPE tissue volume, pressure cycling technique (PCT) would be an available method and effectively minimize sample loss[69], while the lowest operable weight (about 300 μg) is still higher than that in our study (under 100 μg). On the basis of the method of tris-2-carboxyethylphosphine (TCEP) extraction protein, with high internal standard recovery (>90%)[70], we have achieved deep proteome and phosphoproteome coverage of the trace samples in early-stage gastrointestinal cancer[71], and thus the method of TCEP extraction protein was applied to explore the molecular characterization in DC progression.

Irrespective to that DC followed the same transition pattern of CRC, from adenoma to adenocarcinoma[10], the increasing incidence and complex subtypes of DC impose needs for specific strategy urgently, instead of approaching the similar manner of CRC in the clinic. Relatively, the D and NET are malignant subtypes[72,73], and comparably, the G subtype is benign[74]. Observation of the globally akin among the D/NET/L subtypes, and among the G/N/C subtypes (Supplementary Fig. 5d), indicated that the D/NET/L/P subtypes were relatively malignant, and the G/N/C subtypes were comparably benign in DC, providing a guidance in clinical intervention. Banck et al. found the high activity of mTOR signaling in NET[9]. The WES-based analyses disclosed that the DAC stage mutation of *DST*, displayed positive

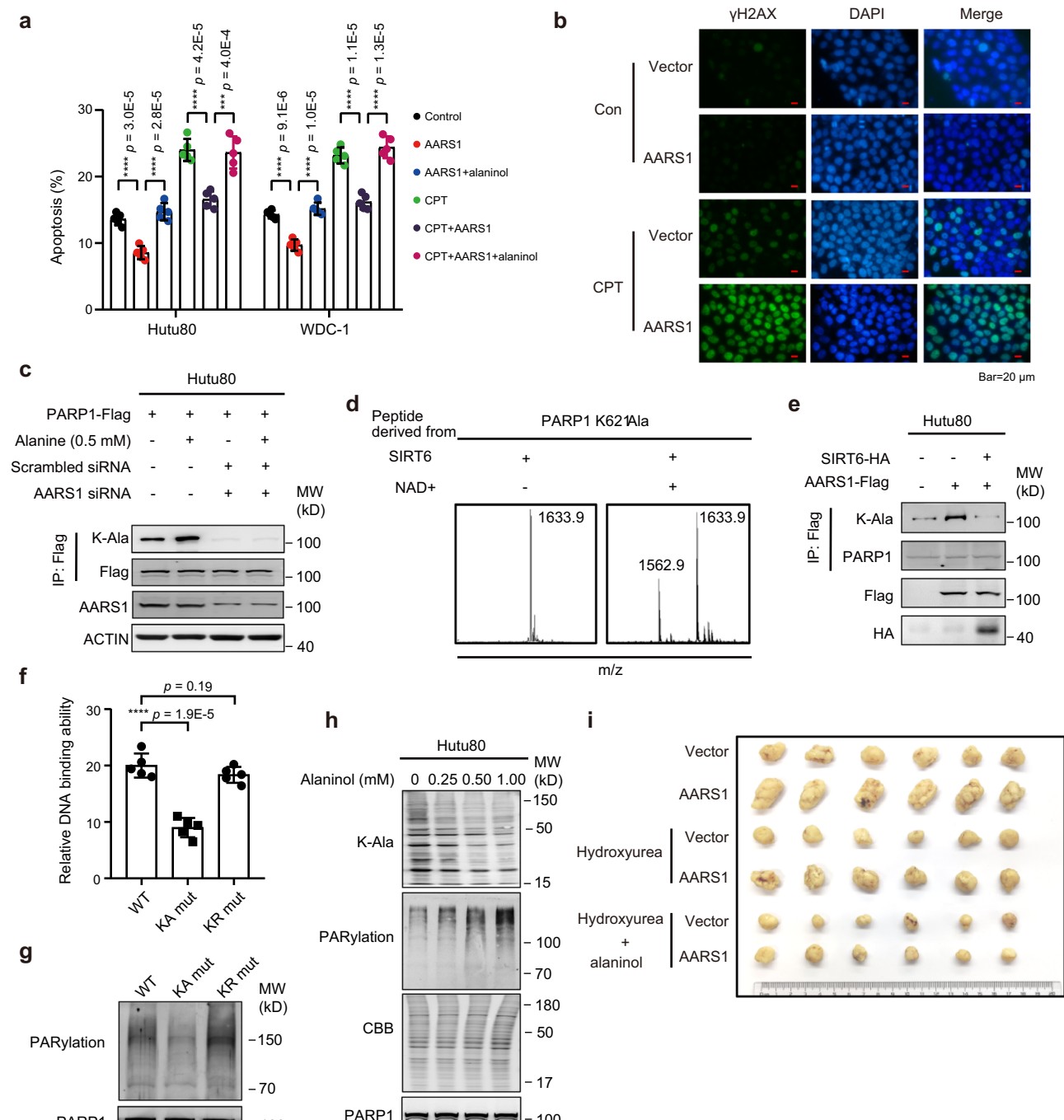

**Fig. 7 | Lysine alanylation of PARP1 decreases cell apoptosis by inhibiting PARP1. a** The impacts of AARS overexpressing (OE) in Hutu80 cells and WDC-1 cells apoptosis (*n* = 5 repeats per group) (two-sided Student's *t*-test, mean ± SD): OE-AARS1, OE-AARS1-alaninol, CPT, OE-AARS1-CPT, and OE-AARS1-CPT-alaninol. CPT camptothecins. **b** Immunofluorescence staining of γH2AX with various treatments: OE-AARS1 and OE-AARS1-CPT. The scale bar indicates 20 μm. **c** The effects of AARS1 and PARP1 on K-Ala levels. **d** The capacity of SIRT6 to dealanylate the synthetic PARP1 K621-Ala peptide. **e** K-Ala levels in cells with different treatments. **f** DNA binding affinities of PARP1 and PARP1 mutants (*n* = 5 repeats per group, two-sided Student's *t*-test, mean ± SD). K-Ala: lysine-alanylation. **g** PARP1 activities in PARP1 WT and PARP1 mutants. **h** The impacts of alaninol on cellular K-Ala and PARylation levels. **i** Tumor sizes of cell xenografts with different treatments in normal nude mice or alaninol-fed nude mice. *n* = 6 mice per group. ****\**p* < 1.0E−4, \*\*\**p* < 1.0E−3, \*\**p* < 1.0E−2, \* *p* < 0.05, ns. >0.05. Source data are provided as a Source data file.

impacts on the proteome-levels of DST and PKC, and thus activated the mTOR signaling. Thus, the integration indicated that the anti-mTOR therapy (e.g., everolimus and rapamycin)[75,76] might be medicative for the malignant subtypes, such as the D/NET/L subtypes.

Several published studies pointed that smoking is a known risk factor in a number of gastrointestinal (GI) diseases, including duodenal ulcer[77] and GI cancer (e.g., ESCA, CRC, and GC)[78]. In addition, tobacco usage has important implications regarding gut environment and

various small intestine disorders, including functional dyspepsia[6], irritable bowel syndrome[7] (Gastroenterology, 2017, PMID: 27725146), coeliac disease[79], and duodenal ulcer disease[8]. However, there is no literature directly suggest tobacco usage increased the risk of duodenal adenocarcinoma. In our cohort, we found more patients (60% vs 18.8%), especially the D subtype patients, with smoking habits were observed in the Tobacco signature group. Integration with the findings in our cohort and the impacts of smoking on GI (e.g., ESCA, CRC, and

GC), indicated that smoking or tobacco usage might be a risk for DC, especially for the D subtype. Furthermore, CYPs were overrepresented in the Tobacco signature group, suggesting the potential mediation of CYPs for the patients with smoking habit.

Familial adenomatous polyposis (FAP) and hereditary non-polyposis colorectal cancer (HNPCC), two rare inherited disorders[80], are infrequent in duodenum. In our cohort, the samples of FAP and HNPCC were also collected and mainly included in the P3, patients had both D and G subtypes. That implied that the FAP and HNPCC had the similar traits of P3, which was characterized by cell cycle, ECM signaling, etc. Whereas the limited samples of the FAP and HNPCC exhibited a barrier for presenting comprehensive characterization, more samples are needed for further analysis in the future.

The immune based investigation delineated the immune characteristics of DC, and disclosed the potential actions of anti-CD248 in the *Cold tumor* cluster, and anti-CD276 or CAR-T/CD276 in the *Hot tumor* cluster, providing a clinic direction for DC. ARS families have been implicated in the etiology of various disorders[41], and can catalyze the posttranslational addition of amino acids to lysine residues in proteins[68]. Interestingly, AARS1, one of ARS families and functioned in the DNA repair panel, showed positive correlation with the TMB and the immune infiltration. In addition, AARS1 was positively associated with the *Hot tumor* cluster dominant pathways, such as DNA repair, cell cycle, apoptosis, etc. PARP1 inhibitory is observed in homologous recombination cancer[81,82]. In this study, we built a connection between AARS1 and PARP1 in DC development, and discovered that overexpressed AARS1 catalyzed K-Ala modification on PARP1, resulting in an inhibitory effect on cell apoptosis. Thus, we speculated AARS1, growingly increased during the tumor progression of DC and along with enhanced DNA damage, was a potential target in the clinic strategy of DC.

However, limits are still represented in our study. The trace amounts of samples restricted the coverage at multi-omics level, which is also a challenge for transcriptomic analysis. The possible bias was the limits for proteome, and showed association with TF/plasma membrane- related proteins, which might be the challenges for understanding the characteristics of DC. The advances of MS with more in-depth coverage of identifies in proteome needed to dissect the bias in proteome, providing a more comprehensive proteomic landscape. In addition, to explore the correlation between tobacco usage and DC, more evidences are needed from other cohorts or epidemiologic studies with more DC patients for further analysis.

In summary, our study presented a comprehensive multi-omics landscape of DC. We discovered the characteristic origins and diverse carcinogenesis tracks, and illustrated the kinetic waves of the dominant cancer pathways of the D and G subtypes via integrative proteogenomic analysis. Proteogenomics elucidated the positive impacts of *LYN* amplification at the chr8q gain on MAPK signaling in the IFT phase, and revealed the functions of *DST* mutation on mTOR signaling in the DAC stage (Supplementary Fig. 9). We illuminated the drug-targetable AARS1 in the TMB/immune infiltration in DC, which was demonstrated the value of this multi-omics mapping strategy to suppress tumor growth. We believe this study provides insights into understanding DC and enables advances in understanding its mechanism and diagnostics, delivering a useful resource for potential therapeutic approaches and personalized medicine for DC.

## Methods
### Patient samples of early DC cohort
**Construction of the DC cohort.** Two hundred early-stage DC (EDC) patients underwent ESD therapy from 2012 to 2019 at Zhongshan hospital, Fudan University, and received no prior anti-cancer treatments. Among of the EDC patients, 23 were precluded duo to the unavailability of their normal tissue samples, and 41 patients failed to pass the pathological quality check. Thus, 136 EDC patients were

eligible for the establishment of the intended study cohort. In addition, 20 advanced-stage DC patients, without prior anticancer treatments, were randomly enrolled for the first visit from 2012 to 2019 at Zhongshan hospital, Fudan University. Therefore, a total of 156 DC patients were selected to construct DC cohort in this study. The present study complied with the ethical standards of Helsinki Declaration II and approved by the Research Ethics Committee of Fudan University Zhongshan Hospital (B2019-200R). Written informed consent was provided by all patients before any study-specific investigation. Clinical information of individual patient is summarized in Supplementary Data 1 (Table 1).

All the samples, collected in our cohort, were up to the following criteria. Firstly, all the samples were well preserved and systematically evaluated to confirm the histopathologic diagnosis and any variant histology according to the WHO classification[11] by more than two expert gastrointestinal pathologists, who determined the acceptable tissues segment based on the tumor content (>95%), the presence and extent of tumor necrosis (<5%), and signs of invasion into the muscularis propria. Secondly, according to the clinical sample collection procedures in CPTAC and other published studies[83], the following criteria were used, (1) successful extraction in DNA and protein; (2) no tumor cells in the normal tissue.

All cases were classified according to the 4th and 5th edition WHO Classification of Digestive system. All substages and subtypes in the cohort were assessed by more than two experienced gastrointestinal pathologists on the basis of hematoxylin and eosin (H&E)-stained sections, who would mark the tumor purity according to the proportion of tumor cells. In our cohort, the tissues were classified into surface epithelial (D subtype) and non-surface epithelial components (G/NET/L/P/N/C subtypes). The D subtype was then classified into 5 stages, including NT stage, LGIN stage, HGIN stage; MIA stage, and DAC stage. The NT stage included normal duodenal epithelial, mild hyperplasia (1_1), heterotopic gastric fundus gland (1_2), gastric antrum gland (1_3), duodenal epithelial hyperplasia (1_4), P-J polyp (1_5), hyperplastic polyp (1_6). The LGIN stage included tubular adenoma with mild atypia (2_1), tubular adenoma with mild-middle atypia (2_2), tubulovillous adenoma with mild atypia (2_3), and tubulovillous adenoma with mild-middle atypia (2_4). The HGIN stage included tubular adenoma with middle-severe atypia (3_1), tubular adenoma with severe atypia (3_2), tubulovillous adenoma with middle-severe atypia (3_3), and tubulovillous adenoma with severe atypia (3_4) (Table 2). The non-surface epithelial components were divided into G subtype ($n = 97$), P subtype, N subtype), L subtype, C subtype, and NET subtype. The G subtype, one subtype of non-surface epithelial components, comprised normal (N−G), hyperplasia (H−G), and adenoma (A−G) (Table 3). Finally, a total of 438 samples were collected and prepared for the early and progressive duodenal cancer cohort.

Considering that sometimes with only 1 or 2 cases for each substage, such as mild hyperplasia and hyperplastic polyp, we integrated the substages into stages, and used the stage-based classification for the proteogenomic analysis in this study (e.g., Figs. 2−4), including the NT, LGIN, HGIN, MIA, DAC stages in the D subtype, and the N−G, H−G, and A−G stages in the G subtype.

**FFPE sampling and processing.** All the FFPE specimens were prepared and provided by Zhongshan Hospital, Fudan University. For clinical sample preparation, slides (10 μm thick) from FFPE blocks were macro-dissected, deparaffinized with xylene and washed with ethanol. One 3 μm-thick slides from FFPE blocks were sectioned for H&E staining. All the selected specimens were scraped according to the substages, and were stored at −80 °C until further processing.

**Laser-capture microdissection (LCM)**
In our cohort, we applied LCM[25] to dissect the sections of substage/subtype samples precisely. All FFPE specimens were deparaffinized

with xylene and rehydrated through graded alcohols and water. The H&E Sections were stained with Mayer's haematoxylin (Sigma) and dehydrated through graded alcohols and xylene. Before microdissection, FFPE specimens were sectioned with a microtome (10 µm thick) and mounted from FFPE blocks were micro-dissected with a Leica LMD 6500 laser microdissection system. According to the WHO classification, all samples were systematically evaluated to confirm the histopathologic diagnosis and variant histology by more than two expert gastrointestinal pathologists, and then an area of $1-5 \times 10^6$ µm$^2$ was collected (derived from dissected area × slide thickness/average mammalian cell volume of 2000 µm$^3$, BNID 100,434) (Supplementary Fig. 1b). To avoid cross-tissues contamination among the substages or subtypes, we collected the tissues at the core area in each substage. The substage/subtype samples were collected in 1.5-ml tubes and kept in storage at −80 °C until further processing. The methods are also applied in other published proteogenomic studies, such as in ovarian cancer[84] and glioblastoma[85].

## Whole-exome sequencing (WES)

WES were all performed by Novogene Co., Ltd. DNA from FFPE tumor tissue samples were collected and used for WES, and matched germline DNA was obtained from non-tumor tissue samples. One hundred and 20 samples from 47 cases were analyzed by WES including the D/G/P/N/C subtypes. Paired-end sequencing (PE150) was performed on an Illumina HiSeq with a 135× target depth (mean) and 15.5 G volume (mean) of 120 raw data (Supplementary Data 2a). The resulting sequence libraries (the paired-end sequence and insert DNA between two ends) were quantified with a Qubit 2.0 (Thermo Fisher) and insert size was determined using an Agilent 2100 Bioanalyzer. Base calling was used to obtain the raw data (sequenced reads) from the primary image data.

## DNA extraction and DNA qualification

One hundred and twenty samples from 47 DC patients were conducted for WES in our cohort. All the samples were firstly dewaxing with dimethylbenzene, and then DNA degradation and contamination were monitored on 1% agarose gels. DNA concentration was measured byQubit® DNA Assay in Qubit® 2.0 Flurometer (Invitrogen, USA). A total amount of at least 0.6 µg genomic DNA per sample was used as input for DNA sample preparation.

## Library preparation

A total amount of 0.6 µg genomic DNA per sample was used as input for DNA sample preparation. Sequencing libraries were generated using Agilent SureSelect Human All Exon kit (Agilent Technologies, CA, USA) following manufacturer's recommendations and index codes were added to each sample.

Fragmentation was carried out by hydrodynamic shearing system (Covaris, Massachusetts, USA) to randomly generate 180–280 bp fragments. Remaining overhangs were converted into blunt ends via exonuclease/polymerase activities. After adenylation of 3′ ends of DNA fragments, adapter oligonucleotides were ligated. DNA fragments with ligated adapter molecules on both ends were selectively enriched in a PCR reaction. Then, libraries hybridize with liquid phase with biotin labeled probe, then use magnetic beads with streptomycin to capture the exons of genes. Captured libraries were enriched in a PCR reaction to add index tags to prepare for sequencing. Products were purified using AMPure XP system (Beckman Coulter, Beverly, USA) and quantified using the Agilent high sensitivity DNA assay on the Agilent Bioanalyzer 2100 system.

The clustering of the index-coded samples was performed on a cBot Cluster Generation System using Hiseq PE Cluster Kit (Illumina) according to the manufacturer's instructions. After cluster generation, the DNA libraries were sequenced on Illumina Hiseq platform and 150 bp paired-end reads were generated.

## Quality control of data processing and analysis

Paired-end sequencing (PE150) was performed on an Illumina HiSeq (Illumina Novaseq 6000). The resulting sequence libraries (the paired-end sequence and insert DNA between two ends) were quantified with a Qubit 2.0 (Thermo Fisher) and insert size was determined using an Agilent 2100 Bioanalyzer. The original fluorescence image files obtained from Hiseq platform are transformed to short reads (raw data) by base calling and these short reads are recorded in FASTQ format, which contains sequence information and corresponding sequencing quality information. Base calling was used to obtain the raw data (sequenced reads) from the primary image data.

Quality control:

(1) Discard a paired read if either one read contains adapter contamination (>10 nucleotides aligned to the adapter, allowing ≤ 10% mismatches;

(2) Discard a paired read if more than 10% of bases are uncertain in either one read;

(3) Discard a paired read if the proportion of low quality (Phred quality < 5) bases is over 50% in either one read.

All the downstream bioinformatic analyses were based on the high-quality clean data, which were retained after these steps. At the same time, QC statistics including total reads number, raw data, raw depth, sequencing error rate, percentage of reads with Q30 (the percent of bases with phred-scaled quality score greater than 30) and QC content distribution were calculated and summarized.

## Reads mapping to reference sequence

Valid sequencing data was mapped to the reference human genome (UCSC hg19) by Burrows–Wheeler Aligner (BWA) software[86] to get the original mapping results stored in BAM format. If one or one paired read(s) were mapped to multiple positions, the strategy adopted by BWA was to choose the most likely placement. If two or more most likely placements presented, BWA picked one randomly. Then, SAMtools[87] and Picard (http://broadinstitute.github.io/picard/) were used to sort BAM files and do duplicate marking, local realignment, and base quality recalibration to generate final BAM file for computation of the sequence coverage and depth. Mapping step was very difficult due to mismatches, including true mutation and sequencing error, and duplicates resulted from PCR amplification. These duplicate reads were uninformative and shouldn't be considered as evidence for variants. We used Picard to mark these duplicates for follow-up analysis.

## Detecting and callings of somatic mutations

BWA and Samblaster were used to genome alignment, and muTect Software[88] was used for targeting Somatic SNV sites, and Strelka[89] was used to test Somatic INDEL information. Statistics used in the manuscript includes moderated $t$-statistics, and Fisher's exact test. The manuscript statistics were used to moderated $t$-test and Fisher's exact test.

## Gain of neo-mutations

The mutation frequency was estimated by the ratio of the number of mutated samples vs. the number of total samples[90]. In our study, the neo-mutations represented the mutations appearing at a certain stage, but was not identified in earlier stages. For example, *DST* mutation was detected in the HGIN stage, but was not in the LGIN stage, indicating *DST* mutation was the neo-mutation of HGIN stage.

## Analysis of mutation spectra and mutational signature in DC cohort

The features of single nucleotide variants (SNVs) in diverse subtypes of DC were disclosed. In our study, we used NMF algorithm to estimate the minimal components explaining maximum variance among DC

samples. Each component was compared to mutation patterns of 30 validated cancer signatures reported from the COSMIC database (https://cancer.sanger.ac.uk/cosmic/signatures_v2.tt) to identify cancer-related mutational signatures. Cosine similarity analysis[91,92] was used to measure the similarity between component and signatures, which ranged from 0 to 1, indicating maximal dissimilarity to maximal similarity.

## Analysis of somatic SNVs signatures (tobacco signature and DNA repair panel)

To identify SBS and portray the contribution across the whole genome based on WES data, we applied the analysis procedure as an R/CRAN package sigminer (Version 2.0.1) (https://cran.r-project.org/web/packages/sigminer/), to extract, analyze, and visualize mutational signatures for genomic variations, providing insights into cancer study. The most common criteria to estimate signature number is the cophenetic correlation coefficient (Supplementary Fig. 3a). Sigminer package (Version 2.0.1) can provide both relative and absolute exposures of cancer signatures. In our study, we performed NMF analysis with 200 runs to extract signatures, and then got the contribution proportion of samples from the relative exposure of each signature. Finally, the samples with the highest SBS29 (Tobacco signature) contribution proportion among all signatures were grouped into SBS29 signature enriched group, or DNA repair enriched group (DNA repair panel).

## Impacts of the detected mutations on the protein and phosphoprotein levels

**Analysis of SCNAs and the impacts on protein expressions.** SCNAs analysis used WES-derived BAM files that were processed in the somatic mutation detection pipeline. To investigate the *cis-/trans-*effects of SCNAs at the chr8q, we focused on the genes were detected both at the SCNA and protein levels, and then the spearman correlation coefficients were calculated between SCNAs and their corresponding proteins (FDR < 0.05).

**Impacts of DST mutation.** To investigate the impacts of the mutation of *DST* on the protein and phosphoprotein levels, fold change (FC, Mut vs. WT) of the proteins/phosphoproteins abundance of the two groups was compared, and the (adjust)*p*-value (Wilcoxon signed-rank test) less than 0.05 was considered. Then, we used gene sets of molecular pathways from KEGG[93]/Reactome[94] databases to compute the pathway.

## Defining Cancer-associated Genes (CAGs)

CAGs were compiled from genes defined by Bailey et al. [45] and CAGs were listed in Mertins et al.[46] and adapted from Vogelstein et al. [95]. The list of genes is provided in Supplementary Data 4b.

## TCEP-based protein extraction and trypsin digestion

All samples in our cohort were dissected with microdissection and then collected in 1.5 mL EP tubes, and then stored in −80 °C refrigerator.

The size of a single FFPE punch of early DC is no more than $0.1 \times 0.1 \times 3$ mm (10 μm thickness), and the mass was under 100 μg including wax. PCT would be an available method that effectively minimize sample loss[69], while the lowest operable weight (about 300 μg, $0.5 \times 0.5 \times 3$ mm, 5 μm thickness) is still higher than that in this study (under 100 μg). TCEP, a priorly reducing agent, leads to high internal standard recovery (>95%) combined with iodoacetamid (IAA)[70], and are applied in other published studies, including cholangiocarcinoma[96], and urothelial carcinoma of the bladder[97]. In addition, we previously have achieved deep proteome and phosphoproteome coverage of the definite volume samples (nearly less than 50 μg) in early-stage gastrointestinal cancer[71] on the basis of the

method of TCEP extraction protein. Therefore, the method of TCEP extraction protein was applied in this study.

Detailly, 50 μL TCEP buffer (2% deoxycholic acid sodium salt, 40 mM 2-chloroacetamide, 100 mM tris-phosphine hydrochloride, 10 mM (2-carboxyl)-phosphine hydrochloride, 1 mM phenylmethylsulfonyl fluoride mixed with MS water, PH 8.5) was added into 1.5 mL EP tubes with prepared samples, and then heated in a 99 °C metal bath for 30 min. After cooling to room temperature, 3 μg trypsin (REF: V528A, PROMEGA) was added into each tube and digested for 18 hours in a 37 °C incubator. Then, 13 μL 10% formic acid was added into each tube and made vortex for 3 min, and then centrifuged for 5 min (12,000*g*). Later, a new 1.5 mL tube with 350 μL buffer (0.1% formic acid in 50% acetonitrile) is needed for collected the supernatant for extraction (vortex for 3 min, and then 12,000*g* centrifuged for 5 min). Then, the supernatant was transferred into a new tube for drying in 60 °C vacuum drier. After drying, 100 μL 0.1% formic acid was needed for dissolving the peptides and vortex for 3 min, and then centrifuged for 3 min (12,000*g*). The supernatant was picked into new tube and then desalinated. Before desalination, the activation of pillars with 2 slides of 3 M C18 disk is required, and the lipid is as follows: 90 μL 100% acetonitrile twice, 90 μL 50 and 80% acetonitriler once in turn, and then 90 μL 50% acetonitrile once. After pillar balance with 90 μL 0.1% formic acid twice, the supernatant of the tubes was loading into the pillar twice, and decontamination with 90 μL 0.1% formic acid twice. Lastly, 90 μL elution buffer (0.1% formic acid in 50% acetonitrile) was added into the pillar for elution twice and only the effluent was collected for MS. And then the collect of lipid was put in 60 °C vacuum drier for drying.

## Proteome analysis in LC−MS/MS analysis

For the proteomic profiling of samples, peptides were analyzed on a Q-Exactive HFX Hybrid Quadrupole-Orbitrap Mass Spectrometer (Thermo Fisher Scientific, Rockford, IL, USA) coupled with a high-performance liquid chromatography system (EASY nLC 1200, Thermo Fisher). Dried peptide samples re-dissolved in Solvent A (0.1% FA in water) were loaded to a 2-cm self-packed trap column (100-μm inner diameter, 3 μm ReproSil-Pur C18-AQ beads, Dr. Maisch GmbH) using Solvent A and separated on a 150-μm-inner-diameter column with a length of 15 cm (1.9 μm ReproSil-Pur C18-AQ beads, Dr. Maisch GmbH) over a 150 min gradient (Solvent A: 0.1% FA in water; Solvent B: 0.1% FA in 80% ACN) at a constant flow rate of 600 nL/min. The eluted peptides were ionized under 2.0 kV and introduced into mass spectrometer). MS was performed under a data-dependent acquisition mode. For the MS1 Spectra full scan, ions with m/z ranging from 300 to 1,400 were acquired by Orbitrap mass analyzer at a high resolution of 120,000. The automatic gain control (AGC) target value was set as 3E6. The maximal ion injection time was 80 ms. MS2 Spectra acquisition was performed in the ion trap mode at a rapid speed. Precursor ions were selected and fragmented with higher energy collision dissociation (HCD) with a normalized collision energy of 27%. Fragment ions were analyzed by the ion trap mass analyzer with the AGC target at 5E4. The maximal ion injection time of MS2 was 20 ms. Peptides that triggered MS/MS scans were dynamically excluded from further MS/MS scans for 12 s.

In our study, 438 samples were randomly assigned number and then performed proteomic analysis. The HEK293T cell (National Infrastructure Cell Line Resource) lysates were measured as the standard every 15 samples injection to assess the stability of the performance of mass spectrometry.

## Phosphopeptide enrichment and analysis

All qualified profiling data were processed at firmiana platform against the human RefSeq protein database (updated on 04-07-2013) in the National Center for Biotechnology Information (NCBI). Owing to the

definite volume of the samples, only 111 samples (from 49 EDC patients) were found to be adequate.

The phosphoproteome samples were prepared by Fe-NTA Phosphopeptide Enrichment Kit (Thermo, Catalog: A32992) according to the manufacturer's instruction. Briefly, 2 mg peptides were resuspended in 200 µL binding/wash buffer and loaded to the equilibrated spin column. The resin was mixed with the sample by gently tapping. The mixture was incubated for 30 min and centrifuged at $1000g$ for 30 s to discard the flowthrough. The column was then washed by 200 µL of binding/wash buffer and centrifuged at $1000g$ for 30 s for 3 times and washed by 200 µL of LC–MS grade water for one time. The phosphopeptide was eluted by adding 100 µL of elution buffer and centrifuged at $1000g$ for 30 s 2 times. Phosphopeptides were dried down for LC–MS/MS analysis.

### Quantification of global proteomic data and phosphoproteomic data

In our study, all MS raw files were processed at firmiana platform[98] (a one-stop proteomic cloud platform: http://www.firmiana.org). Briefly, all MS raw files were searched against the NCBI human RefSeq protein database (updated on 04-07-2013, 32,015 entries) in Mascot search engine (version 2.3, Matrix Science Inc). Trypsin was used as the proteolytic enzyme allowing up to two missed cleavages. Carbamidomethyl (C) was considered as a fixed modification. For the proteome profiling data, variable modifications were oxidation (M) and acetylation (Protein N-term). For the phosphoproteome data, variable modifications were oxidation (M), acetylation (Protein N-term) and phospho (S/T/Y). All the identification peptides were quantified at firmiana platform with peaks area derived from their MS1 intensity. The mass tolerances were 20 ppm for precursor and 50 mmu for the product collected by Q-Exactive HFX, which has been applied in previously published studies[97]. Precursor ion score charges were limited to +2, +3, and +4. The FDRs of the peptide-spectrum matches (PSMs) and proteins were set at a maximum of 1%. The same cutoff strategies of FDR have been widely used in recently published researches[99,100]. Label-free protein quantifications were calculated in our cohort, that so called iBAQ algorithm[26,27], which divided the protein abundance (derived from identification peptides' intensities) by the number of theoretically observable peptides. Then the FOT, defined as a protein's iBAQ divided by the total iBAQ of all identified proteins within one sample, was used to represent the normalized abundance of a particular protein across samples.

A total of 438 samples from 156 DC patients for proteomic profiling, and all the proteome data were processed as follows: EDC1 (14,426 GPs): all 14,426 gene products (GPs) identified in 438 samples (156 EDC cases) on the basis of the match between runs algorithm (MBRs)[101]; EDC2 (11,904 GPs): all the proteins were required to have at least 2 unique strict peptide, and we excluded keratins proteins of which the maximum FOT in all 438 samples were less than 1.0E−5 in FOT; EDC3 (7400 GPs): GPs were identified in more than 20% samples of each stage, which was also applied in other published studies[102,103].

At the phosphoprotein level, a total of 111 samples from 49 DC patients were conducted, and 38,696 phosphosites, corresponding to 9716 phosphoproteins, were identified. The phosphosites ($n = 12,851$) identified in more than 20% samples of every stage were used in the analysis of our cohort. In addition, the phosphoprotein/phosphopeptide abundance was adjusted by the total protein counterpart abundance, which has been applied in previous published studies[104].

### Data imputation

For the missing value in our study, we applied MBRs algorithm[101,105] in this study, which has been proved to be an effective technique to fill the missing values, which was widely used in other proteomic studies[106]. Detailly, we built a dynamic regression function based on common identification peptides in samples. According to correlation value $R^2$, the function chooses linear or quadratic function for regression to calculate retention time (RT) of corresponding hidden peptides, and check the existence of the extracted ion chromatogram (XIC) based on the m/z and calculated RT. The function evaluated the peak area values of those existed XICs. These peak area values are considered as parts of corresponding proteins. This strategy has been applied in other published proteomic studies[102,107].

### Principal component analysis (PCA)

A total of 7400 proteins (FOT) were used for PCA in R (version 3.5.1). We performed PCA to investigated the three principal components and to visualize separation of all the stages of the D and G subtypes.

### The hierarchical clustering for 7 subtypes of DC

To compare the correlation and distance of 2 major and 5 rare subtypes of duodenal cancer, hierarchical clustering was performed by the fastcluster package (version 1.1.25) in R (version 3.5.1). The expressed proteins of 2 major and 5 rare subtypes of duodenal cancer were used to reduce the individuality of different patients. Average was picked for method in the fast cluster. The distance between two clusters, A and B, is the average between the points in the two clusters:

$$d(A,B) = 1/|A||B|\Sigma_-(a \in A, b \in B)\, d(a,b) \qquad (1)$$

### Differential proteomics analysis and pathways enrichment

In comparison of the features of the D and G subtypes, the DEPs of the DAC stage in the P1 and the A-G stage in the P2 were identified. Meantime, the DEPs of the normal tissues in the D and G subtypes were integrated to investigate the origins of the D and G subtypes. Then, we used gene sets of molecular pathways from KEGG[93] and Reactome[94] databases to compute single sample gene set enrichment scores for each sample. Statistical significance was considered when FDR (adjust) was less than 0.05.

When comparing the DEPs of different stages of the D or G subtypes, we focused on protein abundance (average) at each stage, which were enriched by KEGG/GO database[93]. We then annotated the signaling pathways (FDR < 0.05) and manually checked the pathway associated proteins, which were then estimated whether significantly associated with the stages of the D and G subtypes (Kruskal–Wallis test, adjust. $p < 0.05$).

In differential analysis of proteins in DC progression (gradually decreased or increased) at the protein level, the highly expressed proteins of each stage/panel were screened, in which foldchange (FC) and adjust. $p$-value (Kruskal–Wallis test) were considered. Statistical analysis was performed in R (version 3.5.1).

To assess the impacts of the four immune clusters, we applied gene set variation analysis (GSVA)[108] for pathways enrichment analysis, in which molecular Signatures Database (MSigDB) of KEGG gene sets were used. In addition, to assess the correlation between mutation burden/proteins (e.g., LYN and AARS1) and the enrichment pathways, the enrichment value was used. For example, as for the correlation between AARS1 and the represented pathways in Fig. 5, the $R$ values and $P$ values were based on the intensity of AARS1 and the enrichment values of the represented pathways among the samples in the Fudan cohort.

### Correlation analysis

In the analysis of the correlation among proteins, we excluded the outliers and missing values, and two-sided Spearman's correlation test was applied to explore the correlation among the HEK293T cells' samples, and the stage's samples of DC. In addition, two-sided Pearson's correlation test was applied to investigate the correlation (1) between two proteins/phosphoproteins, (2) between proteins and pathways, and (3) between proteins and TMB in the study.

## Analysis of the specific proteins and features of the D and G subtypes

To explore the carcinogenesis lineages of the D and G subtypes, the DEPs (Kruskal-Wallis test, adjust. $p < 0.05$) in P1 and P2 were used into analysis, which were then applied to the pathways enrichment, and consequentially resulted into the characterized features of the D and G subtypes carcinogenesis.

## Consensus clustering analysis of immune infiltration in DC

Unsupervised clustering of the patient samples at the different molecule levels were performed with the ConsensusClusterPlus R package (ConsensusclusterPlus, v1.46.0)[109]. Samples were clustered using Euclidean distance as the distance measure. A total of 7400 proteins were used for $k$-means clustering with up to 10 clusters. The consensus matrix of $k = 4$ showed clear separation among clusters and was appropriate in consideration of the percentage of the samples with stages pathologically. Taken together, proteome clusters were defined using k-means consensus clustering with $k = 4$.

## Analysis of the immune signature of the D and G subtypes

Proteomic data is often applied to investigate the immune filtration and the response to immunotherapy. For example, Keren et.al elucidated a structure tumor-immune microenvironment in triple negative breast cancer[110]. The landscapes of response to immunotherapy in melanomas was based on the proteomic data[106]. As well as RNA data, proteomic data, used for xCell enrichment analysis, is applied to estimate immune characterization[97,111,112]. Thus, in our cohort, we performed the xCell enrichment analysis based on the proteomic data to investigate the immune filtration of DC. The abundance of 64 different cell signatures in 438 samples (156 DC cases) were computed via xCell (https://xcell.ucsf.edu)[57] and then, consensus clustering was performed to identify subtypes of the samples. As a result, 4 immune clusters were profiled, including *Hot tumor* cluster, *Cold tumor* cluster, *Epithelial* cluster, and the *B cells* cluster. Each cell signature and immune/stromal/environmental score were normalized to Z-score. In addition, we performed the differential analysis of the identified CD/MHC molecules based on global proteomic data.

To further evaluate the features of the D and G subtypes, we compared the expression of immune cell markers, which were CellMarker web (http://biocc.hrbmu.edu.cn/CellMarker/). Log$_2$ FC (P1 vs. P2 ratio) of markers expression from the DNA repair panel was taken into consideration of the features and specific carcinogenesis lineages of diverse subtypes of duodenal cancer (D and G subtypes).

## Kinase activity prediction and phosphopeptide analysis

The ratios of identified phosphorylation sites of all samples were used to estimate the kinase activities by Kinase-Substrate Enrichment Analysis (KSEA) algorithm[113]. The information of kinase-substrate relationships was obtained from publicly available databases, including PhosphoSite[114], Phospho.ELM[115] and PhosphoPOINT[116]. The information of substrate motifs was obtained either from the literature[117] or from an analysis of the KSEA dataset with Motif (sP)[118]. The kinase-substrate-motif network analysis was referenced from PhosphoSitePlus (PSP, https://www.phosphosite.org/homeAction)[119] and NetworKIN 3.0[120]. Statistical analysis was performed in R (version 3.5.1) with Kruskal−Wallis test.

## IHC analysis

AARS1 in the tissue by IHC staining, 3-µm-thick sections from FFPE tissue block were de-waxed with xylene and rehydrated through a graded series of ethanol, prepared by Zhongshan Hospital, Fudan University.

Total AARS1 was performed on representative samples from normal to progressive DC (Fig. 6a). Polyclonal antibody for AARS1 (dilution 1:100, Proteintech, catalog No:17394-1-AP) was used for IHC.

Antigen was retrieved by autoclaving for 10 min at 121 °C in Novocastra Epitope retrieval Solution pH 6 (Leica Biosystems). The UltraVision Quanto Detection System HRP DAB (Thermo Fisher Scientific) was used according to the manufacturer's instruction. Finally, the sections were counterstained with hematoxylin and then mounted.

## Cell lines and cell culture

Human HEK293T (ATCC, Catalog: CRL-11268; RRID: CVCL_QW54) cells were cultured in high-glucose Dulbecco's modified Eagle's medium (DMEM; HyClone) supplemented with 10% fetal bovine serum (FBS; Invitrogen), 100 units/mL penicillin (Invitrogen), and 100 µg/mL streptomycin (Invitrogen). Human Hutu80 (ATCC, Catalog: CRL-7928; RRID: CVCL_1301) and WDC-1 (Cobioer, Catalog: CBP61181, RRID: CVCL_R803) cells were cultured in RPMI 1640 medium (Invitrogen) supplemented with 10% fetal bovine serum (Invitrogen, Carlsbad, CA, USA), 100 units/mL penicillin (Invitrogen), and 100 µg/mL streptomycin (Invitrogen). The cells were incubated in 5% $CO_2$ at 37 °C. Cell transfection was performed using Lipofectamine 3000 (Invitrogen). All the cell lines were authenticated with short tandem repeat (STR) profiling method.

## Plasmid constructs and transfection

Whole length human AARS1, PARP1, SIRT1, SIRT2, SIRT6, and SIRT7 were amplified from HEK293T cDNA and cloned into the Xho I and EcoR I restriction sites of the pcDNA3.1-Flag/HA vector using CloneExpress MultiS One Step Cloning Kit (Vazyme Biotech Co., Ltd., Nanjing, China, Catalog: C113-02). The PARP1 and AARS1 mutants were generated by site-directed mutagenesis using the Mut Express MultiS Fast Mutagenesis kit (Vazyme Biotech Co., Ltd., Nanjing, China, Catalog: C215-01) according to the manufacturer's instructions. Each plasmid was transfected using Lipofectamine 3000 (Invitrogen) according to the manufacturer's instructions.

The primers used were as follows:

AARS1: forward, 5′-AACGGGCCCTCTAGACTCGAGATGGACTCTACTCTAACA-3′

reverse, 5′-TAGTCCAGTGTGGTGGAATTCGTTCTTTACATCCCCGAGG-3′

PARP1: forward, 5′-AACGGGCCCTCTAGACTCGAGATGGCGGAGTCTTCGGATA-3′

reverse, 5′-TAGTCCAGTGTGGTGGAATTCCCACAGGGAGGTCTTAAAA-3′

SIRT1: forward, 5′-AACGGGCCCTCTAGACTCGAGATGGCGGACGAGGCGGCC-3′

reverse, 5′-TAGTCCAGTGTGGTGGAATTCTGATTTGTTTGATGGATAG-3′

SIRT2: forward, 5′-AACGGGCCCTCTAGACTCGAGATGGACTTCCTGCGGAACT-3′

reverse, 5′-TAGTCCAGTGTGGTGGAATTCCTGGGGTTTCTCCCTCTCT-3′

SIRT6: forward, 5′-AACGGGCCCTCTAGACTCGAGATGTCGGTGAATTACGCGG-3′

reverse, 5′-TAGTCCAGTGTGGTGGAATTCGCTGGGGACCGCCTTGGCC-3′

SIRT7: forward, 5′-AACGGGCCCTCTAGACTCGAGATGGCAGCCGGGGGTCTG-3′

reverse, 5′-TAGTCCAGTGTGGTGGAATTCCGTCACTTTCTTCCTTTTTGT-3′

AARS1 R751G: forward, 5′-GGGTATCCGGGGGATTGTGGCTGTCACAGGT-3′

reverse, 5′-GCCACAATCCCCCGGATACCCTTGGCAATGG-3′

PARP1 K23A: forward, 5′-TCTTGCAAGGCATGCAGCGAGAGCATCCCCA-3′

reverse, 5′-CTCGCTGCATGCCTTGCAAGAGGCGCGCCCG-3′

PARP1 K23R: forward, 5′-TCTTGCAAGAGATGCAGCGAGAGCATCCCCA-3′

reverse, 5′-CTCGCTGCATCTCTTGCAAGAGGCGCGCCCG-3′
PARP1 K414/418A: forward, 5′-GAGGCACTCGGGGGGGCGTTGA CGGGGACGG-3′

reverse, 5′-CAACGCCCCCCCGAGTGCCTCAATCATGGCC-3′
PARP1 K414/418R: forward, 5′-GAGAGACTCGGGGGGAGGT TGACGGGGACGG-3′

reverse, 5′-CAACCTCCCCCCGAGTCTCTCAATCATGGCC-3′
PARP1 K621A: forward, 5′-TATGAAGAAGCAACCGGGAACGCTT GGCACT-3′

reverse, 5′-GTTCCCGGTTGCTTCTTCATATAATTTCATG-3′
PARP1 K621R: forward, 5′-TATGAAGAAAGAACCGGGAACGCT TGGCAC-3′

reverse, 5′-GTTCCCGGTTCTTTCTTCATATAATTTCATG-3′

*RNA interference.* Synthetic oligos were used for siRNA-mediated silencing of AARS1, and scramble siRNA was used as a control. Cells were transfected with siRNAs using Lipofectamine 3000 according to the manufacturer's protocol. Knockdown efficiency was verified by western blotting. SiRNA sequences was as follows: AARS1: 5′- CAG CAA GTG AAA TCC GGC AGC GATT-3′.

### Stable gene overexpression

For stable AARS1 overexpression, HEK293T cells (ATCC, Catalog: CRL-11268, RRID: CVCL_QW54) were co-transfected with pCMV-VSV-G, pCMV-Gag-Pol, and pBABE-AARS1. Transfected cells were cultured in Dulbecco's Modified Eagle's Medium containing 10% FBS for 6 h. Twenty-four hours after transfection, culture medium supernatant was collected and used for retrovirus preparation to infect Hutu80 cells at 10% confluency in 90-mm-diameter dishes. Cells were re-infected 24 h after the first infection and selected using 5 μg/mL puromycin (Amresco, Solon, OH, USA). AARS1 was subcloned into the BamH I and EcoR I restriction sites of the pBABE vector using the ClonExpress MultiS One-Step Cloning Kit (Vazyme Biotech Co., Ltd, Nanjing, China, Catalog: C113-02). The sequences of primers used were as follows: pBABE-AARS1: forward, 5′-ctc tag gcg ccg gcc gga tcc ATG GAC TCT ACT CTA ACA-3′, and reverse, 5′-ggt ctt ctc gtc cat gaa ttc GTT CTT TAC ATC CCC GAG-3′.

### Western blot analyses

Cultured cells or cells extracted from duodenal cancer tissue and patient-matched normal tissues were homogenized with 0.5% NP-40 buffer containing 50 mM Tris-HCl (pH 7.5), 150 mM NaCl, 0.5% Nonidet P-40, and a mixture of protease inhibitors (Sigma-Aldrich). After centrifuged at 12,000 rpm and 4 °C for 15 min, the supernatants of the lysates were collected for western blotting according to standard procedures. Anti-AARS1 (dilution 1:100, Proteintech, catalog No:17394-1-AP), Anti-Poly (ADP-Ribose) polymer (dilution 1:100, Abcam, catalog No:14459), Anti-Histone H2A.X (dilution 1:100, Cell Signaling Technology, catalog No: 7631), Anti-Phospho-Histone H2A.X(Ser139) (dilution 1:200, Cell Signaling Technology, catalog No: 9718), Anti-SARS1 (dilution 1:500, Abclonal, catalog No: A13350), Anti−TARS1 (dilution 1:500, Abclonal, catalog No: A6993), Anti-Flag (dilution 1:5000, Proteintech, catalog No: 20543-1-AP), Anti-HA (dilution 1:5000, Proteintech, catalog No: 51064-2-AP), Anti-PARP1 (dilution 1:500, Proteintech, catalog No: 13371-1-AP), Anti-TARS1 (dilution 1:500, Proteintech, catalog No: 14773-1-AP), Anti-GARS1 (dilution 1:500, Proteintech, catalog No.: 15831-1-AP), Anti-Actin (dilution 1:1000, Proteintech, catalog No.: 66009-1-Ig), Anti-GAPDH (dilution 1:5000, Proteintech, catalog No.: 60004-1-Ig). Detection was performed by measuring chemiluminescence on a Typhoon FLA 9500 (GE Healthcare, Little Chalfont, UK).

### Sample preparation for LC−MS/MS analysis

For detecting AARS1-interacting proteins or lysine-alanylation sites on PARP1, Hutu80 cells were transfected with AARS1-Flag or PARP1-Flag vector. Cells were harvested and lysed in 0.5% NP-40 buffer, and then supernatants were immunoprecipitated with anti-FLAG M2 magnetic

beads (dilution 1:5000, Proteintech, catalog No: 20543-1-AP) for 3 h in 4 °C. The precipitates were washed twice with 0.1% NP-40 buffer, twice with ddH₂O and thrice with 50 mM NH₄HCO₃, after which on-bead tryptic digestion was performed at 37 °C overnight. The peptides in the supernatants were collected through centrifugation and dried in a speed vacuum (Eppendorf). Obtained peptides were stored at −80 °C until LC−MS/MS analysis.

### K-Ala site identification

Raw MS files were analyzed using MaxQuant version 1.4.1.2. MS/MS spectra were searched using the Andromeda search engine against the SwissProt-human database (Release 2014-04-10) containing forward and reverse sequences. In the main Andromeda search precursor, mass and fragment mass had an initial mass tolerance of 5 ppm and 0.05 Da. The search included alanylation of lysine and oxidation of methionine. Minimal peptide length was set to seven amino acids, and a maximum of four miscleavages was allowed. The FDR was set to 0.01 for peptide and protein identifications.

### PARP1 activity assay in vitro

PARP1 self-PARylaton assay in vitro was modified from a reported method[121]. In brief, experiments were performed in a 40 μl reaction mix contains 50 mM HEPES (pH = 8.0), 4 mM DTT, 20 mM MgCl₂, 25 μl NAD⁺, 200 nM DNA and 2 μg PARP1 protein. The reaction was allowed for 30 min at 30 °C. The reaction mix was then collected for western blotting to detect poly (ADP-ribose) polymer levels. For measuring PARP1 activity in cell extracts and in vitro reactions, a PARP1 colorimetric assay kit (BPS Bioscience, USA, Cat# 80580) was employed according to manufacturer's instruction. To evaluate the reduction of PARP1 activity by interacting with AARS1, purified AARS1 was added into the reaction mix and incubated with PARP1 for 30 min at 4 °C at the sample preparation step of the kit.

### DNA-binding capacity assay

To test the DNA-binding capacity of PARP1 in vitro, Hutu80 cells were transfected with wildtype and mutant PARP1 plasmid. After cells were harvested, a DNA−Protein Binding Assay kit (Abcam, Catalog: 117139) was used to detect the DNA-binding capacity of PARP1 according to the manufacturer's instructions.

### Apoptosis and cell proliferation assay through flow cytometry

Sample preparation: A Beckman Coulter flow cytometer (Beckman Coulter, Brea, CA, USA) was used to detect apoptotic cells. Data were collected on a Beckman Coulter flow cytometer (Beckman Coulter, Brea, CA, USA). An Annexin V-FITC Apoptosis Detection Kit (BD Biosciences) was used to detect apoptotic cells according to manufacturers' instruction. In addition, apoptosis of cells was gated using Annexin V and PI following doublet exclusion using FSCHxW and SSC-HxW. Cell proliferation was assessed by the Cell Counting Kit-8 (Dojindo Laboratories, Kumamoto, Japan, Catalog: CK04). In brief, Hutu80 cells were seeded in a 96-well plate with $4 \times 10^3$ cells per well and allowed to adhere. Cell Counting Kit-8 solution (10 μl) was added to each well, and the cells were cultured in 5% CO₂ at 37 °C for 2 h. Purity of isolated samples was obtained by antibody stain and FACS. Sample purity was greater than 95%. Cell proliferation was determined by measuring the absorbance at 450 nm. Instrument: Beckman Coulter flow cytometer (Beckman Coulter, Brea, CA, USA). Software: Flowjo version 10.7.1 (Becton Dickinson Life Science).

### Comet assay

A Comet Assay Kit (Trevigen, Catalog: 4250-050-K) was used to detect single- and double-stranded DNA breaks in cultured cells and tissues. Slides were examined (under 425−500-nm excitation) using a Leica DMI 4000B epifluorescence microscope. Comet slides were used for each condition. In the assay, the fluorescence in normal cells is mostly

confined to the nucleus because intact DNA cannot migrate. In DNA-damaged cells, DNA is denatured with an alkaline or neutral solution for detecting single-/double- stranded breaks, respectively; negatively charged DNA fragments are released from the nucleus and migrate toward the anode.

### Mouse xenograft studies in vivo

Four- to six-week-old male BALB/c nude mice were obtained (Shanghai SLAC Laboratory Animal Co., Ltd., Shanghai, China) for in vivo xeno-grafts ($n = 6$ mice per group). Mice were housed in polycarbonate cages, and provided free access to food and water with a 12-h light: dark cycle. Control cells and cells overexpressing AARS1 from Hutu80 cell lines were subcutaneously heterotransplanted into the left and right flanks of each mouse. For HU and alaninol treatment, 100 mg/kg HU or 100 mg/kg alaninol was intraperitoneally injected into the abdominal cavities of the animals twice a week. At the end of the experiment, following euthanasia, tumors were excised, weighed and imaged. All procedures were performed with approval from the Animal Care Committee at Fudan University. The maximal tumor burden permitted by the committee is 2000 mm$^3$.

### Alanylation and de-alanylation in vitro

In vitro alanylation and de-alanylation reactions were carried out as described before[68]. In vitro alanylation was carried out in a 30-µl reaction mix containing 50 mM HEPES (pH 7.5), 25 mM KCl, 2 mM MgCl$_2$, 5 mM alanine, 4 mM ATP, 10 nM AARS1, and 0.05 mg/ml synthetic substrate peptide. The final pH of each reaction mix was adjusted to 7.5 before adding AARS1. The reaction was allowed for 3 h at 37 °C. The peptide was desalted by passing through a C18 ZipTip (Millipore) and was subjected to analyzation by a MALDI-TOF/TOF mass spectrometer (SCIEX-5800).

In vitro de-alanylation reactions were carried out in a 30-µl reaction mix containing 50 mM HEPES (pH 7.5), 6 mM MgCl$_2$, 1 mM DTT, 1 mM NAD+, 0.05 mg/ml synthetic peptide, 1 mg/ml SIRT6 and 1 mM PMSF. The reaction was allowed for 4 h at 37 °C. The peptide was desalted by passing through a C18 ZipTip (Millipore) and was subjected to analyzation by a MALDI-TOF/TOF mass spectrometer (SCIEX-5800).

### Quantification and statistical analysis

Statistical details of experiments and analyses can be observed in the (Supplementary data)/figure legends in this study. All the analyses were performed in R (version 3.5.1) and GraphPad Prism (version 7.0). Standard statistical tests were used to analyze the clinical data, including but not limited to Wilcoxon signed-rank test, Fisher's exact test, Kruskal–Wallis test. The statistical significance of differences between two groups was calculated with the Wilcoxon rank-sum test; for more than two group comparisons, Kruskal–Wallis test was used. Fisher's exact test was used for categorical variables and Wilcoxon rank-sum test was used for continuous variables, when testing association of different groups with clinical variables. As for the correlation analysis between two proteins/phosphoproteins, Pearson's correlation (two-sided) of correlation coefficients were used. All statistical tests were two-sided except special explanations. For validation experiments, each was repeated at least three times independently, representative photos were shown. To account for multiple-testing, the $p$ values were adjusted using the Benjamini-Hochberg FDR correction. Data in the boxplot were presented median (central line), upper and lower quartiles (box limits), 1.5× interquartile range (whiskers). Statistical significance was considered when $p$-value < 0.05. For validation experiments, each was repeated at least three times independently, representative photos were shown.

### Reporting summary

Further information on research design is available in the Nature Portfolio Reporting Summary linked to this article.

## Data availability

The proteome and phosphoproteome raw datasets and processed results files generalized in this study have been deposited to the ProteomeXchange Consortium (dataset identifier: PXD038867) via the iProX partner repository (https://www.iprox.cn/)[122] under Project ID IPX0002184000. The VCF files of the WES data files were deposited to the European Genome-Phenome Archive (EGA) associated with the study EGAS00001006357 under project ID EGAD00001008987. The raw WES data are available in the Genome Sequence Archive (GSA)[123] under restricted access HRA004048. The user can register and login to the GSA database website (https://ngdc.cncb.ac.cn/gsa-human/) and follow the guidance of "Request Data" to request the data step by step (https://ngdc.cncb.ac.cn/gsa-human/document/GSA-Human_Request_Guide_for_Users_us.pdf). The approximate response time for accession requests is about 2 weeks. The access authority can be obtained for Research Use Only. The user can also contact the corresponding author directly. Once access has been granted, the data will be available to download for 3 months. The gene expression profiles of DC cell lines in public dataset Expression 21Q2 in this study are available in the Depmap database (https://depmap.org/portal/download/?releasename=DepMap+Public+21Q2&filename=CCLE_expression.csv). The remaining data are available within the Article, Supplementary Information, or Source Data file. Source data are provided in this paper.

## Code availability

No special code was used in this study, and code for all figures in the study are available for research purposes from the corresponding authors on request.

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

## Acknowledgements

This work is supported by National Key R&D Program of China (2022YFA1303200 [C.D.], 2022YFA1303201 [C.D.], 2020YFE0201600 [C.D.], 2018YFE0201600 [C.D.], 2018YFE0201603 [C.D.], 2018YFA0507500 [C.D.], 2018YFA0507501 [C.D.], 2017YFA0505100 [C.D.], 2017YFA0505102 [C.D.], 2017YFA0505101 [C.D.], 2017YFC0908404 [C.D.], 2016YFA0502500 [C.D.]), sponsored by Program of Shanghai Academic/Technology Research Leader (22XD1420100 [C.D.]), Shuguang Program of Shanghai Education Development Foundation and Shanghai Municipal Education Commission (19SG02 [C.D.]), National Natural Science Foundation of China (31972933 [C.D.], 31770886 [C.D.], 31700682 [C.D.], 81702372 [Y.H.]), the Major Project of Special Development Funds of Zhangjiang National Independent innovation Demonstration Zone (ZJ2019-ZD-004 [C.D.]), Shanghai Natural Science Foundation of China (18ZR1406800 [Y.H.]), Shanghai Municipal Science and Technology Major Project (2017SHZDZX01 [C.D.]), and the Fudan original research personalized support project.

## Author contributions

C.D., Y.H., J.Y.Z., L.L., and D.J. conceived the work and designed the experiments; L.L., D.J., R.Z., H.L., C.G., and C.X collected the tissue samples; L.L., D.J., H.L., C.G., Q.Z., and C.X. performed the experiments and acquired the MS data; J.H. P.Z., J.Y.Z., Z.Y.Q., J.F., Y.L., L.B., S.T, S.B.T., B.L., H.W., W.C., X.Z., Y.Z., Z.Y., and L.C. provided expertize and technical support; L.L., D.J., H.L., C.G., Q.Z., and R.Z. analyzed the data; C.D. and L.L. wrote the original paper. All authors contributed to data interpretation, paper editing, and revision.

## Competing interests

The authors declare no competing interests.
