## [Peer review file · Nature Communications]

REVIEWER COMMENTS

Reviewer #1 (Remarks to the Author): Expert in immune tumour microenvironment in duodenal adenocarcinoma

The manuscript contains a wealth of data related to improved classification of duodenal cancers, rare gastrointestinal tumors with limited therapeutic options. Authors analyzed 140 tumors, from which they obtained 406 macrodissected samples by manual scraping, and then performed several multi-omics analyses. The reader is overwhelmed by the impressive amount of data and substages, but I wonder whether the manual approach used qualifies for the complex analysis performed and for the even more complex final message.

- My major concern is with the sampling procedures. Starting from 140 patients, authors analyze 406 samples, obtained by manual macrodissection and scraping. Although expert pathologists annotated the tissue, this technique suffers of many limitations. In some samples, up to 6 different regions were scraped, I really wonder how far one can go with a macro-approach. I would like to receive some more details on this procedure and, most important, evidence should be provided that the scraped tissues correspond to the tissue annotated by the pathologists.

- In line with this, Authors state that the tumor purity of all samples was defined as the Score 5, which is the highest proportion of tumor cells. Then I don't understand why sample 1, which is substaged as normal, scores 5.

- Duodenal cancer is a rare disease, critical point that the Authors clearly discuss in the paper. The rarity of the disease prevents from obtaining adequate number of samples, which here is overcome putting together a cohort that spans from 2012 to 2019. This reviewer, however, is concerned with the reproducibility of such a complex analysis (including proteomic and WES) on FFPE sections that date back to 2012, or at least I am concerned on putting together samples that span such a long time. Quality controls to show the quality of material obtained from the older specimens is needed.

- Authors report a classification of the tissues into several stages and substages. Supp Table 1 reports age and gender of patients, but less clear is the substage information, which is explained in extended Figure 1. A reminding to this Figure would help understanding the Table. Also, in Extended Figure 1, there is no explanation what the dotted lines on the H&E pictures refer to. Finally, considering the high numbers of stages and subtypes, colore-codes should help, while in some figures (e.g. Fig 1A), color-code enhances the complexity.

- Which data was used to perform the xCell enrichment analysis, which is based on RNA sequencing data? Authors mention immune, stromal and microenvironmental signatures. In what do the signatures differ? Which color-code was assigned to each cluster in Fig 1A?

- Extended Figure 1b. Lamina propria should read lamina propria. Panel C is not clearly explained in the legend. What is NA?

Reviewer #2 (Remarks to the Author): Expert in MS-based proteomics and proteogenomics

Li et al. described an ambitious study in which duodenal cancer (DC) tissues were measured by multi-omics. They extensively analyzed the resultant data, acquired preliminary hints for the potential therapeutic targets, and performed some additional experiments on AARS1 and PARP1. Overall, the datasets are large and novel. However, significant issues have to be resolved.

Major concerns:

1. Proteogenomic landscape of DC. In this study, the authors relied on pathological subtypes (based on the evaluation of two or three pathologists) rather than the molecular landscape to perform all their analyses. This has to be improved. All previous proteogenomic studies used proteomic or genomic molecular signatures to understand the cancer heterogeneity and phenotypes. The underlying assumption is that molecular layers like proteome level could provide novel insights in stratifying patients. The authors should follow this best practice and also compare the result to pathological subtypes.

2. Their proteome and phosphoproteome data both contain lots of missing values, according to the Supplementary table 3. Fairer descriptions of really quantified protein and phosphopeptide numbers are urged. Although they mentioned the sample amount is needed, it seems other advanced techniques like the Pressure cycling technique (PCT) can already handle the tiny amount of clinical material very well. Please add relevant discussions if appropriate. The authors are advised to consider the possible bias of protein abundance in their data, result, and interpretation. Also, please provide additional analysis on how missing values are handled and affect the results reported.

3. The phosphoproteomic analysis. Throughout the study, the phosphoproteome result was mentioned with an impressive (total) identification number, but almost not used at all in their data and analysis. This might be due to the limited data quality (about 4500 phosphopeptides identified in each of the samples). Therefore, they should either consider discarding the phosphoproteomic data (if it is not good enough for Nature Communications) or considerably improve the data analysis of phosphoproteomics, and really use this level of their “multi-omics” to support all their analysis, e.g., pathway and signaling annotation, etc. Accordingly, they can also shorten the text of other parts (e.g., immune-cluster, 8q-related), which are purely based on data mapping, P values, and their own data interpretation.

4. Overall, there are lots of claims about the therapeutic potential of DC using their analysis results which are highly touted and should be significantly tuned down. Most of their results (except for the AARS1-PARP1 part) are just based on omics data interpretation with limited sample cohort and lack of supporting evidence. Just as two examples “.. suggested that the combination of capivasertib and everolimus/ sirolimus/ temsirolimus would be potential strategy in the D subtype” (Page 13), or “This observation may indicate the medicative actions of afatinib and lapatinib to anti-ERBB signaling, dinaciclib and pemetrexed, to anti-cell

proliferation, and olaparib to anti-PARPs, etc., in the clinic of the Hot tumor cluster” (Page 15). Their study only highlighted the need of further biological investigation which is far away from clinical applications. Also, conclusions in the Abstract are too strong and need to be tone down, e.g., “Proteogenomics revealed the chromosome 8q gain was the key event driving the transmit from the intraepithelial neoplasia phase to the infiltration tumor phase via RET signaling”

5. The author spent lots of effort analyzing the Tobacco signature in the D type. However, is tobacco usage a significant risk factor in DC, especially D? Please cite the reference. Can they use their clinical data to confirm this finding? Otherwise, what is the rationale of looking at Tobacco signature?

6. AARS1 and PAPR1. The author suggests that the interaction between AARS1 and PARP1 reduced the PARP1 activity through K-Ala. However, what is the abundance and activity status of PAPR1 in the DC proteome and phosphoproteome data? How much of PARP1 activity is reduced by interacting with AARS1? Is PARP1 the top 1 differential protein in the long list of 652 proteins identified as AARS1 interactors? Are there other proteins of these 652 proteins showing different expressions between in DC or between its subtypes? To summarize, although this additional experiment seems to be nicely done, it currently lacks coherency to the “proteogenomic” data.

Minor issues:

1. Page 3. Paragraph 2. “Hence DC was not included in TCGA”. How does the author know that DC is not on the CPTAC to-do list? we do not think the inclusion or exclusion of DC in TCGA is relevant for the present work.

2. Page 5, Paragraph 2. All the percentages here are based on very small numbers of patients (Figure 2c). The author should write down the real case number /total case number here as well.

3. Page 6. Paragraph 1. The FOT spanned eight orders of magnitude- But this does not mean anything as it is not about the protein copies.

4. Page 6. Paragraph 1. Some blood proteins are in the Top 10 abundant list. The authors are requested to perform additional analysis to exclude the possibility of blood contamination or to gauge the corresponding impact.

5. Page 6. Paragraph 2: how Muc19 2q is different from Muc19? Please add explanation.

6. Page 7. Paragraph 2: Please define SBS29 before the usage.

7. Please describe how the 400 samples are injected (if the sequence is randomized or not).

8. Page 11 Paragraph 3 and other places: “temporal molecular models”. The authors did not perform temporal analysis at all. They should consider using “staging” rather than “temporal.”

9. Methods: Please add the correct reference for the PARP1 activity assay.

10. Methods: Please add the correct reference for the Alanylation and de-alanylation reaction.

Reviewer #3 (Remarks to the Author): Expert in small intestine cancers

The authors should be congratulated on their Herculean effort to perform a very large genomic, proteomic, and phosphoproteomic analysis. Some of results appear to be significant to include chromosome 8 gain in the transition from intraepithelial neoplasia to invasive cancer and the relationship between DNA repair defects, upregulation of AARS1 and subsequent effects on PARP activation and reduction in apoptosis.

However, I have several concerns in regard to the paper.

1. While the title and summary suggests this is a comprehensive genomic and proteomic analysis on 140 cases of duodenal neoplasia, genomic analysis was only available from 31 cases with 88 samples. Thus, genomic information is performed on 22% of cases, and not sure what is meant by 88 samples. Multiple tumors were analyzed?

2. With 140 cases, one would expect a few documented cases of FAP and HNPCC, and I see no direct mention of this and how it fits in with the authors genomic and proteomic data.

3. While the authors take time to describe that various subtypes of duodenal neoplasia such as Type D and G, in Figure 1, subtypes P, N, L, and C are listed as substages, which is inaccurate.

4. I am not certain whether the Tobacco Signature is based on clinical history of patients using chewing tobacco. Moreover, in my review of the literature, to include some papers from Asia, I see no epidemiologic evidence to suggest smokeless tobacco or cigarette smoking conclusively increases the risk of duodenal adenocarcinoma, with only some limited data that there may be a risk of neuroendocrine tumors. This should be clearly discussed and explained in their discussion. Lastly, based on my review of the figures, the signature seems to be common in low-grade and high-grade dysplasia, but I don't really "see it" in adenocarcinoma.

5. Likewise, I am unclear if the "DNA Repair signature" is limited to cases with documented MSI-high tumors.

6. The sheer volume of figures and extended figures are overwhelming and made it difficult to follow the textual component of the paper in parallel with the graphical part. I think the data presentation should be substantially edited, and perhaps limit it to the proteomic and phosphoproteomic results of the analysis. Alternatively, to focus purely on Subtype D, the most common seen clinically may be of greater value to the readership.

The point-to-point response of the three referees are as follows:

Reviewer #1: Over Comments:

The manuscript contains a wealth of data related to improved classification of duodenal cancers, rare gastrointestinal tumors with limited therapeutic options. Authors analyzed 140 tumors, from which they obtained 406 macrodissected samples by manual scraping, and then performed several multi-omics analyses. The reader is overwhelmed by the impressive amount of data and substages, but I wonder whether the manual approach used qualifies for the complex analysis performed and for the even more complex final message.

Response: We appreciate the reviewer for the positive evaluation and constructive comments. We have revised the manuscript according to the comments. The point-to-point responses were as follows.

Q1: My major concern is with the sampling procedures. Starting from 140 patients, authors analyze 406 samples, obtained by manual macrodissection and scraping. Although expert pathologists annotated the tissue, this technique suffers of many limitations. In some samples, up to 6 different regions were scraped, I really wonder how far one can go with a macro-approach. I would like to receive some more details on this procedure and, most important, evidence should be provided that the scraped tissues correspond to the tissue annotated by the pathologists.

Response: We appreciate the constructive comments and insightful suggestions to improve the quality of our manuscript. According to the reviewer's comments, we divided the responses into two parts to answer: (1) about the microdissection and scraping of the samples in our cohort; (2) how far one can go with a micro-approach and about the evidence of the scraped tissues.

(1) About the microdissection and scraping of the samples in our cohort.

As for the sample preparation.

In our cohort, the tissues of duodenal cancer (DC) were classified into surface epithelial (D subtype) and non-surface epithelial components (G/NET/L/P/N/C subtypes). A total of 438 samples were collected from 156 DC patients who had not experienced prior chemotherapy or radiotherapy. All samples were separately dissected from the formalin-fixed, paraffin-embedded (FFPE) slides and provided by Zhongshan Hospital, Fudan University.

All the substage/subtype samples were evaluated by two or three experienced gastrointestinal pathologists, who would mark the hematoxylin and eosin (H&E)-stained sections (3 μ m-thick) of FFPE slides according to the proportion of tumor cells. The tumor purity of all the tumor samples was over 95%, indicating the high quality of all samples in our cohort. As for the normal duodenal tissue, the proportion of normal cell is 100%, while the tumor cell is 0% in our cohort.

As for laser-capture microdissection (LCM).

In our cohort, we applied LCM¹ (Science, 1996, PMID: 8875945) to dissect the sections of substage/subtype samples precisely, which is generally used to improve tumor purity² (Nature, 2015, PMID: 25719666).

All FFPE specimens were deparaffinized with xylene and rehydrated through graded alcohols and water. The H&E Sections were stained with Mayer's haematoxylin (Sigma) and dehydrated through graded alcohols and xylene. Before microdissection, FFPE specimens were sectioned with a microtome (10 μm thick) and mounted from FFPE blocks were micro-dissected with a Leica LMD 6500 laser microdissection system. According to the World Health Organization (WHO) classification, all samples were systematically evaluated to confirm the histopathologic diagnosis and variant histology by more than two expert gastrointestinal pathologists, and then an area of $1-5 \times 10^6 \mu\text{m}^2$ was collected (derived from dissected area \times slide thickness/average mammalian cell volume of $2,000 \mu\text{m}^3$, BNID 100434) (**Figure RL1, also shown in Extended Data Fig. 1c**). The substage/ subtype samples were collected in 1.5-ml tubes and kept in storage at -80°C until further processing. The methods are also applied in other published proteogenomic studies, such as in ovarian cancer³ (Nature, 2019, PMID: 31043742) and in glioblastoma⁴ (Nature communications, 2022, PMID: 35013227). We added this section in the “**Methods**” of the revised manuscript with red text (**Line 14–19, Page 26, and Line 1–8, Page 27**).

Figure RL1. The microdissection of the stage compartments of DC samples, also shown in Extended Data Fig. 1c in the revised version.

(2) How far one region can go with a micro-approach and about the evidence of the scraped tissues.

In our cohort, the H&E-stained sections were used to determine the zones of substage, which were then micro-dissected with a Leica LMD 6500 laser microdissection system. To avoid cross-tissues contamination among the substages or subtypes, we collected the tissues at the core area in each substage.

To present the quality of the samples in our cohort clearly, we provided the H&E-stained sections

of 438 samples in the revised version (Details shown in Supplementary material of the revised version).

Q2: In line with this, Authors state that the tumor purity of all samples was defined as the Score 5, which is the highest proportion of tumor cells. Then I don't understand why sample 1, which is substaged as normal, scores 5.

Response: Thanks for the reminder. We are sorry for the misdescription that tumor purity of normal tissue was defined as the Score 5. All substages and subtypes in our cohort were assessed by more than two experienced gastrointestinal pathologists on the basis of H&E-stained sections, who would mark the tumor purity according to the proportion of tumor cells (Score 0 = 0%; 0% < Score 1 < 20%; 20% ≤ Score 2 < 40%; 40% ≤ Score 3 < 60%; 60% ≤ Score 4 < 80%; 80% ≤ Score 5 ≤ 100%). The tumor purity of all the tumor samples was over 95%, and thus defined as the Score 5, indicating the high quality of all samples in our cohort (Figure RL2, also shown in Extended Data Fig. 1d). As for the normal duodenal tissue, the proportion of normal cell is 100%, while the tumor cell is 0% (Score 0) in our cohort (Figure RL2, also shown in Extended Data Fig. 1d). We corrected the description in the revised manuscript with red text (Line 4–7, Page 5).

Figure RL2 | Boxplot showing tumor purity of the samples in DC, also shown in Extended Data Fig. 1d in the revised version.

Q3. Duodenal cancer is a rare disease, critical point that the Authors clearly discuss in the paper. The rarity of the disease prevents from obtaining adequate number of samples, which here is overcome putting together a cohort that spans from 2012 to 2019. This reviewer, however, is concerned with the reproducibility of such a complex analysis (including proteomic and WES) on FFPE sections that date back to 2012, or at least I am concerned on putting together samples that span such a long time. Quality controls to show the quality of material obtained from the older specimens is needed.

Response: We thank the reviewer for the constructive suggestions and agree with reviewer's points. According to the reviewer's comments, we divided the response into two parts to answer: (1) about the quality control of the sample collection and preparation in our cohort; (2) about the stability of the DC tissues/samples in our cohort.

(1) About the quality control of the sample collection and preparation in our cohort.

In our cohort, all substage samples were separately dissected from the FFPE slides, spanning from 2012 to 2019. It is reported that the stability of FFPE samples is up to 15 years for proteome⁵ (Molecular oncology, 2019, PMID: 31495056). Consistently, our previously published studies illustrated the tissues integrity and stability (e.g., protein, DNA) of FFPE samples spanning nearly 10 years⁶ (Journal of hematology & oncology, 2022, PMID: 35659036).

In addition, all the samples, collected in our cohort, were well preserved and systematically evaluated to confirm the histopathologic diagnosis and variant histology according to the World Health Organization (WHO) classification by more than two expert gastrointestinal pathologists, who determined the tissues segment based on the tumor content (> 95%), the presence and extent of tumor necrosis (< 5%), and signs of invasion into the muscularis propria. According to the clinical sample collection procedures in CPTAC and other published studies⁷ (Cell, 2019, PMID: 31585088), the following criteria were used for samples on the 438 samples, 1) successful extraction in DNA and protein; 2) no tumor cells in the normal tissue. We added this section in the “**Methods**” of the revised manuscript (**Line 19–26, Page 24**).

Furthermore, to present the quality of the samples in our cohort clearly, we provided the H&E-stained sections of 438 samples in the revised version (**Details shown in Supplementary material in the revised version**), showing the similar pathological features of the same stages and subtypes in the samples that span from 2012 to 2019.

(2) About the stability of the DC tissues/samples in our cohort.

We thank the reviewer for the insightful comments. The stability of the DC tissues/samples ensures the quality of the data analysis.

In our cohort, 438 samples, spanned from 2012 to 2019, were collected and processed with the same standard operating procedure. To assess the stability of the samples in our cohort, we firstly analyzed the identification number of the samples that span from 2012 to 2019. As a result, we found that there was no significant difference in the protein identification number of the samples with different storage years at the same pathological stage (**Figure RL3a**), indicating that no impacts of the time spanning from 2012 to 2019 on the protein identification of the samples in our cohort.

To further evaluate whether the storage years affect the intra-group correlation of the samples, we focused on the samples at the one and same stage in the adenocarcinoma (D) and Brunner’s gland (G) subtypes, and the performed (Spearman’s) correlation analysis on the samples in different years. As a result, the comparable (Spearman’s) correlation coefficients were observed in the samples at the one and same stage (e.g., non-tumor (NT) stage) spanning from 2012 to 2019 both in D and G subtypes of DC (**Figure RL3b, c**), suggesting that no impacts of the time spanning from 2012 to 2019 on the intra-group correlation of the samples in our cohort.

Housekeeping (HK) genes are involved in basic cell maintenance, and therefore, expected to maintain constant expression levels. Thus, we analyzed the expression of housekeeping genes in our cohort (https://www.tau.ac.il/~elieis/HKG/HK_genes.txt), and found that the expression of the

HK genes (e.g., ASNA1, ACIN1, ADK, etc.) was constant at the protein level in the NT stage samples of the D subtype spanning from 2012 to 2019 (**Figure RL3d**). These results further implied that no impacts of the time spanning from 2012 to 2019 on the protein identifications in this study.

Therefore, the comparable protein identification number and correlations of the samples, and the constant expression of HK genes, indicated the stability and high quality of the samples in our cohort spanning from 2012 to 2019.

Figure RL3 | The quality control of the samples that span from 2012 to 2019, also shown in Extended Data Fig. 2f in the revised version. a, The protein identification number of the samples spanning from 2012 to 2019. **b**, Spearman's correlation coefficients of the NT stage samples spanning from 2012 to 2019 in the D subtype. **c**, Spearman's correlation coefficients of the NT stage samples spanning from 2012 to 2019 in the G subtype. **d**, the protein-levels of housekeeping proteins of the NT stage samples spanning from 2012 to 2019 in the D subtype.

Q4. Authors report a classification of the tissues into several stages and substages. Supp Table 1 reports age and gender of patients, but less clear is the substage information, which is explained in extended Figure 1. A reminding to this Figure would help understanding the Table. Also, in Extended Figure 1, there is no explanation what the dotted lines on the H&E pictures refer to. Finally, considering the high numbers of stages and subtypes, colore-codes should help, while in some figures (e.g. Fig 1A), color-code enhances the complexity.

Response: Thanks for the carefully read and insightful comments. According to the comments, we divided the responses into three parts to answer: (1) about the lack of substage information in

Supplementary Table 1; (2) about the explanation of the dotted lines on the H&E pictures; (3) about the complex color-codes in some figures.

(1) About the lack of substage information in Supplementary Table 1.

We are sorry for the unclear description of the substage information in Supp Table 1. In the revised version, we have added the pathological information of 438 samples in the **Supplementary Table 1a**.

To present the substage information clearly, we added the H&E-stained sections of the two major (D and G) and 5 rare (NET, L, P, N, and C) subtypes of our cohort in **Extended Data Fig. 1e**.

Furthermore, to describe our cohort clearly, we added the supplemented table (**Table RL1–RL3, also shown in Table 1-3**) in the “**Methods**” of the revised manuscript with red text (**Line 1, Page 25, and Line 5–7, Page 26**).

Table RL1. Clinical and histopathologic characteristics of Fudan DC cohort (n = 156).

Characteristic	n (%)	Parameter	n (%)
Age		Survival status	
< 50 yr	38 (24.4)	Alive	156 (100)
50~70 yr	101 (64.7)	Death	0 (0)
>70 yr	17 (10.9)	Histopathologic (438 samples)	
Gender		Epithelial components	
Male	55 (35.3)	Yes	325 (74.2)
Female	101 (64.7)	No	113 (25.8)
Smoking		Histology subtype	
Yes	37 (23.7)	Adenocarcinoma (D)	325 (74.2)
No	92 (59.0)	Brunner's gland (G)	97 (22.1)
NA	27 (17.3)	Heterotopic pancreas (P)	1 (0.2)
Drinking		Intermuscular gland (N)	9 (2.1)
Yes	30 (19.2)	Lymphangioma (L)	2 (0.5)
No	93 (59.6)	Cystic dystrophy (C)	3 (0.7)
NA	33 (21.2)	Neuroendocrine tumor (NET)	1 (0.2)
FAP			
Yes	2 (1.3)		
No	116 (74.4)		
NA	38 (24.4)		
HNPCC			
Yes	4 (2.6)		
No	83 (53.2)		
NA	69 (44.2)		

Table RL2. Subclassification information and the number of substage samples in the D subtype.

Stages	Substages (n = sample number)
Non-tumor stage (NT stage)	1 (n = 150), 1_1 (n = 1), 1_2 (n = 3), 1_3 (n = 7), 1_4 (n = 4), 1_5(n = 1),

	1_6 (n = 1)
Low-grade intraepithelial neoplasia stage (LGIN stage)	2_1 (n = 22), 2_2 (n = 11), 2_3 (n = 21), 2_4 (n = 8)
High-grade intraepithelial neoplasia stage (HGIN stage)	3_1 (n = 13), 3_2 (n = 9), 3_3 (n = 24), 3_4 (n = 25)
Minimally invasion adenocarcinoma stage (MIA stage)	4 (n = 5)
Duodenal adenocarcinoma stage (DAC stage)	5 (n = 20)

Table RL3. Subclassification information and the number of substage samples in the G subtype.

Stages	Substages (n = sample number)
Normal stage (N-G stage)	G_1 (n = 47)
Hyperplasia stage (H-G stage)	G_2 (n = 19), G_2_1(n = 7)
Adenoma Brunner's stage (A-G stage)	G_3 (n = 22), G_3_1 (n = 1), G_3_2 (n = 1)

(2) About the explanation of the dotted lines on the H&E pictures.

We apologize for the no explanation of the dotted lines on the H&E pictures in the **Extended Data Fig. 1c (original version, also shown in Extended Data Fig. 1e of the revised version)**. In the revised version, to present the figure clearly, we draw a dotted box with arrows to show the pathological characterization of the components of DC (**Figure RL4, also shown in Extended Data Fig. 1e**), and added the description in the legend.

Figure RL4 | The H&E staining of the components in DC, also shown in Extended Data Fig. 1e in the revised version. The dotted box with arrow(s) showing the pathological characterization of the substages and subtypes.

(3) About the complex color-codes in some figures.

Thanks for the reminder, and we apologize for the complex color-codes in some figures. To present the figures clearly and readably, we re-assigned the stages, subtypes, clusters, etc. with differentially visualized color-codes. In addition, we focused on the visual and readable color-codes in each figure, and removed the redundant and un-visualized information. For example, we removed the un-visualized color-codes presentation of the stages/subtypes and other clinic information in **Fig. 1a**, **Fig. 1b**, **Fig. 2a**, **Fig. 3f**, **Fig. 5a**, etc. Please see the details in the revised version.

Q5. Which data was used to perform the xCell enrichment analysis, which is based on RNA sequencing data? Authors mention immune, stromal and microenvironmental signatures. In what do the signatures differ? Which color-code was assigned to each cluster in Fig 1A?

Response: We appreciate the professional comments to improve the quality of our manuscript. According to the comments, we divided the responses into three parts to answer: (1) about the xCell enrichment analysis on the basis of proteomic data; (2) about the differences among the immune, stromal, and microenvironmental signatures; (3) about the color-code assigned to each cluster in Fig. 1a.

(1) About the xCell enrichment analysis on the basis of proteomic data.

In our cohort, we presented a comprehensive characterization of early-stage DC at the gene, protein, and phosphoprotein levels. Owing to the limited volume of the samples in the early-stage DC, the RNA sequencing was not conducted in our cohort. To investigate the immune filtration of DC, we performed the xCell enrichment analysis based on the proteomic data, and the same methods were also applied in other published studies, including clear cell renal cell carcinoma⁸ (Cell, 2019, PMID: 31675502), cholangiocarcinoma⁹ (Hepatology, 2022, PMID: 35716043), and urothelial carcinoma of the bladder⁶ (Journal of hematology & oncology, 2022, PMID: 35659036).

In addition, proteomic data is often applied to investigate the immune filtration and the response to immunotherapy. For example, Keren et.al elucidated a structure tumor-immune microenvironment in triple negative breast cancer¹⁰ (Cell, 2018, PMID: 30193111). The landscapes of response to immunotherapy in melanomas was based on the proteomic data¹¹ (Cell, 2019, PMID: 31495571).

To present the application of xCell clearly, we added this section in the “**Methods**” of the revised version with red text (**Line 24–27, Page 35, and Line 1–4, Page 36**).

(2) About the differences among the immune, stromal, and microenvironmental signatures.

According to the reviewer’s comments, we divided the responses into three parts to answer, and the details were listed as follows.

As for the different components of the signatures.

xCell (<https://xcell.ucsf.edu>) is a novel gene signature-based method to present a generate gene signatures for 64 immune and stromal cell types spanning multiple adaptive and innate immunity cells, hematopoietic progenitors, epithelial cells, etc.¹² (Genome biology, 2017, PMID: 29141660). According to the enrichment of the cell types, the immune signature is the sum of immune compositions, including T cell, monocytes, etc. The stromal signature is the composition of stromal related components, such as fibroblasts, endothelial cells, etc. Most of the cell types are part of complex cellular heterogeneity of the tumor microenvironment, and thus the microenvironment signature is the combination of all immune and stromal cell types.

As for the association between the signatures and the immune infiltration in DC.

To investigate the difference of the immune infiltration of the subtypes in DC, we applied xCell¹² (Genome biology, 2017, PMID: 29141660) to the 438 samples of DC cohort. As a result, we found that higher stromal scores (Kruskal-Wallis test, adjust. $p = 9.6E-5$) in the non-surface epithelial components of DC, including G, N, and C subtypes (**Figure RL5a, also shown in Extended Data Fig. 6d**). Specifically, we found gradually enhanced microenvironment score both in the D (Kruskal-Wallis test, adjust. $p = 2.4E-8$) and G (Kruskal-Wallis test, adjust. $p = 4.3E-8$) subtypes progression (**Figure RL5b, c**). Whereas, stromal score was notably descended in the D subtype progression (Kruskal-Wallis test, adjust. $p = 2.0E-5$), which was not observed in the G subtype

progression (Figure RL5b, also shown in Extended Data Fig. 6e). These findings indicated that diversity of the signatures was associated with the subtypes of DC, allowing us to explore the diverse immune characterization of the signatures based on the DC subtypes.

Figure RL5 | Different immune filtration in two major and five rare subtypes of DC, also shown in Extended Data Fig. 6d, e, in the revised version. a, Barplot showing the different immune score (left), stromal score (middle), and microenvironment score (right) in the two major and five rare subtypes of DC (Kruskal-Wallis test). **b,** Scatterplot presenting the immune score (left), stromal score (middle), and microenvironment score (right) in the D subtype progression (Kruskal-Wallis test). **c,** Scatterplot presenting the immune score (left), stromal score (middle), and microenvironment score (right) in the G subtype progression (Kruskal-Wallis test). **** $p < 1.0E-4$, *** $p < 1.0E-3$, ** $p < 1.0E-2$, * $p < 0.05$, ns. > 0.05 .

As for the different molecular characterization of the signatures among the DC subtypes.

To further deconvolute the diverse molecular characterizations of immune, stromal, and microenvironmental signatures, we performed consensus clustering using xCell enrichment scores of 438 samples in DC, which was then stratified into 4 immune-based clusters as *Hot tumor* cluster, *Cold tumor* cluster, *Epithelial* cluster, and *B cells* cluster (Figure RL6a, b, also shown in Fig. 5a and Extended Data Fig. 6a). Remarkably, the non-surface epithelial components of DC, such as the G, N, and C subtypes, were notably observed in the *Cold tumor* cluster and the *B cells* cluster, which were characterized by high stromal score (Figure RL6c, also shown in Extended Data Fig. 6c). The D subtype, closely co-clustered with NET/L/P subtypes, was overrepresented in the *Hot tumor* cluster and the *Epithelial* cluster. The *Hot tumor* cluster with the highest immune score and microenvironment score, exhibited positive association with T cell activation, evidenced by the

highest degree of CD4⁺ T cells, CD4⁺ Tcm, CD8⁺ T cells, CD8⁺ Tcm (**Figure RL6d, also shown in Fig. 5c**). In addition, the highest HLA-I score and HLA-II score were detected in the *Hot tumor* cluster, evidenced by the expressions of the corresponding MHC molecules (e.g., HLA-B/C/E, HLA-DRA, etc.) (**Figure RL6d, e, also shown in Fig. 5b, c**).

Consistently, pathway enrichment analysis revealed that the D subtype prominent pathways were dominant in the *Hot tumor* cluster, such as cell cycle, apoptosis, DNA repair (Kruskal-Wallis test, $p = 2.9E-14$), and so on (**Figure RL6e, f, also shown in Fig. 5b, d**). Moreover, the G subtype featured pathways, such as focal adhesion, cell-cell adhesion, were overrepresented in the *Cold tumor* cluster (**Figure RL6e, f, also shown in Fig. 5b, d**). Together, these results further illustrated the diverse molecular characterization of the immune infiltration of DC was correlated with the subtypes. Specifically, the immune/microenvironment and stromal signatures of DC were relevant to the D subtype and G subtype, respectively.

for proteomic profiling, as well as in Fig. 5a. e, The HLA-I score (left) and HLA-II score (right) in the four immune clusters (Kruskal-Wallis test). f, Density contours of immune and stromal scores of four immune clusters. **** $p < 1.0E-4$, *** $p < 1.0E-3$, ** $p < 1.0E-2$, * $p < 0.05$, ns. > 0.05 .

(3) About the color-code assigned to each cluster in Fig. 1a.

We are sorry for not presenting the immune clusters with color-codes in the Fig. 1a. In the revised version, we added the immune cluster information in Fig. 1a (Figure RL7). Please see the details in the revised version.

Figure RL7 | Overview of the experimental design and the number of samples for the genomic, proteomic, and phosphoproteomic analyses, also shown in Fig. 1a in the revised version.

Q6. Extended Figure 1b. Lamina proria should read lamina propria. Panel C is not clearly explained in the legend. What is NA?

Response: Thanks for the reminder and we apologized for the mistakes. In the revised manuscript, we corrected “lamina proria” as “lamina propria” (shown in Extended Data Fig. 2b in the revised version).

As for the “NA” which was marked in gray, it indicated that there were no samples of that substage. To present the figures clearly, we had changed the figure as follows (Figure RL8, also shown in Extended Data Fig. 1d).

Figure RL8 | Boxplot showing tumor purity of the samples in DC, also shown in Extended Data Fig. 1d in the revised version.

Reviewer #2 (Remarks to the Author): Expert in MS-based proteomics and proteogenomics

Li et al. described an ambitious study in which duodenal cancer (DC) tissues were measured by multi-omics. They extensively analyzed the resultant data, acquired preliminary hints for the potential therapeutic targets, and performed some additional experiments on AARS1 and PARP1. Overall, the datasets are large and novel. However, significant issues have to be resolved.

Response: We appreciate the reviewer’s constructive and insightful comments, which help to improve the quality of the manuscript. The point-to-point responses were as follows.

Major concerns:

Q1. Proteogenomic landscape of DC. In this study, the authors relied on pathological subtypes (based on the evaluation of two or three pathologists) rather than the molecular landscape to perform all their analyses. This has to be improved. All previous proteogenomic studies used proteomic or genomic molecular signatures to understand the cancer heterogeneity and phenotypes. The underlying assumption is that molecular layers like proteome level could provide novel insights in stratifying patients. The authors should follow this best practice and also compare the result to pathological subtypes.

Response: Thanks for the insightful comments, and we agree with the reviewer’s point that molecular landscape at proteome level were applied to validate the pathological subtypes, providing novel insights in cancer heterogeneity and phenotypes.

In our cohort, the tissues were classified into surface epithelial (adenocarcinoma, D subtype) and non-surface epithelial components, including Branner gland (G subtype), heterotopic pancreas (P subtype), intermuscular gland (N subtype), lymphangioma (L subtype), cystic dystrophy (C subtype), well-differentiated neuroendocrine tumor (NET subtype). The subtype-based classification allowed us to explore the diverse molecular characterizations of the DC subtypes.

In the original version, we performed proteogenomic analysis based on the pathological subtypes to

investigate key events and elucidated the molecular characterization of staging waves during the carcinogenesis of duodenal cancer (DC). For example, we found the *DST* mutation enhanced mTOR signaling in the duodenal adenocarcinoma (DAC) stage of the D subtype.

In the revised version, to explore the association between pathological subtypes and proteomic subtyping, we performed consensus clustering analysis on 438 samples of DC. As a result, we identified two clusters (**Figure RL9a, also shown in Extended Data Fig. 5a**). Specifically, cluster 1 (CCP1) mainly consisted of the non-surface epithelial components, including G, NET, L, P, N, and C subtypes (**Figure RL9b, also shown in Extended Data Fig. 5b**). The most of the samples in the surface epithelial component (D subtype) were observed in the cluster 2 (CCP2) (**Figure RL9b, also shown in Extended Data Fig. 5b**). These results indicated that the proteomic methods could be applied to the distinction of the pathological layers.

To investigate the molecular characterizations of the two clusters (CCP1 and CCP2), we integrated the differential expressed proteins (DEPs, $n = 3,102$) between CCP1 and CCP2 (Wilcoxon signed-rank test, adjust. $p < 0.05$, CCP2 vs. CCP1 ratio ≥ 2 or ≤ 0.5), and found 1,110 and 1,992 proteins were overrepresented in the CCP1 and CCP2, respectively. The CCP1 proteins participated in ECM signaling (e.g., COL1A1, THBS3, etc.) and complement cascade (e.g., CFD, C5, etc.). In CCP2, the dominant pathways were cell cycle (e.g., CDK4, RB1, etc.), apoptosis (e.g., BAX, CASP3, etc.), and glycolysis (e.g., ENO1, HK2, etc.) (**Figure RL9c, also shown in Extended Data Fig. 5c**).

According to the subtype-based features of the DC patients, 156 DC cases were divided into 4 panels: Panel 1 (P1, $n = 60$, patients with D subtype only), Panel 2 (P2, $n = 54$, patients with G subtype only), Panel 3 (P3, $n = 32$, patients with both the D and G subtypes), and Panel 4 (P4, $n = 10$, patients with other subtypes) (**Figure RL9d, also shown in Fig. 4a**). In addition, the P1 and P2 were overrepresented in the CCP2 and CCP1 (Chisq-test, $p < 2.2E-16$), respectively (**Figure RL9e, also shown in Extended Data Fig. 5e**). In addition, in the P3, the proportions of CCP1 and CCP2 were comparable, and while the P4 was overrepresented in the CCP1 (**Figure RL9e, also shown in Extended Data Fig. 5e**). The findings indicated that the molecular characterization of subtype-based panels might be relevant to that of the proteomic clusters. In addition, the proteomic cluster presented that the featured protein patterns and dominant pathways were detected in the subtypes and stages in DC.

To further explore the molecular features of the subtype-based panels, we analyzed the DEPs of the panels. As a result, we found the P1 highly expressed proteins were enriched in cell cycle (e.g., CDK1/2/4/6, SKP1, CDKN2A, etc.), apoptosis (e.g., CASP3/8, AARS1, PARP1/9, etc.), and glycolysis (e.g., HK2, PGK1, ENO1/2/3, etc.) (**Figure RL9f, also shown in Fig. 4b**). In the P2, ECM signaling (e.g., COL1A1, LAMA1, TNXB, etc.) and complement cascade (e.g., CFH, C4A, C6, etc.) were predominant. In addition, the P3, both containing the D and G subtypes, had the characteristics of the P1 and P2 (**Figure RL9g, also shown in Extended Data Fig. 5f**). The represented pathways of the P1 (D subtype) and P2 (G subtype) were consistent with those in the CCP1 and CCP2, respectively.

Together, the links of the findings in the proteomic clusters and subtype-based panels, indicated the

correlations between the proteomic clusters and the pathological stages/subtypes of DC (**Figure RL9h**). In addition, those consistency suggested the advances and applications of the proteome in stratifying patients and understanding the cancer heterogeneity and phenotypes. Based on the proteomic methods and subclassification in the pathological layers, diverse subtypes/substages were collected, promoting us to explore the origins and carcinogenesis of DC subtypes. We thank the reviewer for the insightful comments again, and we added this section in the revised manuscript with red text (**Line 15–28, Page 10, Line 1–25, Page 11**).

Figure RL9 | The correlation between pathological layers and molecular layers at proteome level, also shown in Fig. 4 and Extended Data Fig. 5 in the revised version. a, Sankey diagram analysis of all the 438 samples from 156 cases classified into two proteomic clusters. **b,** The proportions of the two clusters in DC subtypes (Chisq-test). **c,** The represented pathways (left) and related proteins (right) in the CCP1 and CCP2 (Wilcoxon signed-rank test). **d,** Sankey diagram

analysis of all the 156 cases classified into 4 panels. **e**, Histogram showing the proportion of the proteomic clusters in the panels (Chisq-test). **f**, Venn diagram depicting the specific driven pathways in the P1 and P2. The number represented the upregulated DEPs. The blue and orange indicated the D and G subtypes, respectively. **g**, The represented pathways of the normal tissues, D subtype, and G subtype in the P3. **h**, Sankey diagram analysis showing a brief overview of the correlation among the stages, subtypes, panels, and proteomic clusters. **** $p < 1.0E-4$, *** $p < 1.0E-3$, ** $p < 1.0E-2$, * $p < 0.05$, ns. > 0.05 .

Q2. Their proteome and phosphoproteome data both contain lots of missing values, according to the Supplementary table 3. Fairer descriptions of really quantified protein and phosphopeptide numbers are urged. Although they mentioned the sample amount is needed, it seems other advanced techniques like the Pressure cycling technique (PCT) can already handle the tiny amount of clinical material very well. Please add relevant discussions if appropriate. The authors are advised to consider the possible bias of protein abundance in their data, result, and interpretation. Also, please provide additional analysis on how missing values are handled and affect the results reported.

Response: Thanks for the constructive comments and insightful suggestions. According to the reviewer's comments, we divided the response into three parts to answer: (1) about the discussion of "Pressure cycling technology (PCT) and the application in proteomics" in revised manuscript; (2) about the description of quantified proteins and phosphopeptide numbers and the possible bias; (3) about the missing values and the impacts on the data analysis and conclusion.

(1) About the discussion of "PCT and the applications in proteomics" in revised manuscript.

Pressure cycling technology (PCT), also known as barocycling, is utilized to extract protein from ultra-small tissue samples¹³ (Proteomics, 2020, PMID: 31881116) and effectively minimized sample loss. The tissue proteins can be extracted and digested in the same tube using PCT, increasing throughput to 16 concurrently processed samples, and decreasing sample processing time less than 8 hours. To effectively and reproducibly digest the tissue samples, the molecular interactions were destabilized through the rapidly and repeatedly raising and lowering pressure in the reaction vessel from ambient to high levels (up to 35,000 psi [240 MPa])¹⁴ (Methods Mol Biol, 2012, PMID: 22639209), and thus enhance the enzymatic activity and speed up the tissue sample digestion. In addition, the complex tissue lysis and protein digestion procedures, such as Barocyclers (NEP3229, NEP2320, and HUB440), used for regulating the exquisite process of PCT¹⁴ (Methods Mol Biol, 2012, PMID: 22639209). Therefore, the PCT-assisted label-free quantitative proteomics workflow allowed cataloging the deepest proteomic analysis on a rapid timescale.

Guo et al. illustrated that the PCT is applicable for the dry mass of FFPE pouch, which weighs about 300 μg ($0.5 \times 0.5 \times 3$ mm, 5 μm thickness)¹⁵ (Molecular Oncology, 2019, PMID: 31495056). However, the mass of the most FFPE tissues in our cohort was under 100 μg . Therefore, the method of tris-2-carboxyethylphosphine (TCEP) extraction protein was applied in our cohort.

TCEP, a priority reducing agent, leads to high internal standard recovery (> 95%) combined with iodoacetamid (IAA)¹⁶ (Molecules, 2022, PMID: 33348658). In addition, we previously have achieved deep proteome and phosphoproteome coverage of the definite volume samples (nearly less than 50 µg) in early-stage gastrointestinal cancer¹⁷ (Proteome science, 2022, PMID: 35397555) on the basis of the method of TCEP extraction protein. In addition, the protein identification number and coverage of ultra-tiny samples by TCEP was comparable to those advanced-stage cancer samples in other published studies (**Figure RL10a, b, c**), such as in GC cohort¹⁸ (Nature communications, 2018, PMID: 29520031), and HCC cohort¹⁹ (Nature, 2019, PMID: 30814741).

Figure RL10 | The identification and coverage in the Fudan cohort and other cohorts (GC cohort, HCC cohort). **a**, The protein identification number in Fudan cohort and other cohorts. **b**, Overlap showing the TFs identified in the Fudan cohort and other cohorts. **c**, Overlap showing the identified plasma membrane related proteins in the Fudan cohort and other cohorts. The transcriptional factors (TFs) datasets and plasma membrane protein datasets were referenced from (HPA: <https://www.proteinatlas.org>).

Note: NATs: tumor paired non-cancerous adjacent tissues.

Therefore, the method of TCEP extraction protein was appropriate for our study (details shown in “**Methods**” in **Line 5–22, Page 31**), and we achieved deep proteome coverage of such a tiny FFPE tissue volume with a total of 14,426 proteins detected in 438 samples in DC progression, providing new insights in the tiny amount of clinical material.

In the revised version, we added the discussion of “Pressure cycling technology (PCT) and its application in proteomics” in “**Discussion**” with red text (**Line 18–24, Page 20**). The details were shown as follows.

“To achieve deep proteome coverage of such a tiny FFPE tissue volume, pressure cycling technique (PCT) would be an available method that effectively minimize sample loss¹³ (Proteomics, 2020, PMID: 31881116), while the lowest operable weight (about 300 µg) is still higher than that in our study (under 100 µg). On the basis of the method of tris-2-carboxyethylphosphine (TCEP) extraction protein, with high internal standard recovery (> 90%)¹⁶ (Molecules, 2022, PMID: 33348658), we have achieved deep proteome and phosphoproteome coverage of the trace samples in early-stage gastrointestinal cancer¹⁷ (Proteome science, 2022, PMID: 35397555). Thus, the method of TCEP extraction protein was applied to explore the molecular characterizations in DC progression”.

(2) About the description of quantified proteins and phosphopeptide numbers, and the possible bias.

As for the protein and peptide identification number.

A total of 438 samples from 156 DC patients for proteomic profiling, and all the proteome data were processed as follows (**Figure RL11a, also shown in Fig. 1f**): EDC1 (14,426 GPs): all 14,426 gene products (GPs) identified in 438 samples (156 EDC cases) on the basis of the match between runs (MBRs) algorithm²⁰ (Nature protocols, 2016, PMID: 27809316); EDC2 (11,904 GPs): all the proteins were required to have at least 2 unique strict peptide, and we excluded keratins proteins of which the maximum FOT of all 438 samples were less than 1.0E-5; EDC3 (7,400 GPs): GPs identified in more than 20% samples of every substage, which was also applied in other published studies^{18,21} (Nature communications, 2018, PMID: 29520031; Cell, 2020, PMID: 34358469).

At the phosphoprotein level, a total of 111 samples from 49 DC patients were conducted, and 38,696 phosphosites, corresponding to 9,716 phosphoproteins, were identified (**Figure RL11b, c, also shown in Fig. 1g and Extended Data Fig. 2h**). The phosphosites (n = 12,851), identified in more than 20% samples of every substage, were used in the analysis of our cohort. In addition, the phosphoprotein/phosphopeptide abundance was adjusted by the total protein counterpart abundance, which has been applied in previous published studies²² (Nature communications, 2021, PMID: 34400640).

Figure RL11 | The identification number of samples in the Fudan cohort. a, The cumulative number of protein identifications in DC progression. **b,** The cumulative number of phosphosite identifications in DC progression. **c,** The cumulative number of phosphoprotein identifications in DC progression.

In our study, though the volume is limited, the identified number of DC samples was comparable to other cohorts at the protein and phosphoprotein levels (**Figure RL12a, b**), such as GC cohort¹⁸ (Nature communications, 2018, PMID: 29520031), HCC cohort¹⁹ (Nature, 2019, PMID: 30814741), and LUAD cohort²³ (Cell, 2020, PMID: 32649877).

Figure RL12 | The comparable identifications in the Fudan cohort and other cohorts (GC cohort, HCC cohort, LUAD cohort). a, The protein identification number in the Fudan cohort, the HCC cohort, and the GC cohort. **b,** The phosphosite identification number in the Fudan cohort and the LUAD cohort.

Note: NATs: tumor paired non-cancerous adjacent tissues.

As for the possible bias in proteome.

We thank the reviewer for the insightful comments. The technical restriction of MS generally causes

bias in proteome. In our cohort, possible bias is still inevitable, though the in-depth coverage of the sample identification was comparable with other cohorts^{19,23} (Cell, 2020, PMID: 32649877; Nature, 2019, PMID: 30814741).

Firstly, we defined 5 protein classes according to their observation percentiles: Class 1 (n = 819, percentiles < 1%), Class 2 (n = 1,603, 1% ≤ percentiles < 5%), Class 3 (n = 804, 5% ≤ percentiles < 10%), Class 4 (n = 1,288, 10% ≤ percentiles < 20%), and Class 5 (n = 1,411, 20% ≤ percentiles < 40%) (**Figure RL13a**). Among of the 5 protein classes, we assumed that the proteins with low observation percentiles, were underrepresented in our cohort.

Secondly, to explore the functional pathways of the underrepresented proteins, we performed GO/KEGG enrichment analysis of the proteins in 5 classes. As a result, we found the underrepresented proteins (Class 1) were related to the plasma membrane/ transcriptional factors (TF) in our cohort (**Figure RL13b**). To explore whether the possible bias was also observed in other cohorts, we downloaded the proteomic datasets from other published studies, such as GC cohort¹⁸ (Nature communications, 2018, PMID: 29520031) and HCC cohort¹⁹ (Nature, 2019, PMID: 30814741), and also defined 5 protein classes (Class 1 to 5) according to their observation percentiles similar as in our cohort. Consistently, the plasma membrane/ transcriptional factors (TF)-related proteins were also underrepresented in the GC cohort and HCC cohort (**Figure RL13b**).

Together, the possible bias was the limits for proteome, and showed association with TF/plasma membrane- related proteins, which might be the challenges for understanding the characteristics of DC. The advances of MS with more in-depth coverage of identification in proteome needed to dissect the bias in proteome, providing a more comprehensive proteomic landscape. We thank the reviewer for the reminder and professional suggestions again, and will investigate the underrepresented proteins for further analysis in the future. We added this section in the “**Discussion**” of the revised manuscript with red text (**Line 17–21, Page 23**).

Figure RL13 | The bias of protein abundance in the Fudan cohort and other cohorts. a, Overview of 5 classes of proteins according to their observation percentiles. **b,** The protein components of the bias proteins in the Fudan cohort and other cohorts (GC and HCC).

(3) About the missing values and the impacts on the results.

As for the missing values.

For the missing value in our study, we firstly applied MBRs algorithm^{20,24} (Nature biotechnology, 2008, PMID: 19029910; Nature protocols, 2016, PMID: 27809316) in this study, which has been proved to be an effective technique to fill the missing values, which was widely used in other proteomic studies¹¹ (Cell, 2019, PMID: 31495571). Detailly, we built a dynamic regression function based on common identified peptides in samples. According to correlation value R^2 , the function chooses linear or quadratic function for regression to calculate retention time (RT) of corresponding hidden peptides, and check the existence of the extracted ion chromatogram (XIC) based on the m/z and calculated RT. The function evaluated the peak area values of those existed XICs. These peak area values are considered as parts of corresponding proteins. This strategy has been applied in other published proteomic studies^{18,25} (Nature communications, 2018, PMID: 29520031; Nature communications, 2019, PMID: 30604760). We added the information of the “Data imputation” in the “**Methods**” of the revised manuscript with red text (**Line 1–9, Page 34**).

As for the impacts of missing value on the data analysis and conclusion.

To investigate the impacts of missing value, we performed comparative analysis of the DEPs (Kruskal-Wallis test, adjust. $p < 0.05$) in DC progression, which were detected in more than 20%, more than 40%, more than 60%, and more than 80% of samples, respectively. The results revealed that DEPs (Kruskal-Wallis test, adjust. $p < 0.05$) from four thresholds (20%, 40%, 60%, and 80%) represented similar pathway enrichment, such as cell cycle, DNA repair were dominant in the D subtype, ECM signaling and focal adhesion were overrepresented in the G subtype (**Figure RL14**). These findings indicated that the missing value could not change the results and conclusion of the further analysis in our cohort.

Figure RL14 | Represented pathways Enrichment of DEPs from four thresholds (20%, 40%, 60%, and 80%).

Q3. The phosphoproteomic analysis. Throughout the study, the phosphoproteome result was mentioned with an impressive (total) identification number, but almost not used at all in their data and analysis. This might be due to the limited data quality (about 4500 phosphopeptides identified in each of the samples). Therefore, they should either consider discarding the phosphoproteomic data (if it is not good enough for Nature Communications) or considerably improve the data analysis of phosphoproteomics, and really use this level of their “multi-omics” to support all their analysis, e.g., pathway and signaling annotation, etc. Accordingly, they can also shorten the text of other parts (e.g., immune-cluster, 8q-related), which are purely based on data mapping, P values, and their own data interpretation.

Response: We appreciate the constructive comments and suggestions to improve the quality of our research. According to the reviewer’s comments, we divided the responses into two parts to answer: (1) about the limited data quality for phosphoproteome; (2) about the integrated proteogenomic analysis; (3) about shortening the text. The details were shown as follows.

(1) About the limited data quality for phosphoproteome.

We are sorry for not presenting the data quality for phosphoproteome clearly, and we added this section in the “Methods” of the revised manuscript (Line 24–28, Page 33). The details were shown as follows.

As for the protein identification number of the phosphoproteome.

In the revised version, to further improve the solid results of our research, we re-evaluate the volume of 406 samples, of which 71 samples from the 31 DC patients were adequate for phosphoproteomic profiling. In addition, apart from 406 samples, we added 16 DAC stage samples with paired normal tissues from 16 D subtype patients for proteomic profiling and whole-exome sequencing (WES), in which 4 samples (2 DAC stage sample + paired normal tissues samples) from 2 D subtype patients were conducted for phosphoproteomic profiling. That indicated 75 samples from 33 DC patients were added for phosphoproteomic profiling (Table RL4). Therefore, a total of 111 samples from 49 DC patients were conducted for phosphoproteomic profiling (Figure RL15, also shown in Fig. 1a).

Table RL4 | The number of the samples for genome (WES), proteome, and phosphoproteome in the Fudan cohort.

Omics	Original version			Revised version		
	Total	Major subtypes	Rare subtypes	Total	Major subtypes	Rare subtypes
Genome	88	82	6	120	114	6
Proteome	406	390	16	438	422	16
Phosphoproteome	36	36	0	111	111	0

Major subtypes: D and G subtypes Rare subtypes: NET, P, NET, L, and C subtypes

Figure RL15 | Overview of the experimental design and the number of samples for the genomic, proteomic, and phosphoproteomic analyses, also shown in Fig. 1a in the revised version.

At the phosphoprotein level, a total of 111 samples from 49 DC patients were conducted, and 38,696 phosphosites, corresponding to 9,716 phosphoproteins, were identified (**Figure RL16a, b, also shown in Fig. 1g and Extended Data Fig. 2h**). The phosphosites (n = 12,851), identified in more than 20% samples of every substage, were used in the analysis of our cohort. Specifically, 35,339 (97 samples) and 12,543 (14 samples) phosphosites were quantified in the D and G subtypes, respectively.

In our study, although the volume is limited, the identified number of DC samples was comparable to other cohorts at the phosphoprotein level (**Figure RL16c**), such as the LUAD cohort²³ (Cell, 2020, PMID: 32649877).

Figure RL16 | The identified number of the samples in DC progression at the phosphoprotein level, also shown in Fig. 1g and Extended Data Fig. 2h in the revised version. a, The cumulative number of phosphosites in the D subtype progression. **b,** The cumulative number of phosphoproteins of 111 samples in DC progression. **c,** The phosphosite identification number in the Fudan cohort and the LUAD cohort.

Note: NATs: tumor paired non-cancerous adjacent tissues.

(2) About the integrated proteogenomic analysis.

We thank the reviewer again for the constructive suggestions to improve the quality of our manuscript. In the original version, 36 samples were conducted for phosphoproteomic profiling, while was not used well in further analysis. However, the signal transduction driven by

phosphorylation is notable and critical to understanding the carcinogenesis of DC. Therefore, we added 75 samples from 33 DC patients for phosphoproteomic profiling in the revised version. That indicated a total of 111 samples from 49 DC patients were conducted for phosphoproteomic profiling. Based on the multi-omics data, we performed the integrated proteogenomic analysis. Several examples were shown as follows.

As for the involvement of chromosome 8q (chr8q) gain on MAPK signaling in the infiltration tumor (IFT) phase.

In our study, we found the gain of chromosome 8q (chr8q) was notably observed in the IFT phase but not in the intraepithelial neoplasia (IEN) phase (**Figure RL17a, also shown in Fig. 3b**). To explore the biological functions of the gain of chr8q, we performed *cis* effects of the genes with CNA regions at the protein level, and found 29 genes at chr8q gain perturbation profiles had significantly positive *cis* effects on their cognate proteins (**Figure RL17b, also shown in Fig. 3c**). Of these *cis* effect genes at chr8q gain, 6 genes (*LYN*, *PTK2*, *IMPA1*, *YWHAZ*, *SCRIB*, *RAB2A*) were involved in MAPK signaling, which were also detected in other gastrointestinal cancers²⁶ (Cell, 2018, PMID: 29625048), such as esophageal adenocarcinoma (ESCA), gastric cancer (GC), and colorectal cancer (CRC) (**Figure RL17b, c, also shown in Fig. 3c and Extended Data Fig. 4c**). To assess the effects of these amplified genes, we annotated outliers for the degree of which short hairpin RNA (RNAi)-mediated depletion reduced DC cell lines (AZ521 and HUTU80 cell lines)²⁷ (Cell, 2017, PMID: 28753430). As a result, we found the amplifications of *PTK2*, *PUF60*, *RAB2B*, and *LYN*, had negative effects on the proliferation of duodenal cancer cells, in which *LYN* was the only one gene recorded in cancer associated genes (CAGs)^{28,29} (Cell, 2018, PMID: 29625053; Nature, 2016, PMID: 27251275) (**Figure RL17b, also shown in Fig. 3c**).

LYN, a member of Src kinases, regulates the activation of ERK1/2 signaling³⁰ (Nature, 1996, PMID: 8606776), and promotes the cell proliferation and survival in many diseases including CRCs³¹ (Molecular cell, 2012, PMID: 22731636) and chronic myelogenous leukemia³² (Blood, 2003, PMID: 12509383). In our cohort, the amplification of *LYN* had positive impacts on the proteome-levels of MAPK signaling and cell proliferation (**Figure RL17d, e, also shown in Fig. 3e, f**), evidenced by the adaptors (e.g., CRK, CRKL, etc.), and *LYN*-protein complex (e.g., CBL, PTPN6, SHC1, etc.), which activated the MAPK signaling³¹ (Molecular cell, 2012, PMID: 22731636). In addition, *LYN* was positively associated with PTPN6 at the protein level (Pearson's $R = 0.31$, $p = 7.2E-9$), likely as the proteome-levels of CRKL and SHC1 (Pearson's $R = 0.29$, $p = 9.3E-7$), CRK and MAPK1 (Pearson's $R = 0.48$, $p < 2.2E-16$) (**Figure RL17e, also shown in Fig. 3f**). Furthermore, the phosphoprotein-levels of MAPK signaling (e.g., MAP2K2 S26, MAPK1 Y187, etc.) were overrepresented in the IFT phase (**Figure RL17e, also shown in Fig. 3f**). These findings indicated the impacts of *LYN* amplification on MAPK signaling based on phosphorylation in DC progression.

To further investigate how *LYN* amplification regulated the activation of MAPK signaling in DC, we performed kinase substrate enrichment analysis (KSEA) of phosphoproteome on the *LYN* amplification group. As a result, MAPK signaling (e.g., MAPK1/3/7, RPS6KA3, etc.) related kinases were overrepresented in the *LYN* amplification group, in which MAPK1 showed positive association with *LYN* (Pearson's $R = 0.25$, $p = 5.4E-6$) (**Figure RL17f, g, also shown in Fig. 3g**

and Extended Data Fig. 4e). Based on kinase-substrate regulation network analysis, among of the associated substrates, we found that MAPK Y187 was the only substrate of LYN, and was overrepresented in the *LYN* amplification group (Wilcoxon signed-ranked test, adjust. $p = 3.3E-3$, *LYN* Amp vs. WT ratio = 2.52) (**Figure RL17h, i, also shown in Fig. 3h and Extended Data Fig. 4f**). In addition, the enhanced phosphorylation of MAPK1 (Y187) could activate the substrates of RPS6KA3 (S369, S375, S715, and T365) and TOP2A (S1213) (**Figure RL17j, also shown in Extended Data Fig. 4g**), which is one of the markers in cell proliferation³³ (Nat Rev Cancer. 2006, PMID: 1649106). These results implied the *LYN* amplification had positive impacts on cell proliferation in DC progression. To further validate the assumptions, we analyzed the correlation between LYN and cell proliferation, and found that LYN showed positive association with those markers (e.g., MCM4, MCM6, PCNA, TOP2A, etc.), and was significantly correlated with cell proliferation pathway (Pearson's correlation, $R = 0.33$, $p = 6.6E-10$) (**Figure RL17k, also shown in Fig. 3i**). Together, the chr8q gain functioned in the transmit process from the IEN phase to the IFT phase, in which the amplification of *LYN* enhanced the protein-level of LYN, which activated MAPK signaling and promoted the DC cell proliferation, hinting the potential medical actions of saracatinib³⁴ (J Clin Invest, 2014, PMID: 24316974) in the DC clinic (**Figure RL17l, also shown in Fig. 3j**).

Figure RL17 | The impacts of chr8q gain on MAPK signaling in DC, also shown in Fig. 3 and Extended Data Fig. 4 in the revised version. **a**, Volcano plot showing the *cis*- correlation of the SCNA (x axis) and the associated $-\log_{10}(p)$ value (y axis) on the genes at chr8q gain (Spearman's correlation). **b**, The *cis* SCNA-protein regulations of significantly correlated genes (top) on their corresponding proteins expression (bottom). The right representing relative survival averaged across all available DC cell lines after depletion of the indicated genes by RNAi (Dependency map-supported (<https://depmap.org>) panels) and showing the correlation between the genes and CAGs. The square directed to a subset of patient samples in the D subtype used for WES. **c**, Volcano plot showing the *cis*- correlation of the SCNA (x axis) at chr8q gain and the associated $-\log_{10}(p)$ value

(y axes) on the corresponding proteins of TCGA cohorts (Spearman's correlation). **d**, Venn diagram depicting the amplification of LYN positive associated proteins (top), and the represented signaling pathways (bottom). **e**, Heatmap showing the represented proteins in MAPK signaling at the protein (top) and phosphoprotein (bottom) levels (Pearson's correlation). **f**, The represented kinases in MAPK signaling overrepresented in the *LYN* amplification group. **g**, Scatterplot showing the relationship between \log_{10} LYN and MAPK1 at the protein level (Pearson's correlation). **h**, Kinase-substrate regulation network showing the substates of LYN and MAPK1. **i**, Boxplot showing the expression of MAPK Y187 in the *LYN* amplification group and WT group (Wilcoxon signed-ranked test). **j**, A brief summary showing the regulation of LYN on the downstream substrates. **k**, Scatterplot showing the relationship between LYN and cell proliferation at the protein level (Pearson's correlation). **l**, A brief summary of the impacts of the chr8q gain in the IFT phase of the D subtype.

As for the functions of *DST* mutation on mTOR signaling in the duodenal adenocarcinoma (DAC) stage.

In our cohort, among of the mutations in DC progression, we found that *DST* mutation, detected in the IFT phase and overrepresented in the DAC stage (Fisher's exact test, $p = 2.5E-3$), enhanced the expression of its counterpart protein (Wilcoxon signed-ranked test, adjust. $p = 0.019$, *DST* Mut vs. WT ratio = 3.19), and had negative impacts on the proliferation of DC cell lines (**Figure RL18a, also shown in Fig. 4f**).

Generally, *DST* plays a key role in calcium (Ca^{2+}) signal and increased Ca^{2+} level³⁵ (Autoimmunity, 2002, PMID: 12482196), which is the activator in the transmit process from the phospholipid and diacylglycerol (DAG) to protein kinase C (PKC)³⁶ (Nature medicine, 2004, PMID: 14966518). In our study, we found that PRKCD, one of PKC³⁶ (Nature medicine, 2004, PMID: 14966518), was overrepresented in the *DST* mutation group (Wilcoxon signed-ranked test, adjust. $p = 0.023$, *DST* Mut vs. WT ratio = 1.96), which exhibited positive association with *DST* at the protein level (Pearson's $R = 0.33$, $p = 0.011$) (**Figure RL18b, also shown in Extended Data Fig. 5k**).

To explore the impacts of *DST* mutation on DC progression, we performed GO/KEGG enrichment analysis on the overrepresented proteins in the *DST* mutation group, and found those proteins participated in mTOR signaling (e.g., MTOR, RPS6KA1/3) (**Figure RL18c, d, also shown in Fig. 4g and Extended Data Fig. 5l**). At the phosphoprotein level, 1,608 phosphosites mapped to 885 phosphoproteins showed greater changes than corresponding protein abundance (Wilcoxon signed-ranked test, adjust. $p < 0.05$, *DST* Mut vs. WT ratio ≥ 2), and were significantly enriched in mTOR signaling (e.g., MTOR S1262, DEPTOR S265, etc.) (**Figure RL18e, f, also shown in Fig. 4h and Extended Data Fig. 5m**). These findings indicated that *DST* mutation had positive impacts of mTOR signaling based on phosphorylation.

To investigate how *DST* mutation regulated phosphorylation and enhanced mTOR signaling, we performed KSEA of the phosphoproteome of the *DST* mutation group and WT group. As a result, the dominant kinases were found to be activated in the *DST* mutation group, including PDK1, PRKG1, MAP2K7, PRKCI, PRKDC, and ABL1, in which the protein-levels of PDK1 and PRKDC

were overrepresented in the *DST* mutation group (Wilcoxon signed-ranked test, adjust. $p < 0.05$) (Figure RL18g, h, also shown in Fig. 4i and Extended Data Fig. 5n).

PDK1, a kinase upstream of mTOR, binds to inositol triphosphate (IP3) and activated PKB/AKT/mTOR signaling³⁷ (Science, 2005, PMID: 15728470). The kinase-substrate regulation network analysis revealed that the elevated substrates of PDK1 in our cohort, were involved in mTOR signaling, such as AKT1 S124, PRS6KA3 S415, MTOR S1261, etc. (Figure RL18i, also shown in Fig. 4j). These findings further implied the enhancement of mTOR signaling in the *DST* mutation group. PRKDC, one of the catalytic subunits of DNA-dependent protein kinase, of which the Sh2 domain can bind to the ser/thr sites and thus activates ser/thr kinases^{38,39} (Cell, 1995, PMID: 7671312; Cell, 2002, PMID: 11955432). In our study, PRKDC showed significant association with PRKCD (Pearson's $R = 0.35$, $p = 7.4E-3$), and activated the substrate AKT1 (S124) (Figure RL18i, j, also shown in Fig. 4j, k). The integrated findings of the kinase-substrate analysis (PDK1 and PRKDC) suggested the overrepresentation of the mTOR signaling in the *DST* mutation group. Taken together, these results indicated that the overrepresented *DST* protein in the *DST* mutation group, enhanced the protein-levels of the kinases (PDK1 and PRKDC), and thus activated mTOR signaling in the DAC stage (Figure RL18k, also shown in Fig. 4l), providing a reference database for the corresponding personalized medicine of DAC.

Figure RL18 | The impacts of *DST* mutation in DAC stage, also shown in Fig. 4 and Extended Data Fig. 5. **a**, The Venn diagram (left) and heatmap (right) showing the significantly differential mutations between the D and G subtypes (Fisher's exact test) and the protein levels of *DST* and *PRKDC* (Wilcoxon signed-rank test). The square directed to a subset of patient samples in the D subtype used for WES. **b**, Scatterplot showing the relationship between \log_{10} *DST* and \log_{10} *PRKDC*

at the protein level (Pearson's correlation). **c**, The represented pathways in the *DST* Mut group at the protein level. **d**, Heatmap showing the *DST* mutation positive associated proteins in mTOR signaling. **e**, Fold changes of proteins and phosphosites, and their correlations in the *DST* mutation group and WT group. Red dots: phosphosites are greater than twofold changes in the *DST* mutation group vs. WT group, and changes of phosphosites abundance are greater than changes of their corresponding protein abundance. **f**, The represented pathways in the *DST* Mut group at the phosphoprotein level. **g**, KSEA analysis of the kinase activity in the *DST* mutation group and WT group. **h**, Volcano plot showing the impacts of *DST* mutation on the protein-levels of kinases. **i**, Kinase-substrate interaction network in mTOR signaling. **j**, Scatterplot showing the relationship between \log_{10} PRKCD and \log_{10} PRKDC at the protein level (Pearson's correlation). **k**, a brief summary of the impacts of *DST* mutation in the DAC stage of the D subtype. **** $p < 1.0E-4$, *** $p < 1.0E-3$, ** $p < 1.0E-2$, * $p < 0.05$, ns. > 0.05 .

(3) About the shortening the text.

We thank the reviewer for the careful read and valuable suggestions to present the statement of our manuscript clearly and concisely. In the revised version, we streamlined the manuscript, shorting the description more general observations, and removed the redundant information in the revised manuscript, such as the redundant p values, repetitive data presentation, complex data mapping, and so on.

As for tuning down the statement.

In addition, to make our manuscript clearer and broader, we tuned down our statement in the revised manuscript, such as the description of the conclusion at the “**Abstract**”. Please see more details in the revised manuscript with red text.

As for the potential drug targets, we found alanyl-tRNA synthetase 1 (AARS1) was gradually enhanced in DC progression, and played a key role in immune clusters. Therefore, we designed a series of experiments including drug inhibition to further validate the functions of AARS1 *in vivo* and *in vitro*. However, for other drug inhibitions, such as capivasertib to the D subtype, olaparib to anti-PARPs, etc., needs to be further investigated. Thus, we shorted the related description and tuned down the statement.

As for removing the redundant information.

In the revised version, to represent the data and manuscript logically and readably, we streamlined the manuscript and re-arranged the figures and supplementary figures in the revised version. In addition, we removed the redundant and unrelated information. The examples were shown as follows.

For example, **Fig. 2 and Extended Data Fig. 3** showed two major findings: 1) the Tobacco signature, related to the clinic information of DC patients, was overrepresented in the D subtype and might induce NNK-O-glucuronic acid secretion via the overrepresentation of GAGs; 2) the

DNA repair signature were featured with TMB, and drove the duodenal epithelial neoplasia, in which ARS-PARPs signaling was overrepresented. In the revised version, we added the MSI score, which were overrepresented in the DNA repair signature (**Fig. 2h**). In addition, we removed the unclear **Extended Data Fig. 3b, c**. Owing to **Extended Data Fig. 3a**, showing the correlation among of the DC subtypes, we moved the **Extended Data Fig. 3a** to **Extended Data Fig. 5** in the revised version, which revealed that divergency of DC subtypes was derived from their origin normal tissues and diverse carcinogenesis tracks.

As for other figures, to present the figures readably and concisely, we integrated the compatible figures and removed the complex information. For example, the **Fig. 4e** (revised version) was integrated from the **Fig. 4e and Extended Data Fig. 5f** of the original version. Besides, we removed the redundant figures, such as **Extended Data Fig. 2f and Extended Data Fig. 5c, e**. In addition, we removed the un-visualized color-codes presentation of the stages/subtypes and other clinic information in **Fig. 1a, Fig. 1b, Fig. 2a, Fig. 3f, Fig. 5a**, etc. Please see the details in the revised version.

Q4. Overall, there are lots of claims about the therapeutic potential of DC using their analysis results which are highly touted and should be significantly tuned down. Most of their results (except for the AARS1-PARP1 part) are just based on omics data interpretation with limited sample cohort and lack of supporting evidence. Just as two examples “.. suggested that the combination of capivasertib and everolimus/ sirolimus/ temsirolimus would be potential strategy in the D subtype” (Page 13), or “This observation may indicate the medicative actions of afatinib and lapatinib to anti-ERBB signaling, dinaciclib and pemetrexed, to anti-cell proliferation, and olaparib to anti-PARPs, etc., in the clinic of the Hot tumor cluster” (Page 15). Their study only highlighted the need of further biological investigation which is far away from clinical applications. Also, conclusions in the Abstract are too strong and need to be tone down, e.g., “Proteogenomics revealed the chromosome 8q gain was the key event driving the transmit from the intraepithelial neoplasia phase to the infiltration tumor phase via RET signaling”.

Response: We appreciate the reviewer for the comments, and we agree with the reviewer’s points that the therapeutic potential of DC based on proteogenomic analysis results should be significantly tuned down. In the revised version, to make our statement more accurate, we tuned down our statement and removed related statements in the manuscript.

As for the potential drug target.

In our cohort, proteogenomic analysis showed that the intermediate of AARS1, growingly enhanced in the D subtype progression, may be effective in the cancer-associated signaling pathways, implying the necessary of exploiting the utility of AARS1 in DC therapy. Therefore, we designed a series of experiments and drug inhibition, and found that increased AARS1 promoted cancer cell proliferation, inhibited cell apoptosis and blocking lysine-alanylation (K-Ala) activated PARP1, and suppressed tumor growth in *vivo* and *vitro*.

However, the other potential drug targets, such as afatinib and lapatinib to anti-ERBB signaling, lack of evidence and needs to be further investigated. In the revised manuscript, to improve our statement accurately and description clearly, we removed the related statement of “This observation may indicate the medicative actions of afatinib and lapatinib to anti-ERBB signaling, dinaciclib and pemtetrexed to anti-cell proliferation, and olaparib to anti-PARPs, etc., in the clinic of the *Hot tumor cluster*”.

As for the Abstract.

We tuned down the statement of conclusion in the “**Abstract**” with red text (**Line 2–6, Page 2**). Please see more details in the revised manuscript with red text.

Q5. The author spent lots of effort analyzing the Tobacco signature in the D type. However, is tobacco usage a significant risk factor in DC, especially D? Please cite the reference. Can they use their clinical data to confirm this finding? Otherwise, what is the rationale of looking at Tobacco signature?

Response: We thank the reviewer sincerely for the careful read and thoughtful comments. According to the reviewer’s comments, we divided the responses into two parts to answer: (1) whether the clinical data of DC patients is rationale at the Tobacco signature; (2) whether tobacco usage is a significant risk factor in DC, especially in the D subtype.

(1) Whether the clinical data of DC patients is rationale at the Tobacco signature.

In our cohort, the defined Tobacco signature was based on the WES data and the clinic history of patients using chewing tobacco.

Firstly, to investigate the specific etiological factors that may contribute to the mutagenesis of DC, we performed mutational analysis and found that Tobacco signature was overrepresented in the D subtype (**Figure RL19a, also shown in Fig. 2a**). Secondly, to explore whether tobacco usage is associated with the Tobacco signature, we analyzed the clinic information of DC patients, and found that, more patients (60% vs 18.8%, Fisher’s exact test, $p = 1.4E-3$) with smoking habit were observed in the Tobacco signature group with higher SBS29 (Tobacco signature) score (Wilcoxon signed-rank test, $p = 0.024$) (**Figure RL19b, c, also shown in Fig. 2c and Extended Data Fig. 3d**). These findings indicated that the clinical data of the DC patients is rationale at the Tobacco signature.

Figure RL19 | The association between the Tobacco signature and the clinic information of DC patients, also shown in Fig. 2a, c, and Extended Data Fig. 3d in the revised version. a, The

major somatic SNVs signature in DC progression. **b**, Histogram showing the proportion of the samples with the habit of smoking in the Tobacco signature and DNA repair panel (Fisher's exact test). **c**, Barplots presenting the SBS29 contribution in the patients with the habit of smoking or not (Wilcoxon signed-rank test). **** $p < 1.0E-4$, *** $p < 1.0E-3$, ** $p < 1.0E-2$, * $p < 0.05$, ns. > 0.05 .

(2) Whether tobacco usage is a significant risk factor in DC, especially in the D subtype.

In our review of the literatures, there is no literature directly suggest tobacco usage increased the risk of duodenal adenocarcinoma (D subtype in our cohort). However, several published studies pointed that smoking is a known risk factor in a number of gastrointestinal (GI) diseases, including duodenal ulcer⁴⁰ (Microbiome, 2018, PMID: 30157953) and GI cancer⁴¹ (JAMA, 2008, PMID: 19088354). In addition, tobacco usage has important implications regarding gut environment and various small intestine disorders, including functional dyspepsia⁴² (Gut, 2017, PMID: 27489239), irritable bowel syndrome⁴³ (Gastroenterology, 2017, PMID: 27725146), coeliac disease⁴⁴ (Clin Gastroenterol Hepatol, 2005, PMID: 16234024), and duodenal ulcer disease⁴⁵ (N Engl J Med, 1956, PMID: 13334789).

Furthermore, Buchthal, J. et al.⁴⁶ (Eur J Clin Pharmacol, 1995, PMID: 7720765) demonstrated Cytochromes P450 families (CYPs) induction by constituents of cigarette smoke in the human duodenum. In our cohort, CYPs were overrepresented in the Tobacco signature at the protein level (**Figure RL20, also shown in Fig. 2d**). Besides, we found more patients (60% vs 18.8%), especially the D subtype patients, with smoking habit were observed in the Tobacco signature group (**Figure RL19b, also shown in Fig. 2c**). Integration with the findings in our cohort and the impacts of smoking on gastrointestinal cancer, including esophageal cancer, gastric cancer, colorectal cancer, etc., indicated that smoking or tobacco usage might be a risk for DC, especially for the D subtype.

Figure RL20 | The represented proteins from CYPs and the predominant pathways of the Tobacco signature (SBS29 signature) and DNA repair panel, also shown in Fig. 2d in the revised version.

However, to explore the correlation between tobacco usage and DC, more evidences are needed from other cohorts or epidemiologic studies with more DC patients for further analysis. In addition,

more samples from DC patients with diverse clinic features, are required to illustrate molecular mechanisms and provide corresponding potential clinic therapy for DC, especially for the two major (D and G) subtypes. We had added this section in the “**Discussion**” of the revised manuscript with red text (Line 22–28, Page 21, Line 1–5, Page 22).

Q6. AARS1 and PAPR1. The author suggests that the interaction between AARS1 and PARP1 reduced the PARP1 activity through K-Ala. However, what is the abundance and activity status of PAPR1 in the DC proteome and phosphoproteome data? How much of PARP1 activity is reduced by interacting with AARS1? Is PARP1 the top 1 differential protein in the long list of 652 proteins identified as AARS1 interactors? Are there other proteins of these 652 proteins showing different expressions between in DC or between its subtypes? To summarize, although this additional experiment seems to be nicely done, it currently lacks coherency to the “proteogenomic” data.

Response: We appreciate the insightful comments. According to the reviewer’s comments, we designed a series of experiments and divided the responses into four parts to answer: (1) about the abundance and activity status of PAPR1 in the DC proteomic and phosphoproteomic data; (2) about the reduced PARP1 activity by interacting with AARS1; (3) whether PARP1 is the top 1 differential protein in the long list of 652 proteins identified as AARS1 interactors; (4) about the expression of 652 proteins in DC.

(1) About the abundance and activity status of PAPR1 in the DC proteomic and phosphoproteomic data.

In our cohort, the expression PARP1 was increased in the D subtype (Kruskal-Wallis test, adjust. $p = 3.1E-8$) progression at the protein level (**Figure RL21**). However, the phosphorylation of PARP1 was not prevalent in DC progression (2/111). These findings indicated the function of PARP1 in DC was not based on phosphorylation.

Figure RL21 | The expression of PARP 1 in DC at the protein (left) and phosphoprotein (right) levels. ** $p < 1.0E-4$, *** $p < 1.0E-3$, ** $p < 1.0E-2$, * $p < 0.05$, ns. > 0.05 .**

(2) About the reduced PARP1 activity by interacting with AARS1.

To investigate the potential non-translational role of AARS1, we used tandem affinity purification to identify AARS1-interacting proteins in Hutu80 cells. As a result, a total of 652 proteins were detected, in which PARP1 showed significantly positive interaction with AARS1 as a high score and abundant peptide coverage. To explore the impacts of AARS1 on PARP1 activity, we designed

a PARP1 enzymatic activity assay, and found that overexpression of AARS1 decreased PARP1 specific activity (t-test, $p = 5.5E-6$ for Hutu80 and $2.9E-4$ for WDC-1) in both Hutu80 cells and WDC-1 cells (**Figure RL22a, also shown in Fig. 6i**). In addition, the overexpression of other ARS families (e.g., TARS1 and GARS1) would not decrease PARP1 activity (**Figure RL22a, also shown in Fig. 6i**). Therefore, among of ARS families, PARP1 interacted with AARS1, reducing the reduction of PARP1 activity.

To further evaluate how much of the PARP1 was reduced by interacting with AARS1, we performed PARP1 activity colorimetric assay *in vitro*, by adding purified AARS1 into the reaction mix. As a result, we found that AARS1 led to an approximately 20% reduction of PARP1 activity (t-test, $p = 2.6E-3$) (**Figure RL22b, also shown in Fig. 6j**). In addition, we added the description of the experiments (**Line 9–12, Page 18**), and the methods (**Line 27, page 39, and Line 1–8, page 40**) in the revised manuscript with red text.

Figure RL22 | The effects of AARS1 on the activity of PARP1, also shown in Fig. 6i, j in the revised version. a, The effects of overexpression of AARS1, TARS1, and GARS1 on PARP1 enzymatic activity (t-test). **b**, The relative reduction of PARP1 activity by interacting with AARS1 (t-test). **** $p < 1.0E-4$, *** $p < 1.0E-3$, ** $p < 1.0E-2$, * $p < 0.05$, ns. > 0.05 .

(3) Whether PARP1 is the top 1 differential protein in the long list of 652 proteins identified as AARS1 interactors.

In our cohort, to explore the potential role of PARP1, we used tandem affinity purification to identify AARS1-interacting proteins in Hutu80 cells, and detected a total of 652 differential proteins. Gene Ontology (GO)-enrichment analysis showed that DNA damage repair-related pathways were enriched (**Figure RL23a, also shown in Fig. 6f**), which was the dominant pathway regulated by AARS1 in our cohort. Among the 652 proteins, TUBA1B, TUBA1A, TUBB2A, PARP1, and FASN, were the top five ranked proteins, while the top three ranked proteins (TUBB1A, TUBB1B, TUBB2A) were tubulins and had no relevance to the DNA damage repair-related pathways (**Figure RL23b, also shown in Extended Data Fig. 7f**). Therefore, PARP1, one of poly-ADP-ribosyltransferases that mediates poly-ADP-ribosylation of proteins and plays a key role in DNA repair⁴⁷ (Molecular cell, 2015, PMID: 26344098), was listed at the fourth top protein and was the prior preferential target that was interacted with AARS1, and was validated in our cohort.

Figure RL23 | AARS1-interacting proteins and the related pathways, also shown in Fig. 6f and Extended Data Fig. 7f in the revised version. a, GO enrichment analysis of AARS1-interacting proteins in biological processes. These proteins were identified by a proteomics survey using tandem affinity purification. **b,** Scatterplot presenting the proteins involved in DNA damage repair exhibit strong interactions with AARS1.

(4) About the expression of 652 proteins in DC.

We thank the reviewer for the constructive comments. Among of the 652 proteins, we found significantly higher protein-levels of AARS1 (Wilcoxon signed-rank test, adjust. $p = 4.0E-5$, D vs. G ratio = 2.09) and PARP1 (Wilcoxon signed-rank test, adjust. $p = 0.045$, D vs. G ratio = 1.75) were detected in the D subtype (**Figure RL24a**).

Furthermore, most of 652 proteins were significantly overrepresented in the D subtype ($n = 517$, $p < 0.05$) (**Figure RL24b**, also shown in **Extended Data Fig. 7d**). GO-enrichment analysis revealed that those proteins were overrepresented in DNA repair (e.g., MSH2, POLD, and cell cycle (e.g., ACTR1A, DSTN2) (**Figure RL24c, d**, also shown in **Extended Data Fig. 7e**), which was consistent with the dominant pathways of the D subtype. These results further indicated the functions of AARS1 combined with PARP1 in the D subtype.

Figure RL24 | The expression of 652 proteins in DC, also shown in Extended Data Fig. 7 in the

revised version. a, Boxplot showing the expression of AARS1 and PARP1 in the D and G subtypes (Wilcoxon signed-ranked test). **b**, Venn diagram showing the overrepresented proteins from 652 proteins in the D and G subtypes. **c**, Gene Ontology enrichment analysis displaying the represented pathways of the overrepresented proteins from 652 protein in the D subtype. **d**, Boxplot showing the expression of the represented proteins of the dominant pathways. **** $p < 1.0E-4$, *** $p < 1.0E-3$, ** $p < 1.0E-2$, * $p < 0.05$, ns. > 0.05 .

Minor issues:

Q5. Page 3. Paragraph 2. “Hence DC was not included in TCGA”. How does the author know that DC is not on the CPTAC to-do list? we do not think the inclusion or exclusion of DC in TCGA is relevant for the present work.

Response: We thank the reviewer for the constructive suggestions, and agree with reviewer’s point that the inclusion or exclusion of DC in TCGA is not relevant for the present work.

We searched TCGA data portal, CPTAC data portal (<https://proteomics.cancer.gov/programs/cptac>) and NCBI database (<https://www.ncbi.nlm.nih.gov/>) carefully, and found that DC was not recorded so far. In the revised version, we had deleted the description of “Hence DC was not included in TCGA” in the revised manuscript.

Q6. Page 5, Paragraph 2. All the percentages here are based on very small numbers of patients (Figure 2c). The author should write down the real case number /total case number here as well.

Response: Thanks for the comments. We are sorry for no description of the case number in Fig. 2 of the original version. In the revised version, we added the real case number/total case number in **Fig. 2b** of the revised version, added the description in the revised manuscript with red text (**Line 8–11, Page 7**).

To further validate the results of our study, apart from 406 samples in the original version, we added another 16 duodenal adenocarcinoma (DAC) stage samples with paired normal tissues from 16 advanced-stage DC patients for whole-exome sequencing (WES), which were also conducted for proteomic profiling. Therefore, a total of 120 samples from 47 DC patients were conducted for WES in the revised version.

After adding 32 samples (16 DAC stage + 16 paired normal tissues samples) for WES, we found that 20 of 68 samples were grouped into SBS29 enriched group (Tobacco signature group), which was detected in the non-tumor (NT) stage and last till to the DAC stage of the D subtype (**Figure RL25a, also shown in Fig. 2b**). Consistently, more patients (60% vs. 18.8%, Fisher’s exact test, $p = 1.4E-3$) with smoking habit were observed in the Tobacco signature group with higher SBS29 score (Wilcoxon signed-rank test, $p = 0.024$) (**Figure RL25b, c, also shown in Fig. 2c and Extended Data Fig. 3d**). These findings reflected that tobacco usage might be a risk factor for DC, especially for the D subtype.

Compared with the DNA repair panel (48/68), the tumor mutation burden (TMB) of the Tobacco signature (20/68) was significantly lower (Wilcoxon signed-rank test, $p < 1.0E-4$), and showed negative association with the mutation frequency (Pearson's $R = -0.44$, $p = 1.7E-4$) (**Figure RL25d**, also shown in **Fig. 2g** and **Extended Data Fig. 3f**). We added this section in the revised manuscript with red text (**Line 8–15, Page 7**).

Figure RL25 | The impacts of the tobacco signature in DC, also shown in Fig. 2 and Extended Data Fig. 3 in the revised version. a, Heatmap showing the clinic information (top), correlated mutations (middle), and the SBS29 (Tobacco) signature contribution (bottom) in the D and G subtypes. The square directed to a subset of patient samples used for WES. **b**, Histogram showing the proportion of the samples with the habit of smoking in the Tobacco signature and DNA repair panel (Fisher's exact test). **c**, Bar plot presenting the SBS29 contribution in the patients with the habit of smoking or not (Wilcoxon rank-signed test). **d**, Boxplot presenting the MSI score in the Tobacco signature and DNA repair panel (Wilcoxon signed-ranked test) (left) and scatterplot showing the relationship between Tobacco signature and mutation burden (right) (Pearson's correlation). **** $p < 1.0E-4$, *** $p < 1.0E-3$, ** $p < 1.0E-2$, * $p < 0.05$, ns. > 0.05 .

Q7. Page 6. Paragraph 1. The FOT spanned eight orders of magnitude- But this does not mean anything as it is not about the protein copies.

Response: Thanks for the comments. In our cohort, protein abundance was calculated by intensity-based absolute quantification (iBAQ)^{48,49} (Nature, 2011, PMID: 21593866; Nature, 2013, PMID: 23407496) and then normalized as a fraction of the total (FOT). In our cohort, the FOT spanned eight orders of magnitudes, indicating the highly dynamic reference proteome, did not mean the protein copies. In the revised manuscript, to improve our statement accurately and description clearly, we removed the corresponding figure and statements of “The reference proteome was highly dynamic based on the protein intensity, which spanned over eight orders of magnitude”.

Q8. Page 6. Paragraph 1. Some blood proteins are in the Top 10 abundant list. The authors are requested to perform additional analysis to exclude the possibility of blood contamination or to gauge the corresponding impact.

Response: Thanks for the comments. In our cohort, all samples were separately dissected from the

formalin-fixed, paraffin-embedded (FFPE) slides, and were evaluated by two or three experienced gastrointestinal pathologists, who would then mark the hematoxylin and eosin (H&E)-stained sections (3 μm-thick) of FFPE slides according to the proportion of tumor cells. The tumor purity was over 95%, indicating the high quality of all samples of our cohort. Therefore, the strict criterion of preparation of FFPE samples with high quality guaranteed no blood contamination in the collected samples of our cohort.

To further validate the results, we performed immunohistochemistry (IHC) analysis to detect the marker proteins of blood, including HBA1 and HBD, in the samples of our cohort. As a result, we found the overrepresentation of HBA1 and HBD was detected in the normal/tumor cells (**Figure RL26**), indicating those proteins were expressed in normal/tumor tissues instead of blood contamination in the samples of our cohort. In the revised manuscript, to improve our statement accurately and description clearly, we removed the top 10 abundant list related statement of “The top 10 proteins extensively expressed in the D subtype were HBB, HBA1, HBA2, ALB, HBD, etc.”.

Figure RL26 | IHC analysis of the expression of HBA1 and HBD on the DC sample tissues.

In addition, the findings of the some blood proteins (e.g., HBA1, HBD, etc.) were also highly expressed (top 10) in other published studies, such as human stomach cancer²⁵ (Nature Communications, 2019, PMID: 30604760) and lung adenocarcinoma²³ (Cell, 2020, PMID: 32649877).

Q9. Page 6. Paragraph 2: how Muc19 2q is different from Muc19? Please add explanation.

Response: Thanks for the comments. We are sorry for the mistake of Muc19 2q, which is actually the *MUC19* 12q (**Figure RL27, also shown in Extended data Fig. 5d**). At the Page 6, the amplification of *MUC19* 12q was an arm event of the N and C subtypes, which was different from the mutation of *MUC19*.

Figure RL27 | The specific characterization and mutational signature of two major and five rare subtypes of DC, also shown in Extended Data Fig. 5d in the revised version. a, Hierarchical clustering analysis of 2 major and 5 rare subtypes of DC. b, Clonal evolutionary tree of the D/P/G/N/C subtypes of DC. The prominent SCNAs were shown in the box.

Q10. Page 7. Paragraph 2: Please define SBS29 before the usage.

Response: Thanks for the reminder and we apologize for not presenting the definition of the SBS29 signature in the original version. In the revised version, we had defined SBS29 before the usage. Please see the details in the revised manuscript with red text (Line 7–8, Page 7).

Q11. Please describe how the 400 samples are injected (if the sequence is randomized or not).

Response: Thanks for the constructive comments and reminder, and all samples were randomly injected for proteomic analysis in our cohort. Details were shown as follows.

In our study, 438 samples were randomly assigned number and then performed proteomic analysis. The HEK293T cell (National Infrastructure Cell Line Resource) lysates were measured as the standards every 15 samples injection to assess the stability of the performance of mass spectrometry. The Spearman’s correlation coefficient of HEK293T cells was 0.93 (Figure RL28a, also shown in Extended Data Fig. 2c), presenting the stability of the MS data. In addition, we randomly selected 200 samples, which were conducted on proteomic profiling at different time, and then performed principal component analysis (PCA). As a result, we found no clear boundary between the samples injections at different profiling time (Figure RL28b). On the contrary, there were obvious separation trend of samples along with the different substages in D and G subtypes progression. Therefore, these findings indicated there is no batch effect in the samples for proteomic profiling. In the revised manuscript, we added this section in the “Methods” with red text (Line 12–14, Page 32).

Figure RL28 | The samples injections in our cohort, also shown in Extended Data Fig. 2c in the revised version. a, Spearman's correlation of HEK293T cells. **b,** PCA on 200 samples profiled at different time. The similar color representing the one same stage of DC. The differently hollow shape showing the injections for proteomic profiling.

Q12. Page 11 Paragraph 3 and other places: “temporal molecular models”. The authors did not perform temporal analysis at all. They should consider using “staging” rather than “temporal.”

Response: Thanks for your insightful suggestions and we apologize for the errors. We read the manuscript thoroughly and corrected “temporal” as “staging” in the revised manuscript with red text. Please see the details in the revised manuscript.

Q13. Methods: Please add the correct reference for the PARP1 activity assay.

Response: Thanks for the comments, and we added a reference (Journal of Biological Chemistry, 2012, PMID: 22736760) for the PARP1 activity assay in the section of “PARP1 activity assay”, which was listed as follow. Please see the details in the “**Methods**” of the revised manuscript with red text (Line 27, Page 39).

Reference:

Aya Masaoka, Natalie R. Gassman, Padmini S. Kedar, Rajendra Prasad, Esther W. Hou, Julie K. Horton, Michael Bustin, Samuel H. Wilson. HMG1 Protein Regulates Poly (ADP-ribose) Polymerase-1 (PARP-1) Self-PARylation in Mouse Fibroblasts*, Journal of Biological Chemistry, Volume 287, Issue 33, 2012.

Q14. Methods: Please add the correct reference for the Alanylation and de-alanylation reaction.

Response: Thanks for the comments, and we added a reference (Cell metabolism, 2018, PMID: 29198988) for the alanylation and de-alanylation assay in the “**Methods**” of the revised manuscript with red text (Line 7–8, Page 41). The reference was listed as follows.

Reference:

He XD, Gong W, Zhang JN, Nie J, Yao CF, Guo FS, Lin Y, Wu XH, Li F, Li J, Sun WC, Wang ED, An YP, Tang HR, Yan GQ, Yang PY, Wei Y, Mao YZ, Lin PC, Zhao JY, Xu Y, Xu W, Zhao SM. Sensing and Transmitting Intracellular Amino Acid Signals through Reversible Lysine Aminoacylations. *Cell Metab.* 2018 Jan 9;27(1):151-166.e6.

Reviewer #3 (Remarks to the Author): Expert in small intestine cancers

The authors should be congratulated on their Herculean effort to perform a very large genomic, proteomic, and phosphoproteomic analysis. Some of results appear to be significant to include chromosome 8 gain in the transition from intraepithelial neoplasia to invasive cancer and the relationship between DNA repair defects, upregulation of AARS1 and subsequent effects on PARP activation and reduction in apoptosis.

Response: We appreciate the reviewer for the constructive and professional comments. We have revised the manuscript according to the comments, especially focusing on the points as: (1) about the cases for whole-exome sequencing (WES) and phosphoproteomic profiling; (2) whether a few documented cases associated with familial adenomatous polyposis (FAP) and hereditary nonpolyposis colorectal cancer (HNPCC), and about the molecular characterization of the cases of FAP and HNPCC; (3) about the links between clinical data and the Tobacco signature; (4) about the association between DNA repair signature and MSI; (5) about shorting the text and data presentation. We have addressed all the points raised by the reviewer, and the point-to-point responses were as follows.

However, I have several concerns in regard to the paper.

Q1. While the title and summary suggest this is a comprehensive genomic and proteomic analysis on 140 cases of duodenal neoplasia, genomic analysis was only available from 31 cases with 88 samples. Thus, genomic information is performed on 22% of cases, and not sure what is meant by 88 samples. Multiple tumors were analyzed?

Response: Thanks for the comments. We are sorry for explaining the samples for whole-exome sequencing (WES) unclearly. We now explain it as follows.

As for the 140 cases for proteomic analysis, while 31 cases for genomic analysis.

In our cohort (original version), we collected 406 trace samples from 140 DC patients (original version), covering 2 major (D and G subtypes) and 5 rare (NET, P, L, N, and C subtypes) pathological subtypes of DC. According to the 4th and 5th edition World Health Organization Classification of Digestive system, the D and G subtypes were divided into 17 and 6 substages, respectively. Therefore, we presented proteomic profiling of 406 samples from 140 DC patients, of which 88 samples from 31 DC patients were adequate for whole-exome sequencing (WES).

As for the meaning of 88 samples from 31 cases.

In our cohort, one early-stage DC case has more than one substage samples in DC progression. For example, No. 086 DC patient had 5 substage samples, including normal duodenal epithelial (1), tubulovillous adenoma with mild atypia (2_3), tubulovillous adenoma with middle-severe atypia (3_3), tubulovillous adenoma with severe atypia (3_4), and minimally invasion adenocarcinoma (MIA, 4), in which 4 samples (substages 1, 2_3, 3_3 and 3_4) were conducted for WES. Therefore, 88 samples from 31 cases (original version) could be collected and conducted for WES in our cohort. The advantages of our samples in a pathological region resolved mode provided the chances to portray molecular profiles of DC in a stage resolved mode.

As for the findings based on the multiple tumors for WES.

In our cohort, multiple tumors in DC progression were collected and were conducted for WES, allowing us to explore the key events during the carcinogenesis of DC. For example, based on the WES data, we found the number of the novel mutations was peaked at the high-grade intraepithelial neoplasia (HGIN) stage, indicating the significant events in DC progression. In addition, proteogenomics revealed that *DST*, highly mutated in the duodenal adenocarcinoma (DAC) stage, enhanced mTOR signaling.

In the revised version, to further validate the result in our study, apart from 406 samples, we added 16 DAC stage samples with paired normal tissues samples from 16 D subtype patients for whole-exome sequencing (WES), which were also conducted for proteomic profiling. That indicated the 32 samples (16 DAC stage + 16 paired normal tissues samples) were both added for WES and proteomic profiling (Table RL5). Therefore, a total of 438 samples from 156 DC patients were used to perform a comprehensive proteogenomic analysis in our study, in which 120 samples were conducted for WES (Figure RL29, also shown in Fig. 1a).

Table RL5 | The number of the samples for genome (WES) and proteome in the Fudan cohort.

Omics	Original version			Revised version		
	Total	Major subtypes	Rare subtypes	Total	Major subtypes	Rare subtypes
Genome	88	82	6	120	114	6
Proteome	406	390	16	438	422	16

Major subtypes: D and G subtypes Rare subtypes: NET, P, NET, L, and C subtypes

Figure RL29 | Overview of the experimental design and the number of samples for the genomic, proteomic, and phosphoproteomic analyses, also shown in Fig. 1a in the revised version.

In the revised version, we processed the Mutation Annotation Format (MAF) file and copy number alterations (CNAs) in DC progression, and found consistent results compared to the original version.

After adding 32 samples (16 DAC stage + 16 paired normal tissues samples) for WES, more solid results were observed. Several examples were listed as follows.

As for the mutational signatures.

After adding 32 samples for WES in the revised version, higher proportion (25.0% vs. 6.7%) of SBS29 signature (Tobacco signature) was detected in the DAC stage of the D subtype (**Figure RL30a**). In addition, the enrichment of SBS29 contribution (Wilcoxon signed-ranked test, Tobacco signature vs. DNA repair panel ratio = 3.49, $p < 1.0E-4$) was more significantly overrepresented in the Tobacco signature in the revised version (**Figure RL30b**). Therefore, these findings further indicated the Tobacco signature was featured in the D subtype, and was detected as early as in the NT stage and last till to the DAC stage.

Figure RL30 | Tobacco signature in DC. **a**, Venn diagram depicting the stage proportion of the SBS29 signature in the original version (left) and revised version (right). **b**, Boxplot showing the SBS29 contribution in the two major signatures of DC in the original version (left) and revised version (right). **** $p < 1.0E-4$, *** $p < 1.0E-3$, ** $p < 1.0E-2$, * $p < 0.05$, ns. > 0.05 .

As for the somatic CNAs.

After adding 32 samples (16 DAC stage + 16 paired normal tissues samples) for WES, more arm events were detected in DC progression, in which the gain of chromosome 8q gain (chr8q) was overrepresented in the infiltration tumor (IFT) phase (**Figure RL31a, also shown in Fig. 3a**). In the revised version, we found more samples with chr8q gain were detected in the IFT phase (**Figure RL31b, also shown in Fig. 3c**). At the chr8q gain, we found the amplification of *LYN* played a key role in MAPK signaling. In the revised version, the amplification of *LYN* showed more significant *cis*-effects on its corresponding protein level (**Figure RL31c**). These findings further indicated the functions of *LYN* amplification at the chr8q gain, on MAPK signaling.

Figure RL31 | The arm events in DC progression, also shown in Fig. 3a, c in the revised version.
a, Profiling of absolute CNAs observed in DC. The square directed to a subset of patient samples used for WES. **b**, Heatmap showing the distribution of the chr8q gain in the D subtype. **c**, Volcano plot showing the *cis* SCNA-protein regulations of *LYN* amplification (Spearman's correlation). **** $p < 1.0E-4$, *** $p < 1.0E-3$, ** $p < 1.0E-2$, * $p < 0.05$, ns. > 0.05 .

As for the impacts of *DST* mutation.

After adding 32 samples for WES, we found consistent results and conclusions compared with the original version. Specifically, we found that more samples with *DST* mutation were detected in the DAC stage in the revised version (**Figure RL32a, also shown in Fig. 4f**). In addition, *DST* mutation significantly enhanced the expression of *DST* at the protein level (Wilcoxon signed-ranked test, adjust. $p = 0.027$, *DST* Mut vs. WT ratio = 3.19) (**Figure RL32a, also shown in Fig. 4f**).

Generally, *DST* plays a key role in calcium (Ca^{2+}) signal and increased Ca^{2+} level³⁵ (Autoimmunity, 2002, PMID: 12482196), which is the activator in the transmit process from the phospholipid and diacylglycerol (DAG) to protein kinase C (PKC)³⁶ (Nature medicine, 2004, PMID: 14966518). In the revised version, we found that *PRKCD*, one of PKC³⁶ (Nature medicine, 2004, PMID: 14966518), was significantly overrepresented in the *DST* mutation group (Wilcoxon signed-ranked test, adjust. $p = 0.023$, *DST* Mut vs. WT ratio = 1.96) (**Figure RL32a, also shown in Fig. 4f**), which exhibited positive association with *DST* protein level (Pearson's $R = 0.33$, $p = 0.011$) (**Figure RL32b, also shown in Extended Data Fig. 5k**). These results further indicated the functions of *DST* mutation on mTOR signaling in the DAC stage, providing a reference database for the corresponding personalized medicine of DAC.

Figure RL32 | The impacts of *DST* mutation in DAC stage, also shown in Fig. 4f and Extended Data Fig. 5k in the revised version. a, Heatmap showing the impacts of *DST* mutation on the protein-levels of *DST* and *PRKCD* in the original version (left) and revised version (right) (Wilcoxon signed-ranked test). **b**, Scatterplot showing the comparative analysis of relationship between log₁₀ *DST* and log₁₀ *PRKCD* expression at the protein level in the original version (left) and revised version (right) (Pearson's correlation). **** $p < 1.0E-4$, *** $p < 1.0E-3$, ** $p < 1.0E-2$, * $p < 0.05$, ns. > 0.05 .

Q2. With 140 cases, one would expect a few documented cases of FAP and HNPCC, and I see no direct mention of this and how it fits in with the authors genomic and proteomic data.

Response: We thank the reviewer for the constructive comments and valuable suggestions. According to the reviewer's comments, we divided the response into two parts to answer: (1) whether a few documented cases associated with FAP and HNPCC; (2) about the molecular characterizations of the cases of FAP and HNPCC.

(1) Whether a few documented cases associated with FAP and HNPCC.

FAP and HNPCC are two rare and autosomally inherited disorders⁵⁰ (Nature genetics, 1995, PMID: 7550326), and are characterized by the development premalignant and malignant lesions throughout the gastrointestinal tract, such as adenomatous polyps of the stomach, small bowel, etc.

According to the reviewer's suggestion, we re-follow up DC patients and found that 2 (1.3%) FAP patients and 4 (2.6%) HNPCC patients were enrolled in our study (**Table RL6, also shown in Table 1 in "Methods" of the revised manuscript**). We added this section in the "Methods" of the revised manuscript with red text (**Line 1, Page 25**).

Table RL6. Clinical and histopathologic characteristics of Fudan DC cohort (n = 156).

Characteristic	n (%)	Parameter	n (%)
Age		Survival status	
< 50 yr	38 (24.4)	Alive	156 (100)
50~70 yr	101 (64.7)	Death	0 (0)
>70 yr	17 (10.9)	Histopathologic (438 samples)	
Gender		Epithelial components	
Male	55 (35.3)	Yes	325 (74.2)
Female	101 (64.7)	No	113 (25.8)
Smoking		Histology subtype	
Yes	37 (23.7)	Adenocarcinoma (D)	325 (74.2)
No	92 (59.0)	Brunner's gland (G)	97 (22.1)
NA	27 (17.3)	Heterotopic pancreas (P)	1 (0.2)
Drinking		Intermuscular gland (N)	9 (2.1)
Yes	30 (19.2)	Lymphangioma (L)	2 (0.5)
No	93 (59.6)	Cystic dystrophy (C)	3 (0.7)
NA	33 (21.2)	Neuroendocrine tumor (NET)	1 (0.2)
FAP			
Yes	2 (1.3)		
No	116 (74.4)		
NA	38 (24.4)		
HNPCC			
Yes	4 (2.6)		
No	83 (53.2)		
NA	69 (44.2)		

(2) About the molecular characterization of the cases of FAP and HNPCC.

In our cohort, only 5 samples from 2 FAP patients and 13 samples from 4 HNPCC patients were conducted for proteomic profiling. The histopathological features of FAP and HNPCC was showed in **Figure RL33a (also shown in Supplementary material of the revised version)**.

According to the classification of subtype-based panels, 156 DC patients were divided into 4 panels: Panel 1 (P1, patients with D subtype only), Panel 2 (P2, patients with G subtype only), Panel 3 (P3, patients with both the D and G subtypes), and Panel 4 (P4, patients with other subtypes). The samples of FAP and HNPCC were mainly included in the Panel 3 (P3), which both contained the D and G subtypes (**Figure RL33b**). Thus, we supposed that the molecular characterization of FAP and HNPCC was consistent with those in P3, featured by cell proliferation, DNA repair, and ECM signaling. Owing to the limited volume and restricted samples in FAP, we focused on the investigation of HNPCC.

Microsatellite instability (MSI) status (MSI-high and MSI-low) is associated with HNPCC⁵¹ (Nature genetics, 2004, PMID: 15184898), while the impacts on protein-level are still unknown. In our study, compared with the G subtype, higher MSI score was detected in the D subtype of HNPCC (Kruskal-Wallis test, $p = 0.036$) (**Figure RL33c, also shown in Extended Data Fig. 5g**).

To investigate the molecular characterization of the MSI-high group (D subtype samples) and MSI-low group (G subtype samples) in HNPCC, we analyzed the significantly high expressed proteins

(Kruskal-Wallis test, adjust. $p < 0.05$), and found that MSI-high group proteins were enriched in glycolysis (e.g., TPI, ENO1, etc.), cell cycle (e.g., TOP2A, ORC2, etc.), and DNA repair (e.g., XPC, PARP1, etc.) (**Figure RL33d, e, also shown in Extended Data Fig. 5h**). In the MSI-low group, ECM signaling related proteins were overrepresented, such as COL1A1, LAMA2, etc. (**Figure RL33d, e, also shown in Extended Data Fig. 5h**). The molecular characterizations of the MSI-high group and MSI-low group in HNPCC was similar with those in the D and G subtypes in P3. Consistent with the features of the normal tissues in P3, metabolic process was dominant, including purine catabolism, fatty acid metabolism, and biological oxidation in the normal tissues of HNPCC (**Figure RL33d, e, also shown in Extended Data Fig. 5h**). Therefore, the findings indicated that the molecular characterization of HNPCC was consistent with that in P3.

Schwitalle et al., pointed that the patients with MSI-high HNPCC are sensitive to immune response⁵² (Gastroenterology, 2008, PMID: 18395080). To evaluate the immune infiltration of HNPCC, we applied xCell (<https://xcell.ucsf.edu>)¹² (Genome biology, 2017, PMID: 29141660) to the samples in HNPCC, and found that higher immune score was detected in the MSI-high group (D subtype) of HNPCC (Kruskal-Wallis test, adjust. $p = 0.011$), evidenced by the higher HLA-I score (Kruskal-Wallis test, adjust. $p = 0.034$) and immune signatures of CD4⁺ T cells (Kruskal-Wallis test, adjust. $p = 0.045$) and CD8⁺ T cells (Kruskal-Wallis test, adjust. $p = 0.039$) (**Figure RL33f, also shown in Extended Data Fig. 6k**).

Briefly, integration of the links of published studies and our cohort, revealed that HNPCC represented diverse molecular characterization on the basis of MSI status, and specifically, MSI-high group (like D subtype) was featured by DNA repair and cell cycle, and the ECM signaling was predominant in the MSI-low group (like G subtype) with lower immune infiltration (**Figure RL33g**). The limited samples of HNPCC exhibited a barrier for presenting comprehensive features of HNPCC, more samples are needed for further analysis in the future. We thank the reviewer again for the insightful comments. We had added this section in the revised manuscript with red text (**Line 6–22, Page 22**)

Figure RL33 | The molecular characterization of FAP and HNPCC, also shown in Extended Data Fig. 5 and Extended Data Fig. 6 in the revised version. a, H&E-staining of FAP (top) and HNPCC (Bottom). b, Sankey diagram analysis of all the samples in FAP (top) and HNPCC (bottom). c, Boxplots showing the MSI score in the NT tissues, D subtype tissue, and G subtype tissues in HNPCC (Kruskal-Wallis test). d, The represented pathways in the NT tissues, MSI-high group (D), and MSI-low group (G). e, The molecules in the represented pathways in the NT tissues, MSI-high group (D), and MSI-low group (G). f, Boxplots showing the immune infiltration in HNPCC (Kruskal-Wallis test). g, A brief summary of the molecular characterization in HNPCC. ** $p <$**

1.0E-4, *** $p < 1.0E-3$, ** $p < 1.0E-2$, * $p < 0.05$, ns. > 0.05 .

Q3. While the authors take time to describe that various subtypes of duodenal neoplasia, such as Type D and G, in Figure 1, subtypes P, N, L, and C are listed as substages, which is inaccurate.

Response: Thanks for the reviewer's comments and reminder, and we agree with the reviewer's points that the description of various subtypes of duodenal neoplasia as substages was inaccurate. In the revised version, we corrected the description of "substages NET, P, L, P, N, L, C" as "subtypes NET, L, P, N, L, C" throughout the manuscript and figures. Please see the details in the revised version.

Q4. I am not certain whether the Tobacco Signature is based on clinical history of patients using chewing tobacco. Moreover, in my review of the literature, to include some papers from Asia, I see no epidemiologic evidence to suggest smokeless tobacco or cigarette smoking conclusively increases the risk of duodenal adenocarcinoma, with only some limited data that there may be a risk of neuroendocrine tumors. This should be clearly discussed and explained in their discussion. Lastly, based on my review of the figures, the signature seems to be common in low-grade and high-grade dysplasia, but I don't really "see it" in adenocarcinoma.

Response: Thank you again for the carefully read and constructive comments to improve the quality of our manuscript. According to the reviewer's comments, we divided the responses into three parts to answer: (1) whether the Tobacco signature is based on the clinic history of patients using chewing tobacco; (2) about the literatures to presenting the risk factor of smoking to duodenal adenocarcinoma (D subtype); (3) about the Tobacco signature in the duodenal adenocarcinoma (DAC) stage of the D subtype.

(1) Whether the Tobacco signature is based on the clinic history of patients using chewing tobacco;

In our cohort, the defined Tobacco signature was based on the WES data and the clinic history of patients using chewing tobacco.

Firstly, mutational analysis revealed that Tobacco signature was overrepresented in the D subtype (**Figure RL34a, also shown in Fig. 2a**). Secondly, to investigate whether tobacco usage is associated with the Tobacco signature, we integrated the clinic information of DC patients, and found that, more patients (60% vs. 18.8%, Fisher's exact test, $p = 1.4E-3$) with smoking habit were observed in the Tobacco signature group with higher SBS29 (Tobacco signature) score (Wilcoxon signed-rank test, $p = 0.024$) (**Figure RL34b, c, also shown in Fig. 2c and Extended Data Fig. 3d**). These findings reflected that tobacco might be a risk factor for DC, especially for the D subtype. (**Line 8–15, Page 7**).

Figure RL34 | The association between the Tobacco signature and the clinic information of DC patients, also shown in Fig. 2a, c, and Extended Data Fig. 3d in the revised version. a, The major somatic SNVs signature in DC progression. **b,** Histogram showing the proportion of the samples with the habit of smoking in the Tobacco signature and DNA repair panel (Fisher’s exact test). **c,** Barplots presenting the SBS29 contribution in the patients with the habit of smoking or not (Wilcoxon signed-rank test). **** $p < 1.0E-4$, *** $p < 1.0E-3$, ** $p < 1.0E-2$, * $p < 0.05$, ns. > 0.05 .

(2) About the literatures to presenting the risk factor of smoking to duodenal adenocarcinoma;

In our review of the literatures, there is no literature directly suggest tobacco usage increased the risk of duodenal adenocarcinoma (D subtype in our cohort). However, several published studies pointed that smoking is a known risk factor in a number of gastrointestinal (GI) diseases, including duodenal ulcer⁴⁰ (Microbiome, 2018, PMID: 30157953) and GI cancer⁴¹ (JAMA, 2008, PMID: 19088354). In addition, tobacco usage has important implications regarding gut environment and various small intestine disorders, including functional dyspepsia⁴² (Gut, 2017, PMID: 27489239), irritable bowel syndrome⁴³ (Gastroenterology, 2017, PMID: 27725146), coeliac disease⁴⁴ (Clin Gastroenterol Hepatol, 2005, PMID: 16234024), and duodenal ulcer disease⁴⁵ (N Engl J Med, 1956, PMID: 13334789).

Furthermore, Buchthal, J. et al.⁴⁶ (Eur J Clin Pharmacol, 1995, PMID: 7720765) demonstrated cytochromes P450 families (CYPs) induction by constituents of cigarette smoke in the human duodenum. In our cohort, CYPs were overrepresented in the Tobacco signature at the protein level (**Figure RL35, also shown in Fig. 2d**). Besides, we found more patients (60% vs 18.8%), especially the D subtype patients, with smoking habit were observed in the Tobacco signature group (**Figure RL34b, also shown in Fig. 2c**). Integration with the findings in our cohort and the impacts of smoking on gastrointestinal cancer, including esophageal cancer, gastric cancer, colorectal cancer, etc., indicated that smoking or tobacco usage might be a risk for DC, especially for the D subtype.

Figure RL35 | The represented proteins from CYPs and in the predominant pathways of the Tobacco signature (SBS29 signature), also shown in Fig. 2d in the revised version.

However, to explore the correlation between tobacco usage and DC, more evidences are needed from other cohorts or epidemiologic studies with more DC patients for further analysis. In addition, more samples from DC patients with diverse clinic features, are required to illustrate molecular mechanisms and provide corresponding potential clinic therapy for DC, especially for the two major subtypes (D and G). We had added this section in the “**Discussion**” of the revised manuscript with red text (Line 3–5, Page 22).

(3) About the Tobacco signature in the duodenal adenocarcinoma (DAC) stage of the D subtype.

We thank the reviewer for the constructive suggestions, and we are sorry for not presenting the Tobacco signature in the DAC stage clearly, which was detected in the low-grade intraepithelial neoplasia (LGIN) stage and last till to the DAC stage.

In the revised version, to further validate the solid result in our study, apart from 406 samples, we added 16 DAC stage samples with paired normal tissues from 16 D subtype patients for proteomic profiling and WES. That indicated 32 samples (16 DAC stage + 16 paired normal tissues samples) were both added for proteomic profiling and WES (Table RL7). Therefore, a total of 438 samples from 156 DC patients were used to perform a comprehensive proteogenomic analysis in our study, in which 120 samples were conducted for WES (Figure RL36, also shown in Fig. 1a). We added this section the revised manuscript with red text (Line 12–15, Page 5).

Table RL7 | The number of the samples for genome (WES) and proteome in the Fudan cohort.

Omics	Original version			Revised version		
	Total	Major subtypes	Rare subtypes	Total	Major subtypes	Rare subtypes
Genome	88	82	6	120	114	6
Proteome	406	390	16	438	422	16

Major subtypes: D and G subtypes Rare subtypes: NET, P, NET, L, and C subtypes

Figure RL36 | Overview of the experimental design and the number of samples for the genomic, proteomic, and phosphoproteomic analyses, also shown in Fig. 1a in the revised version.

We thank the reviewer for valuable comments to improve the quality of our manuscript. After adding 32 samples for WES in the revised version, higher proportion (25.0% vs. 6.7%) of SBS29 signature (Tobacco signature) was detected in the DAC stage of the D subtype (**Figure RL37a**). In addition, the enrichment of SBS29 contribution (Wilcoxon signed-ranked test, Tobacco signature vs. DNA repair panel ratio = 3.49, $p < 1.0E-4$) was more significantly overrepresented in the Tobacco signature in the revised version (**Figure RL37b**). Therefore, the findings further indicated the Tobacco signature was featured in the D subtype, and was detected as early as in the NT stage and last till to the DAC stage.

Figure RL37 | Tobacco signature in DC. a, Venn diagram depicting the stage proportion of the SBS29 signature in the original version (left) and revised version (right). **b**, Boxplot showing the SBS29 contribution in the two major signatures of DC in the original version (left) and revised version (right). **** $p < 1.0E-4$, *** $p < 1.0E-3$, ** $p < 1.0E-2$, * $p < 0.05$, ns. > 0.05 .

In the revised version, after adding 32 samples for WES, other more solid results were also observed, apart from the more DAC stage samples and high SBS29 contributions in the Tobacco signature in the revised version. Several examples were listed as follows.

As for the somatic CNAs.

After adding 32 samples for WES, more arm events were detected in DC progression, in which the gain of chromosome 8q gain (chr8q) was overrepresented in the infiltration tumor (IFT) phase (**Figure RL38a, also shown in Fig. 3a**). In the revised version, we found more samples with chr8q gain were detected in the IFT phase (**Figure RL38b, also shown in Fig. 3c**). At the chr8q gain, we found the amplification of *LYN* played a key role in MAPK signaling. In the revised version, the amplification of *LYN* showed more significant *cis*-effects on its corresponding protein level (**Figure RL38c**). That findings further indicated the functions of *LYN* amplification at the chr8q gain, on MAPK signaling.

Figure RL38 | The arm events in DC progression, also shown in Fig.3a, c in the revised version.
a, Profiling of absolute copy number alterations observed in DC. The square directed to a subset of patient samples used for WES. **b**, Heatmap showing the distribution of the chr8q gain in the D subtype. **c**, Volcano plot showing the *cis* SCNA-protein regulations of *LYN* amplification (Spearman's correlation). **** $p < 1.0E-4$, *** $p < 1.0E-3$, ** $p < 1.0E-2$, * $p < 0.05$, ns. > 0.05 .

As for the impacts of *DST* mutation.

After adding 32 samples for WES, we found consistent results and conclusions compared with the original version. Specifically, we found that more samples with *DST* mutation were detected in the DAC stage in the revised version (**Figure RL39a, also shown in Fig. 4f**). In addition, *DST* mutation significantly enhanced the expression of *DST* at the protein level (Wilcoxon signed-ranked test, adjust. $p = 0.027$, *DST* Mut vs. WT ratio = 3.19) (**Figure RL39a, also shown in Fig. 4f**).

Generally, *DST* plays a key role in calcium (Ca^{2+}) signal and increased Ca^{2+} level³⁵ (Autoimmunity, 2002, PMID: 12482196), which is the activator in the transmit process from the phospholipid and diacylglycerol (DAG) to protein kinase C (PKC)³⁶ (Nature medicine, 2004, PMID: 14966518). In the revised version, we found that *PRKCD*, one of PKC³⁶ (Nature medicine, 2004, PMID: 14966518), was significantly overrepresented in the *DST* mutation group (Wilcoxon signed-ranked test, adjust. $p = 0.023$, *DST* Mut vs. WT ratio = 1.96) (**Figure RL39a, also shown in Fig. 4f**), which exhibited positive association with *DST* protein level (Pearson's $R = 0.33$, $p = 0.011$) (**Figure RL39b, also shown in Extended Data Fig. 5k**). These solid results further indicated the functions of *DST* mutation on mTOR signaling in the DAC stage, providing a reference database for the corresponding personalized medicine of DAC.

Figure RL39 | The impacts of DST mutation in DAC stage, also shown in Fig. 4f and Extended Data Fig. 5k in the revised version. a, Heatmap showing the impacts of *DST* mutation on the protein-levels of *DST* and *PRKCD* in the original version (left) and revised version (right) (Wilcoxon signed-rank test). **b**, Scatterplot showing the comparative analysis of relationship between log₁₀ *DST* and log₁₀ *PRKCD* expression at the protein level in the original version (left) and revised version (right) (Pearson's correlation). **** $p < 1.0E-4$, *** $p < 1.0E-3$, ** $p < 1.0E-2$, * $p < 0.05$, ns. > 0.05 .

Q5. Likewise, I am unclear if the "DNA Repair signature" is limited to cases with documented MSI-high tumors.

Response: We appreciate the constructive and insightful comments. In our study, we used non-negative matrix factorization (NMF) algorithm⁵³ (Nature, 1999, PMID: 10548103) to estimate the minimal components explaining maximum variance among DC samples. The signatures, such as SBS6, 15, and 88, correlated with DNA repair⁵⁴ (Nature, 2020, PMID: 32025018), were named as "DNA repair panel" in our cohort (**Figure RL40a, also shown in Fig. 2a**).

MSI occurs alterations in short-repetitive DNA sequences⁵⁵ (Crit Rev Oncol Hematol, 2022, PMID: 35033663), as consequence of DNA mismatch repair (MMR) deficiency⁵⁶ (Pharmacol Ther, 2018, PMID: 29669262), and shows positive association with the presence of hypermutation^{57,58} (Cancer research, 2002, PMID: 12097267; Nature medicine, 2015, PMID: 25894828). In our cohort, compared with the Tobacco signature, significantly higher tumor mutation burden (TMB) of the DNA repair panel was detected (Wilcoxon signed-rank test, $p < 1.0E-4$) (**Figure RL40b, also shown in Fig. 2g**), suggesting MSI may be exist in the DNA repair panel of DC.

In order to validate the assumptions, we performed MSI analysis and found higher MSI score were detected in the DNA repair panel (Wilcoxon signed-rank test, $p = 0.016$), evidenced by the overrepresented MSI signature related proteins in the DNA repair panel (**Figure RL40c, d, also shown in Fig. 2i, h**). These findings indicated that MSI-high samples of DC were more overrepresented in the DNA repair panel. The result will be evidenced with more investigations in the future.

Figure RL40 | The impacts of the tobacco signature and DNA repair panel in DC, also shown in Fig. 2 in the revised version. **a** The major signatures of somatic SNVs in the D and G subtypes. **b** Boxplot showing the mutation frequency in the Tobacco signature and DNA repair panel (Wilcoxon signed-rank test). **c** Heatmap depicting the represented proteins of MSI signature in the Tobacco signature group. The square directed to a subset of patient samples in the D subtype used for WES. **d** Boxplot showing the MSI score in the Tobacco signature and DNA repair panel (Wilcoxon signed-rank test). **** $p < 1.0E-4$, *** $p < 1.0E-3$, ** $p < 0.01$, * $p < 0.05$.

Q6. The sheer volume of figures and extended figures are overwhelming and made it difficult to follow the textual component of the paper in parallel with the graphical part. I think the data presentation should be substantially edited, and perhaps limit it to the proteomic and phosphoproteomic results of the analysis. Alternative, to focus purely on Subtype D, the most common seen clinically may be of greater value to the readership.

Response: We sincerely thank the reviewer for the professional suggestions and constructive comments. We have studied the comments carefully and addressed all the points raised by the reviewer. In addition, we streamlined the manuscript and removed redundant figures and text, which will lead to a clearer, broader, and more compelling manuscript. We divided the responses into three parts to answer: (1) about streamlining the figures and extended figures; (2) about the comparison and findings of the DC subtypes; (3) about the data presentation and the major findings of the manuscript.

(1) About streamlining the figures and extended figures.

In our study, we performed a comprehensive proteogenomic analysis of 438 trace tumor samples from 156 duodenal cancers (DC), covering 2 major and 5 rare subtypes. COSMIC-based comparative analysis revealed that more patients with the habit of smoking were observed in the Tobacco signature, which was prominent in the D subtype and might induce NNK-O-glucuronic

acid secretion via the overrepresentation of GAGs. Proteogenomics revealed chromosome 8q gain may functioned in the transmit from the IEN phase to the IFT phase, in which MAPK signaling was activated. Proteome-based analysis elucidated the stage-specific molecular characterization, and defined the cancer-driving waves along with the mutation accumulation in D and G subtypes. The mutation of *DST* improved mTOR signaling in the DAC stage. In addition, we found the DNA repair signature drove the duodenal epithelial neoplasia, in which gradually and significantly overexpressed AARS1 played a key role, and catalyzed the lysine-alanylation of poly-ADP-ribose polymerases (PARP1), which decreased the apoptosis of cancer cells, eventually promoting cell proliferation and tumorigenesis. We assessed the proteogenomic landscape of early DC, and provided insights into the molecular features corresponding therapeutic targets.

In the revised version, to present the data and manuscript logically and readably, we streamlined the manuscript and re-arranged the figures and supplementary figures in the revised version. In addition, we removed the redundant and unrelated information.

For example, **Fig. 2 and Extended Data Fig. 3** showed two major findings: 1) the Tobacco signature, related to the clinic information of DC patients, was overrepresented in the D subtype and might induce NNK-O-glucuronic acid secretion via the overrepresentation of GAGs; 2) the DNA repair signature were featured with TMB, and drove the duodenal epithelial neoplasia, in which ARS-PARPs signaling was overrepresented. In the revised version, we added the MSI score, which were overrepresented in the DNA repair signature (**Fig. 2h**). In addition, we removed the unclear **Extended Data Fig. 3b, c**. Owing to Extended Data Fig. 3a, showing the correlation among of the DC subtypes, we moved the **Extended Data Fig. 3a** to **Extended Data Fig. 5 in the revised version**, which revealed that divergency of DC subtypes was derived from their origin normal tissues and diverse carcinogenesis tracks.

As for other figures, to present the figures readably and concisely, we integrated the compatible figures and removed the complex information. For example, the **Fig. 4e** (revised version) was integrated from the **Fig. 4e and Extended Data Fig. 5f** of the original version. Besides, we removed the redundant figures, such as **Extended Data Fig. 2f and Extended Data Fig. 5c, e**. In addition, we removed the un-visualized color-codes presentation of the stages/subtypes and other clinic information in **Fig. 1a, Fig. 1b, Fig. 2a, Fig. 3f, Fig. 5a**, etc. Please see the details in the revised version.

Furthermore, to make our statement more accurate and systematical, we streamlined our manuscript thoroughly, and tuned down our statement, which was updated in the revised manuscript with red text. Please see more details in the revised manuscript with red text.

(2) About the comparison and findings of different DC subtypes (D, NET, L, P, G, N, C).

As for the major findings of the D subtypes.

According to the locations of DC, D subtype (epithelial component) and G subtype (non-surface epithelial component)⁵⁹ (Advances in therapy, 2021, PMID: 33914269) are 2 major subtypes of DC.

On the basis of the clinicopathological characteristics, the D subtype could be divided into 17 substages during the carcinogenesis. In this study, we performed a comprehensive proteogenomic landscape of the D subtype. According to the pathological features, the D subtype was divided into the NT phase, IEN phase, and IFT phase. Compared to the other duodenal cancer types, low mutation loads were observed in the IEN phase of D subtype, indicating low mutation loads in the earlier phase in DC progression.

Mutational analysis revealed that the DNA repair panel (including SBS6, SBS15, SBS30) and SBS29 signature, which is correlated with tobacco chewing, were prominent in the D subtype. Specifically, the Tobacco signature (SBS29 signature) with lower TMB, might induce tumor invasion via the overrepresentation of glycosaminoglycans (GAG). In addition, we found higher MSI score in the DNA repair panel, in which ARS-PARPs signaling was overrepresented. Whole exome-based SCNAs analyses illustrated the chr8q gain was a key event in the transmit process from the IEN phase to the IFT phase, in which the amplification of *LYN* enhanced the MAPK signaling and thus promoted the DC cell proliferation. To further depict staging molecular models drove carcinogenesis from early to progressive D subtype, we performed substage-based supervised clustering analysis and described the proposed driving pathways and characteristic molecules of different substages, and found that the DAC stage prominent mutation of *DST* exhibited positive impacts on mTOR signaling.

As for the reason why keeping other DC subtypes (NET/L/P/G/N/C) in our study.

Firstly, it is common that one DC patient had more than one subtype.

DC is one of the malignantly cancers with low incidence, and featured with complicated subtypes, including D, NET, L, P, G, N, and C subtypes. It is common that one DC patient had more than one subtype. The rarity and complicated subtypes of DC remain challenging to collect the DC samples, especially for the rare subtype NET/L/P/N/C subtypes.

In our cohort, we collected 2 major (D/G) and 5 rare subtypes (NET/L/P/N/C) of DC. In addition, according to the 4th and 5th edition World Health Organization Classification of Digestive system, the D and G subtype were subclassified into 17 and 6 substages, respectively. All the samples of DC patients were processed with the same standard operating procedure, providing the valuable datasets of DC, and allowing us to explore the diversity and characterization of the DC subtypes. Moreover, the advances of samples in a pathological region resolved mode in the D and G subtypes, provided the chances to portray molecular profiles of DC in a stage resolved mode.

Furthermore, one DC patient had more than one subtype in our cohort. For example, 60 patients (38.5%) had the D subtype only, 32 patients (20.5%) had the G subtype only, 54 patients (34.6%) had both D and G subtypes, and 10 patients (6.4%) had other subtypes. Whereas, the origins of the subtypes are yet unknown, the features of DC patients with different subtypes provided us a chance to track the origins of DC subtypes.

Secondly, the comparison and findings of the DC subtypes contributed to a comprehensive

proteogenomic characterization of DC.

i) The malignant potential of DC subtypes.

Among of the DC subtypes, the D and NET are relatively malignant subtypes, comparably, the G subtype is benign. However, the malignant potential of P/L/N/C subtypes are not clear, building a barrier in clinical intervention. To explore the malignant potential of the rare subtypes, we performed comparative analysis of DC subtypes. As a result, we found that the N/C subtypes were more closely co-clustered with the G subtype, while the expression profiles of the L/P subtypes were globally akin to the D/NET subtype (**Figure RL41, also shown in Extended Data Fig. 5d**). The results indicated that the D/NET/L/P subtypes were relatively malignant, and the G/N/C were comparably benign in DC, providing a guidance in clinical intervention.

Figure RL41 | Hierarchical clustering analysis of 2 major and 5 rare subtypes of DC, also shown in Extended Data Fig. 5d in the revised version.

ii) The diverse immune infiltration of DC subtypes.

Generally, immune infiltration is used to assess the tumor progression and prognosis. Owing to the low incidence of DC, especially of the rare subtypes, the immune characterization is yet uncovered, and the prognosis of the DC subtypes is still undefined. To evaluate the immune infiltration of the DC subtypes, we applied xCell (<https://xcell.ucsf.edu>)¹² (Genome biology, 2017, PMID: 29141660) to the 438 samples of DC cohort, and deconvoluted immune, stromal, and microenvironmental cell signatures of DC. Consensus clustering identified four immune-based clusters as *Hot tumor* cluster, *Cold tumor* cluster, *Epithelial* cluster, and *B cells* cluster (**Figure RL42a, b, also shown in Fig. 5a and Extended Data Fig. 6a**). Remarkably, the D subtype, closely co-clustered with NET/L/P subtypes, was overrepresented in the *Hot tumor* cluster and the *Epithelial* cluster, implying the better prognosis of DC (**Figure RL42c, also shown in Fig. 5d**). The non-epithelial components of DC, such as the G, N, and C subtypes, were notably observed in the *Cold tumor* cluster and the *B cells* cluster, which was characterized by high stromal score (**Figure RL42c, also shown in Fig. 5d**). These results further indicated the malignant potential of the DC subtypes, and implied the possible prognosis and immune infiltration of the DC subtypes, improving the recognition of DC in the clinic.

Figure RL42 | Characterization of immune-based clusters in DC subtypes, also shown in Fig. 5a, b, d, and Extended Data Fig. 6a in the revised version. a, Immune-based consensus clustering of DC samples. **b**, Heatmap illustrating cell type compositions and activities of the functional genes/proteins across the four immune clusters. The square directed to a subset of patient samples used for proteomic profiling. **c**, Density contours of immune and stromal scores of four immune clusters. **** $p < 1.0E-4$, *** $p < 1.0E-3$, ** $p < 1.0E-2$, * $p < 0.05$, ns. > 0.05 .

iii) The diverse characteristic origins between the D and G subtypes.

The complexity of DC that one patient had more than one subtype, remains challenging to the molecular characterization and therapeutic strategy. In addition, the unclear origins, especially for the two major subtypes (D and G), impede the exploration of the carcinogenesis of DC. In our cohort, 60 patients (38.5%) had the D subtype only, 32 patients (20.5%) had the G subtype only, 54 patients (34.6%) had both D and G subtypes, and 10 patients (6.4%) had other subtypes, allowing us to explore the diverse origins and molecular characterizations of DC. We divided 156 DC cases into 4 panels shown in Sankey diagram: panel 1 (P1, patients with D subtype only), panel 2 (P2, patients with G subtype only), panel 3 (P3, patients with both the D and G subtypes), and panel 4 (P4, patients with other subtypes) (Figure RL43a, also shown in Fig. 4a). Based on the expression trend, growingly upregulated DEPs in the progression of the P1 (D subtype panel) were enriched in cell cycle (e.g., CDK1/2/4/6, SKP1, CDKN2A, etc.), apoptosis (e.g., CASP3/8, AARS1, PARP1/9, etc.), and glycolysis (e.g., HK2, PGK1, ENO1/2/3, etc.), while the UDEPs in the P2 (G subtype panel) demonstrated the positive association of ECM signaling (e.g., COL1A1, LAMA1, TNXB, etc.) and complement cascade (e.g., CFH, C4A, C6, etc.) in the carcinogenesis of the G subtype (Figure RL43b, also shown in Fig. 4b).

To access whether the differential features of the D and G subtypes were derived from the origins of their normal tissues, we compared the proteomes of normal tissues in the P1 and the P2. Specifically, the highly expressed proteins of normal tissues in the P1 were enriched in cell cycle, apoptosis, and glycolysis (**Figure RL43b, also shown in Fig. 4b**). In the P2, complement cascade and ECM signaling were dominant in the normal tissues, suggesting the diversity of the D and G subtypes was derived from their origin normal tissues (**Figure RL43b, also shown in Fig. 4b**). In addition, the P3 consisted of the patients with both D and G subtypes, which shared the common normal tissues. Compared to the P1 (D subtype panel), ECM signaling (e.g., COL1A1, COL1A2, LAMA1, etc.) was highlighted in the normal tissues of the P3 (**Figure RL43c, also shown in Fig. 4c**). While compared to the P2, cell cycle (e.g., CDK1, CDK2, CDK4, etc.) was observed in the normal tissues of the P3 (**Figure RL43c, also shown in Fig. 4c**). Together, these findings further supported that the divergency of DC subtypes was derived from their origin normal tissues and thus contributed to diverse carcinogenesis tracks.

Therefore, the comparative analysis of DC subtypes provided valuable multi-omics data and improve the knowledge about different characterizations of DC subtypes. We thank the reviewer again for the valuable comments, and will continue to collect more samples of the G, NET, L, P, N, and C subtypes for further analysis in the future.

Figure RL43 | The diverse characteristic origins of the D and G subtypes, also shown in Fig. 4a, b, c, and Extended Data Fig. 5f in the revised version. **a**, Sankey diagram analysis of all the 156 cases classified into 4 panels. **b**, Venn diagram depicting the specific driven pathways in the normal (left) and tumor (right) tissues of the P1 and P2. The number represented the UDEPs. The green highlighted the normal tissue. The blue and orange indicated the D and G subtypes, respectively. **c**, The morphological feature (H&E staining) (left) and represented pathways (right) of normal tissue (stage 1) among the P1, P2, and P3. **d**, The represented pathways of the normal tissues, D subtype, and G subtype in the P3.

(3) About the data presentation and the major findings of the manuscript.

We streamlined and updated the manuscript with red text, focusing on the most important findings and shortening the description more general observations. We performed a comprehensive proteogenomic analysis of 438 trace tumor samples from 156 DC patients, covering 2 major and 5 rare subtypes (**Fig. 1 and Extended Data Fig. 1, 2**). There are **5 major parts** in this study as follows: (1) COSMIC-based signature analysis revealed that more patients with the habit of smoking were observed in the Tobacco signature, which was prominent in the D subtype and might induce NNK-O-glucuronic acid secretion via the overrepresentation of glycosaminoglycans (GAGs). DNA repair panel was featured with high TMB and MSI, in which ARS-PARPs signaling was overactivated (**Fig. 2 and Extended Data Fig. 3**); (2) The chr8q gain might function in the transmit from the IEN phase to the infiltration tumor (IFT) phase, in which MAPK signaling was activated (**Fig. 3 and Extended Data Fig. 4**); (3) Proteome-based analysis disclosed that divergency of DC subtypes was derived from their origin normal tissues, and elucidated the stage-specific molecular characterization (**Fig. 4 and Extended Data Fig. 5**); (4) Stage-based proteomic analysis illustrated the cancer-driving waves along with the mutation accumulation in the D and G subtypes, in which the mutation of *DST* improved mTOR signaling in the DAC stage (**Fig. 4 and Extended Data Fig. 5**); (5) Immune-based cluster delineated the unique profiles of four immune cluster of DC, and illuminated that the *Hot immune cluster* was featured with high TMB and overrepresentation of AARS1 (**Fig. 5 and Extended Data Fig. 6**), which was gradually enhanced in DC progression and catalyzed the lysine-alanylation of PARP1, eventually promoting cell proliferation and tumorigenesis (**Fig. 6, 7, and Extended Data Fig. 7, 8**).

Firstly, to explore the specific etiological factors that may contribute the mutagenesis of DC, we performed the COSMIC-based comparative analysis, and detected two mutational signatures as Tobacco signature and DNA repair panel. Integration with the clinic information of DC patients, more patients with tobacco usage were observed in the Tobacco signature, which was prominent in the D subtype and might induce NNK-O-glucuronic acid secretion via the overrepresentation of GAGs. The DNA repair panel was generally observed in DC, including D, P, G, N, and C subtypes. Specifically, DNA repair signature was featured with high TMB and high MSI score, in which ARS-PARPs signaling was overrepresented (**Fig. 2 and Extended Data Fig. 3**).

Secondly, the WES-based SCNAs analyses represented the chr8q functioned in the transmit process from the IEN phase to the IFT phase in the D subtype, in which the amplification of *LYN* enhanced the protein-level of LYN and enhanced phosphorylation. KSEA revealed MAPK signaling (e.g., MAPK1/3/7, RPS6KA3, etc.) related kinases were overrepresented in the *LYN* amplification group, in which MAPK1 showed positive association with LYN. In addition, the substrate of MAPK1 Y187 was overrepresented in the *LYN* amplification group, which could activate the substrates of RPS6KA3 (S369, S375, S715, and T365) and TOP2A (S1213). Furthermore, LYN positively correlated with cell proliferation, evidenced by MCM4, MCM6, PCNA, TOP2A, etc. Together, the chr8q gain functioned in the transmit process from the IEN phase to the IFT phase, in which the amplification of *LYN* enhanced the protein-level of LYN, which activated MAPK signaling and promoted the DC cell proliferation, hinting the potential medical actions of saracatinib³⁴ (J Clin

Invest, 2014, PMID: 24316974) in the DC clinic (**Fig. 3 and Extended Data Fig. 4**).

Thirdly, view of Sankey program analysis observed 4 panels of DC with 2 major subtypes and 5 rare subtypes of 156 DC cases, and revealed the divergency of the D and G subtypes were derived from their own origins, allowing us to explore the diverse molecular landscape during carcinogenesis of the D and G subtypes based on the differential proteomic patterns. We identified and summarized a carcinogenetic lineage with 5 dynamic pathways in the D subtype progression at multi-omics level: digestion pathway – AMPK signaling/ insulin resistance – cell cycle/ DNA repair – apoptosis/ aminoacyl-tRNA biosynthesis – mTOR signaling/ EGFR signaling. In addition, we displayed a carcinogenesis path with 3 kinetic pathways in the G subtype progression for the first time: digestion pathway – ECM signaling/ focal adhesion – complement cascade/ platelet activation. The DAC stage prominent mutation of *DST*, plays a key role in calcium (Ca^{2+}) signal and increases Ca^{2+} level³⁵ (Autoimmunity, 2002, PMID: 12482196), enhanced the protein level of *DST* which regulated the levels of the kinases (PDK1 and PRKDC), and thus activated mTOR signaling at the protein and phosphoprotein levels. Overall, the strong correlation between proteome and genome further validated the influences of carcinogenesis lineages-pathways of diverse subtypes of DC, and provided a referred proteogenomic dataset of DC (**Fig. 4 and Extended Data Fig. 5**).

Fourthly, the immune based investigation delineated the immune characteristics of DC, and the proteomic immune clustering analysis disclosed the potential actions of anti-CD248 in the *Cold tumor* cluster, and anti-CD276 or CAR-T/CD276 in the *Hot tumor* cluster, providing a newly clinic direction for DC. As well as in the DNA repair panel, highest TMB was detected in the *Hot tumor* cluster. ARS families have been implicated in the etiology of various disorders⁶⁰ (Nature reviews drug discovery, 2019, PMID: 31073243), including cancer and dysregulated metabolic conditions, and can catalyze the posttranslational addition of amino acids to lysine residues in proteins⁶¹ (Cell metabolism, 2018, PMID: 29198988). Interestingly, AARS1, one of ARS families and functioned in the DNA repair panel, showed positive correlation with the TMB and the immune infiltration. In addition, AARS1 was positively associated with the potential clinical markers of the *Hot tumor* cluster dominant pathways, such as DNA repair, cell cycle, apoptosis, ERBB signaling, and so on. These results indicated the involvement of AARS1 in the mutation burden/immune infiltration, and thus we speculated AARS1 was a potential novel target in the clinic strategy of DC (**Fig. 5 and Extended Data Fig. 6**).

Lastly, we designed a series of experiment to validate the functions of AARS1. In this study, we built a new connection between AARS1 and PARP1 in DC development for the first time. We discovered that overexpressed AARS1 catalyzed K-Ala modification on PARP1. Increased K-Ala of PARP1 inhibited PARP1 activity, thus resulting in an inhibitory effect on cell apoptosis. Although the PARP1 expression increased from early to advanced duodenal cancer, the total PARylation in cells decreased significantly. We speculated that PARP1 expression elevation was probably induced by the compensation because its activity was inhibited. Although we did not explore the metabolism of alanine in the current study, we found that upregulated AARS1 was strong enough to cause the occurrence of K-Ala and played an important role in early DC (**Fig. 6, 7, and Extended Data Fig. 7, 8**).

In summary, our study presented a comprehensive multi-omics landscape of DC for the first time. We delineated the characteristic origins and diverse carcinogenesis tracks, and discovered the kinetic waves of the dominant cancer pathways of the D and G subtypes via integrative proteogenomic analysis. Proteogenomics elucidated the positive impacts of *LYN* amplification at the chr8q gain on MAPK signaling in the IFT phase, and revealed the functions of *DST* mutation on mTOR signaling in the DAC stage. We illuminated the drug-targetable AARS1 in the TMB/immune infiltration in DC, which was demonstrated the value of this multi-omics mapping strategy to suppress tumor growth (**Figure RL44, also shown in Extended Data Fig. 9**). We believe this study provides insights into understanding DC and enables new advances in understanding its mechanism and diagnostics, delivering a useful resource for potential therapeutic approaches and personalized medicine for DC.

Figure RL44 | The model of the key events in the progression of DC, also shown in Extended Data Fig. 9 in the revised version.

References

- 1 Emmert-Buck, M. R. *et al.* Laser capture microdissection. *Science* **274**, 998-1001, doi:10.1126/science.274.5289.998 (1996).
- 2 Waddell, N. *et al.* Whole genomes redefine the mutational landscape of pancreatic cancer. *Nature* **518**, 495-501, doi:10.1038/nature14169 (2015).
- 3 Eckert, M. A. *et al.* Proteomics reveals NNMT as a master metabolic regulator of cancer-associated fibroblasts. *Nature* **569**, 723-728, doi:10.1038/s41586-019-1173-8 (2019).
- 4 Lam, K. H. B. *et al.* Topographic mapping of the glioblastoma proteome reveals a triple-axis model of intra-tumoral heterogeneity. *Nat Commun* **13**, 116, doi:10.1038/s41467-021-27667-w (2022).
- 5 Zhu, Y. *et al.* High-throughput proteomic analysis of FFPE tissue samples facilitates tumor stratification. *Mol Oncol* **13**, 2305-2328, doi:10.1002/1878-0261.12570 (2019).
- 6 Xu, N. *et al.* Integrated proteogenomic characterization of urothelial carcinoma of the bladder. *J Hematol Oncol* **15**, doi:ARTN 7610.1186/s13045-022-01291-7 (2022).
- 7 Gao, Q. *et al.* Integrated Proteogenomic Characterization of HBV-Related Hepatocellular Carcinoma (vol 179, pg 561, 2019). *Cell* **179**, 1240-1240, doi:10.1016/j.cell.2019.10.038 (2019).
- 8 Clark, D. J. *et al.* Integrated Proteogenomic Characterization of Clear Cell Renal Cell Carcinoma. *Cell* **179**, 964-983 e931, doi:10.1016/j.cell.2019.10.007 (2019).

- 9 Deng, M. *et al.* Proteogenomic characterization of cholangiocarcinoma. *Hepatology*, doi:10.1002/hep.32624 (2022).
- 10 Keren, L. *et al.* A Structured Tumor-Immune Microenvironment in Triple Negative Breast Cancer Revealed by Multiplexed Ion Beam Imaging. *Cell* **174**, 1373-1387 e1319, doi:10.1016/j.cell.2018.08.039 (2018).
- 11 Harel, M. *et al.* Proteomics of Melanoma Response to Immunotherapy Reveals Mitochondrial Dependence. *Cell* **179**, 236-250 e218, doi:10.1016/j.cell.2019.08.012 (2019).
- 12 Aran, D., Hu, Z. & Butte, A. J. xCell: digitally portraying the tissue cellular heterogeneity landscape. *Genome Biol* **18**, 220, doi:10.1186/s13059-017-1349-1 (2017).
- 13 Bao, K. *et al.* Pressure Cycling Technology Assisted Mass Spectrometric Quantification of Gingival Tissue Reveals Proteome Dynamics during the Initiation and Progression of Inflammatory Periodontal Disease. *Proteomics* **20**, e1900253, doi:10.1002/pmic.201900253 (2020).
- 14 Powell, B. S., Lazarev, A. V., Carlson, G., Ivanov, A. R. & Rozak, D. A. Pressure cycling technology in systems biology. *Methods Mol Biol* **881**, 27-62, doi:10.1007/978-1-61779-827-6_2 (2012).
- 15 Zhu, Y. *et al.* High-throughput proteomic analysis of FFPE tissue samples facilitates tumor stratification. *Mol Oncol* **13**, 2305-2328, doi:10.1002/1878-0261.12570 (2019).
- 16 Sagu, S. T., Landgraber, E., Rackiewicz, M., Huschek, G. & Rawel, H. Relative Abundance of Alpha-Amylase/Trypsin Inhibitors in Selected Sorghum Cultivars. *Molecules* **25**, doi:10.3390/molecules25245982 (2020).
- 17 Li, L. *et al.* Integrative proteomic characterization of trace FFPE samples in early-stage gastrointestinal cancer. *Proteome Sci* **20**, 5, doi:10.1186/s12953-022-00188-0 (2022).
- 18 Ge, S. *et al.* A proteomic landscape of diffuse-type gastric cancer. *Nat Commun* **9**, 1012, doi:10.1038/s41467-018-03121-2 (2018).
- 19 Jiang, Y. *et al.* Proteomics identifies new therapeutic targets of early-stage hepatocellular carcinoma. *Nature* **567**, 257-+, doi:10.1038/s41586-019-0987-8 (2019).
- 20 Tyanova, S., Temu, T. & Cox, J. The MaxQuant computational platform for mass spectrometry-based shotgun proteomics. *Nat Protoc* **11**, 2301-2319, doi:10.1038/nprot.2016.136 (2016).
- 21 Satpathy, S. *et al.* A proteogenomic portrait of lung squamous cell carcinoma. *Cell* **184**, 4348-4371 e4340, doi:10.1016/j.cell.2021.07.016 (2021).
- 22 Liu, W. *et al.* Large-scale and high-resolution mass spectrometry-based proteomics profiling defines molecular subtypes of esophageal cancer for therapeutic targeting. *Nat Commun* **12**, 4961, doi:10.1038/s41467-021-25202-5 (2021).
- 23 Xu, J. Y. *et al.* Integrative Proteomic Characterization of Human Lung Adenocarcinoma. *Cell* **182**, 245-+, doi:10.1016/j.cell.2020.05.043 (2020).
- 24 Cox, J. & Mann, M. MaxQuant enables high peptide identification rates, individualized p.p.b.-range mass accuracies and proteome-wide protein quantification. *Nat Biotechnol* **26**, 1367-1372, doi:10.1038/nbt.1511 (2008).
- 25 Ni, X. *et al.* A region-resolved mucosa proteome of the human stomach. *Nat Commun* **10**, 39, doi:10.1038/s41467-018-07960-x (2019).

- 26 Hoadley, K. A. *et al.* Cell-of-Origin Patterns Dominate the Molecular Classification of 10,000 Tumors from 33 Types of Cancer. *Cell* **173**, 291-304 e296, doi:10.1016/j.cell.2018.03.022 (2018).
- 27 Tsherniak, A. *et al.* Defining a Cancer Dependency Map. *Cell* **170**, 564-+, doi:10.1016/j.cell.2017.06.010 (2017).
- 28 Bailey, M. H. *et al.* Comprehensive Characterization of Cancer Driver Genes and Mutations. *Cell* **174**, 1034-1035, doi:10.1016/j.cell.2018.07.034 (2018).
- 29 Mertins, P. *et al.* Proteogenomics connects somatic mutations to signalling in breast cancer. *Nature* **534**, 55-+, doi:10.1038/nature18003 (2016).
- 30 Wan, Y., Kurosaki, T. & Huang, X. Y. Tyrosine kinases in activation of the MAP kinase cascade by G-protein-coupled receptors. *Nature* **380**, 541-544, doi:10.1038/380541a0 (1996).
- 31 Su, N. *et al.* Lyn is involved in CD24-induced ERK1/2 activation in colorectal cancer (vol 11, 43, 2012). *Mol Cancer* **11**, doi:Artn 6810.1186/1476-4598-11-68 (2012).
- 32 Donato, N. J. *et al.* BCR-ABL independence and LYN kinase overexpression in chronic myelogenous leukemia cells selected for resistance to STI571. *Blood* **101**, 690-698, doi:DOI 10.1182/blood.V101.2.690 (2003).
- 33 Whitfield, M. L., George, L. K., Grant, G. D. & Perou, C. M. Common markers of proliferation. *Nat Rev Cancer* **6**, 99-106, doi:10.1038/nrc1802 (2006).
- 34 El Touny, L. H. *et al.* Combined SFK/MEK inhibition prevents metastatic outgrowth of dormant tumor cells. *J Clin Invest* **124**, 156-168, doi:10.1172/JCI70259 (2014).
- 35 Suzuki, M., Murata, S., Yaoita, H. & Nakagawa, H. An antibody to BP 180 kDa antigen is able to induce an increase of intracellular Ca²⁺ concentration in DJM-1 (human squamous cell carcinoma) cells. *Autoimmunity* **35**, 271-276, doi:10.1080/0891693021000010721 (2002).
- 36 Braz, J. C. *et al.* PKC-alpha regulates cardiac contractility and propensity toward heart failure. *Nat Med* **10**, 248-254, doi:10.1038/nm1000 (2004).
- 37 Sarbassov, D. D., Guertin, D. A., Ali, S. M. & Sabatini, D. M. Phosphorylation and regulation of Akt/PKB by the rictor-mTOR complex. *Science* **307**, 1098-1101, doi:10.1126/science.1106148 (2005).
- 38 Hartley, K. O. *et al.* DNA-dependent protein kinase catalytic subunit: a relative of phosphatidylinositol 3-kinase and the ataxia telangiectasia gene product. *Cell* **82**, 849-856, doi:10.1016/0092-8674(95)90482-4 (1995).
- 39 Ma, Y., Pannicke, U., Schwarz, K. & Lieber, M. R. Hairpin opening and overhang processing by an Artemis/DNA-dependent protein kinase complex in nonhomologous end joining and V(D)J recombination. *Cell* **108**, 781-794, doi:10.1016/s0092-8674(02)00671-2 (2002).
- 40 Shanahan, E. R. *et al.* Influence of cigarette smoking on the human duodenal mucosa-associated microbiota. *Microbiome* **6**, 150, doi:10.1186/s40168-018-0531-3 (2018).
- 41 Botteri, E. *et al.* Smoking and Colorectal Cancer A Meta-analysis. *Jama-J Am Med Assoc* **300**, 2765-2778, doi:DOI 10.1001/jama.2008.839 (2008).
- 42 Zhong, L. *et al.* Dyspepsia and the microbiome: time to focus on the small intestine. *Gut* **66**, 1168-1169, doi:10.1136/gutjnl-2016-312574 (2017).

- 43 Tap, J. *et al.* Identification of an Intestinal Microbiota Signature Associated With Severity of Irritable Bowel Syndrome. *Gastroenterology* **152**, 111-123 e118, doi:10.1053/j.gastro.2016.09.049 (2017).
- 44 Ludvigsson, J. F., Montgomery, S. M. & Ekbom, A. Smoking and celiac disease: a population-based cohort study. *Clin Gastroenterol Hepatol* **3**, 869-874, doi:10.1016/s1542-3565(05)00414-3 (2005).
- 45 Murthy, S. N., Dinoso, V. P., Jr., Clearfield, H. R. & Chey, W. Y. Serial pH changes in the duodenal bulb during smoking. *Gastroenterology* **75**, 1-4 (1978).
- 46 Buchthal, J. *et al.* Induction of cytochrome P4501A by smoking or omeprazole in comparison with UDP-glucuronosyltransferase in biopsies of human duodenal mucosa. *Eur J Clin Pharmacol* **47**, 431-435, doi:10.1007/BF00196857 (1995).
- 47 Xie, S. *et al.* Timeless Interacts with PARP-1 to Promote Homologous Recombination Repair. *Mol Cell* **60**, 163-176, doi:10.1016/j.molcel.2015.07.031 (2015).
- 48 Schwanhauser, B. *et al.* Global quantification of mammalian gene expression control. *Nature* **473**, 337-342, doi:10.1038/nature10098 (2011).
- 49 Schwanhauser, B. *et al.* Global quantification of mammalian gene expression control (vol 473, pg 337, 2011). *Nature* **495**, 126-127, doi:10.1038/nature11848 (2013).
- 50 Papadopoulos, N., Leach, F. S., Kinzler, K. W. & Vogelstein, B. Monoallelic Mutation Analysis (Mama) for Identifying Germline Mutations. *Nat Genet* **11**, 99-102, doi:DOI 10.1038/ng0995-99 (1995).
- 51 Lipkin, S. M. *et al.* The MLH1 D132H variant is associated with susceptibility to sporadic colorectal cancer. *Nat Genet* **36**, 694-699, doi:10.1038/ng1374 (2004).
- 52 Schwitalle, Y., Doeberitz, M. V. & Kloor, M. Immune response against frameshift-induced neopeptides in HNPCC patients and healthy HNPCC mutation carriers - Reply. *Gastroenterology* **135**, 712-713, doi:10.1053/j.gastro.2008.07.002 (2008).
- 53 Lee, D. D. & Seung, H. S. Learning the parts of objects by non-negative matrix factorization. *Nature* **401**, 788-791, doi:10.1038/44565 (1999).
- 54 Alexandrov, L. B. *et al.* The repertoire of mutational signatures in human cancer. *Nature* **578**, 94-101, doi:10.1038/s41586-020-1943-3 (2020).
- 55 Fanale, D. *et al.* Can the tumor-agnostic evaluation of MSI/MMR status be the common denominator for the immunotherapy treatment of patients with several solid tumors? *Crit Rev Oncol Hematol* **170**, 103597, doi:10.1016/j.critrevonc.2022.103597 (2022).
- 56 Baretta, M. & Le, D. T. DNA mismatch repair in cancer. *Pharmacol Ther* **189**, 45-62, doi:10.1016/j.pharmthera.2018.04.004 (2018).
- 57 Mori, Y. *et al.* Instabilotyping reveals unique mutational spectra in microsatellite-unstable gastric cancers. *Cancer Res* **62**, 3641-3645 (2002).
- 58 Cristescu, R. *et al.* Molecular analysis of gastric cancer identifies subtypes associated with distinct clinical outcomes. *Nat Med* **21**, 449-U217, doi:10.1038/nm.3850 (2015).
- 59 Zhu, M. *et al.* Brunner's Gland Hamartoma of the Duodenum: A Literature Review. *Adv Ther* **38**, 2779-2794, doi:10.1007/s12325-021-01750-6 (2021).
- 60 Kwon, N. H., Fox, P. L. & Kim, S. Aminoacyl-tRNA synthetases as therapeutic targets. *Nature Reviews Drug Discovery* **18**, 629-650, doi:10.1038/s41573-019-0026-3 (2019).

- 61 He, X. D. *et al.* Sensing and Transmitting Intracellular Amino Acid Signals through Reversible Lysine Aminoacylations. *Cell Metab* **27**, 151-+, doi:10.1016/j.cmet.2017.10.015 (2018).

REVIEWER COMMENTS

Reviewer #1 (Remarks to the Author):

The authors appropriately answered to all the point raised by this reviewer.

Reviewer #2 (Remarks to the Author):

We thank the author for the comprehensive revision. We appreciate the new extended Figure 5 which nicely illustrates the importance of proteome-based subtyping in DC. Below are the remaining issues to address:

Page 11, Paragraph 2: Based on the proteome similarity between tumor adjacent tissues and isolated tumor tissues per patient, the author claimed that their data supports that the “diversity of DC subtype was derived from their original normal tissues”. This implies that, depending on the normal tissue proteome, the D and C subtypes are already determined and developed. We cannot understand the logic and found this claim lacks essential experimental evidence and seems to be a big message for cancer biology studies. The authors should revise the corresponding statement.

Minor

1. Page 9, Paragraph 2: “We found that MAPK Y187 was the only substrate of LYN” is misleading. Should it be changed to “MAPK Y187 was the only substrate of LYN identified and supported by our data”?
2. Page 9, Paragraph 2: “we analyzed correlation between LYN and cell proliferation” indicates an experiment the author did not do. This text piece can be removed. Also, how did the author get the P value of 6.6E-10 and R value of 0.33 from all the makers shown in extended Figure 4h which have their respective R and P values? Similar analyses were used in many other figures and should be clarified in the Methods.
3. Page 11, Paragraph 2: “access” or “assess”?
4. Page 19: It is nice to know that the protein extraction method the authors used is better than PCT in analyzing lower amounts of tissues. The authors should consider providing a detailed protocol along the methods or supplementary and cite the relevant papers of theirs in the Methods as well, because the sample amount is a major challenge for analyzing the clinical proteomes.

Reviewer #3 (Remarks to the Author):

Although the authors have made an effort to clarify some of queries generated by the first submission, I have not found the revisions to have added any clarity nor cohesiveness to the very large amount of complex data presented.

I personally believe most of the submitted data is aggregated in ways that do not allow the reader to fully understand the narrative provided. Moreover, I remain unconvinced that each subtype should be further broken down into substages, which leads to dozens of subclassifications, sometimes with only 1 or 2 cases for each substage. This does not help readers aggregate the information into a usable classification system.

Lastly, while the discussion seems to have been expanded, much of it appears to be additional data reporting or reiteration of previously stated parts from their results. This is not the purpose of a discussion. Rather,

The point-to-point response of the referees are as follows:

Reviewer #2:

We thank the author for the comprehensive revision. We appreciate the new extended Figure 5 which nicely illustrates the importance of proteome-based subtyping in DC.

Response: We appreciate the reviewer for the positive evaluation and constructive comments. We have revised the manuscript according to the comments. The point-to-point responses were as follows.

Below are the remaining issues to address:

Q1: Page 11, Paragraph 2: Based on the proteome similarity between tumor adjacent tissues and isolated tumor tissues per patient, the author claimed that their data supports that the “diversity of DC subtype was derived from their original normal tissues”. This implies that, depending on the normal tissue proteome, the D and C subtypes are already determined and developed. We cannot understand the logic and found this claim lacks essential experimental evidence and seems to be a big message for cancer biology studies. The authors should revise the corresponding statement.

Response: We appreciate the professional comments to improve the quality of our manuscript.

In our cohort, patients with two major subtypes (duodenal carcinoma (D) subtype and Brunner’s Gland (G) subtype) were enrolled, which account for nearly 90% of duodenal cancer (DC)¹ (Diagn Interv Imaging, 2017, PMID: 28185840). Especially, 60 DC patients (38.5%) had the D subtype only (named Panel 1 in this study), 32 DC patients (20.5%) had the G subtype only (named Panel 2 in this study), and 54 DC patients (34.6%) had both D and G subtypes (named Panel 3 in this study) (**Figure RL1a, also shown in Fig. 4a**), providing a chance for exploring the diversity of the D and G subtypes.

(1) About the logic of the origins of the D and G subtypes.

We analyzed the differential expressed proteins (DEPs) of the Panel 1 (D subtype panel) and the Panel 2 (G subtype panel), and found that Panel 1 overrepresented proteins were enriched in cell cycle (e.g., CDK1/2, CDKN2A, etc.), apoptosis (e.g., CASP3/8, PARP1/9, etc.), and glycolysis (e.g., HK2, PGK1, etc.). The Panel 2 highly expressed proteins participated in ECM signaling (e.g., COL1A1, LAMA, etc.) and complement cascade (e.g., CFH, C4A, etc.) (**Figure RL1b, also shown in Fig. 4b**). The Panel 3, patients had both D and G subtypes, were characterized by cell cycle, ECM signaling, etc., which were also overrepresented in the Panel 1 and Panel 2, respectively (**Figure RL1c, also shown in Extended Data Fig. 5f**).

Interestingly, we found that the morphological features of the normal tissues were diverse among the Panel 1, Panel 2, and Panel 3 (**Figure RL1d, also shown in Fig. 4c**), implying the different characterization of the normal tissues. We further analyzed the different molecular

characterization of the Panel 1, Panel 2, and Panel 3. As a result, the differences of the normal tissues were consistent with the corresponding tumor tissues of the Panel 1, Panel 2, and Panel 3 at the protein level. For example, the highly expressed proteins of normal tissues in the Panel 1 were enriched in cell cycle, apoptosis, and glycolysis. In the Panel 2, complement cascade and ECM signaling were dominant in the normal tissues (**Figure RL1d, also shown in Fig. 4c**). In addition, the Panel 3 consisted of the patients with both D and G subtypes, which shared the common normal tissues. Compared to the Panel 1, ECM signaling (e.g., COL1A1, COL1A2, etc.) was highlighted in the normal tissues of the Panel 3 (**Figure RL1d, also shown in Fig. 4c**). While compared to the Panel 2, cell cycle (e.g., CDK1, CDK2, etc.) was observed in the normal tissues of the Panel 3 (**Figure RL1d, also shown in Fig. 4c**). Together, the different morphological features and molecular characterizations suggested the diversity in the D and G subtypes was associated with their origin normal tissues.

Figure RL1 | The diverse characteristics of the origins of the subtype-based panels, also shown in Fig. 4a, b, c, and Extended Data Fig. 5f in the revised version. a, Sankey diagram analysis of all the 156 cases classified into 4 panels. **b**, Venn diagram depicting the specific driven pathways in the normal (left) and tumor (right) tissues of the Panel 1 and the Panel 2. The green highlighted the normal tissue. The blue and orange indicated the D and G subtypes, respectively. **c**, The represented pathways of the normal tissues, D subtype, and G subtype in the Panel 3. **d**, The morphological feature (H&E staining) (left) and represented pathways (right) of normal tissue (stage 1) among the Panel 1, Panel 2, and Panel 3.

(2) About the revise statement of the origins of the D and G subtypes.

We agree with the reviewer’s points and we revised the conclusion of the origins of the DC subtypes, which was corrected as “Together, the different pathological features and molecular characterizations suggested the diversity in the D and G subtypes was associated with their origin normal tissues.” (Line 6–8, Page 12).

Minor:

Q1. Page 9, Paragraph 2: “We found that MAPK Y187 was the only substrate of LYN” is misleading. Should it be changed to “MAPK Y187 was the only substrate of LYN identified and supported by our data”?

Response: We thank the reviewer for the constructive suggestions and agree with reviewer’s points. In the revised manuscript, we changed the statement as “MAPK Y187 was the only substrate of LYN identified and supported by our data” (**Line 2–3, Page 10**).

Q2. Page 9, Paragraph 2: “we analyzed correlation between LYN and cell proliferation” indicates an experiment the author did not do. This text piece can be removed. Also, how did the author get the P value of 6.6E-10 and R value of 0.33 from all the makers shown in extended Figure 4h which have their respective R and P values? Similar analyses were used in many other figures and should be clarified in the Methods.

Response: We thank the reviewer for the insightful comments. To present the text clarify, we removed the statement “we analyzed correlation between LYN and cell proliferation”.

In our cohort, to assess the correlation between mutation burden/proteins (e.g., LYN, AARS1, etc.) and the enrichment pathways, we applied set variation analysis (GSVA)² (BMC Bioinformatics, 2013, PMID: 23323831) for pathways enrichment analysis, in which molecular Signatures Database (MSigDB) of KEGG gene sets were used. As for the correlation between LYN and cell proliferation, the R values and P values were based on the intensity of LYN and the enrichment values of the pathways among the samples in the Fudan cohort. To present the revised manuscript clearly, we added the section in the “**Methods**” with red text (**Line 24–27, Page 33**).

Q3. Page 11, Paragraph 2: “access” or “assess”?

Response: Thanks for the reminder and we apologized for the mistakes. In the revised manuscript, we corrected “access” as “assess”. In the revised manuscript, we re-presented the logic of the origins of the D and G subtypes, and revised the related statement, in which the description of “assess” was removed. Please see the details in the revised manuscript with red text.

Q4. Page 19: It is nice to know that the protein extraction method the authors used is better than PCT in analyzing lower amounts of tissues. The authors should consider providing a detailed protocol along the methods or supplementary and cite the relevant papers of theirs in the Methods as well, because the sample amount is a major challenge for analyzing the clinical proteomes.

Response: Thanks for the constructive comments and we agree with the reviewer’s points. In the revised manuscript, we added the protein extraction method in the “**Methods**” with red text (**Line 11–20, Page 29**). Please see the details in the revised manuscript.

Reviewer #3 (Remarks to the Author):

Although the authors have made an effort to clarify some of queries generated by the first submission, I have not found the revisions to have added any clarity nor cohesiveness to the very large amount of complex data presented.

Response: We appreciate for the constructive comments of the reviewer. In this round of revision, we have carefully studied the new comments of the reviewer, and the point-to-point responses were as follows.

Q1: I personally believe most of the submitted data is aggregated in ways that do not allow the reader to fully understand the narrative provided. Moreover, I remain unconvinced that each subtype should be further broken down into substages, which leads to dozens of subclassifications, sometimes with only 1 or 2 cases for each substage. This does not help readers aggregate the information into a usable classification system.

Response: Thanks for the constructive and professional comments, and we are sorry for presenting the data aggregately, and explaining the subtypes and substages unclearly. According to the reviewer's comments, we divided the responses into two parts to answer: (1) about explaining the subtypes and substages; (2) about clarifying the data presentation and the major findings of the manuscript.

(1) About explaining the subtypes and substages.

As for the explanation of the subtypes and substages.

Duodenal cancer (DC) is one of the malignantly cancers with low incidence, and featured with complicated subtypes, including 2 major subtypes (adenocarcinoma (named D subtype in this study) and Brunner's gland (named G subtype in this study)) and other subtypes, such as heterotopic pancreas (named P subtype in this study), intermuscular gland (named N subtype in this study), lymphangioma (named L subtype in this study), cystic dystrophy (named C subtype in this study), well-differentiated neuroendocrine tumor (named NET subtype in this study)¹ (Diagn Interv Imaging, 2017, PMID: 28185840). It is common that one DC patient had more than one subtype. The rarity and complicated subtypes of DC remain challenging to collect the DC samples, especially for the rare subtype NET/L/P/N/C subtypes. In our cohort, we collected 2 major (D/G) and 5 rare subtypes (NET/L/P/N/C) of DC, allowing us to explore the diversity and characterization of the DC subtypes.

In DC progression, the morphological characteristics are complicated, especially for the two major subtypes (D and G subtypes). Due to sporadic nature and locations on the posterior or lateral wall of the second part of the duodenum, D subtype follows the transition pattern of colorectal cancer (CRC), from adenoma (T1 stage) to adenocarcinoma, including minimally invasion adenocarcinoma (named MIA stage in this study) and advanced-stage adenocarcinoma (e.g., T2, T3, T4 stages, named DAC stage in this study)³ (J Clin Pathol, 1994, PMID: 7962621). According

to the differentiation degree, the epithelial adenoma can be divided into low low-grade intraepithelial neoplasia stage (named LGIN in this study), high-grade intraepithelial neoplasia stage (named HGIN stage in this study)⁴ (Histopathology, 2020, PMID: 31433515). The G subtype showed the features of high-grade dysplasia, and transformed from normal (named N-G in this study) to hyperplasia (named H-G in this study) and to adenoma (named A-G in this study)⁵⁻⁷ (Am J Gastroenterol, 1995, PMID: 7847303; Physiol Rev, 1958, PMID: 13590933; Gastroenterology, 1948, PMID: 18100243) (**Figure RL2, also shown in Fig. 1a**).

Figure RL2 | The pattern of the stages during the carcinogenesis process of the D and G subtypes, also shown in the Fig. 1a in the revised version.

In our cohort, all cases were classified according to the 4th and 5th edition World Health Organization (WHO) Classification of Digestive system⁴ (Histopathology, 2020, PMID: 31433515). Specifically, the D subtype was then classified into 5 stages, including NT stage, LGIN stage, HGIN stage, MIA stage, and DAC stage. The NT stage included normal duodenal epithelial, mild hyperplasia (1_1), heterotopic gastric fundus gland (1_2), gastric antrum gland (1_3), duodenal epithelial hyperplasia (1_4), P-J polyp (1_5), hyperplastic polyp (1_6). The LGIN stage included tubular adenoma with mild atypia (2_1), tubular adenoma with mild-middle atypia (2_2), tubulovillous adenoma with mild atypia (2_3), and tubulovillous adenoma with mild-middle atypia (2_4). The HGIN stage included tubular adenoma with middle-severe atypia (3_1), tubular adenoma with severe atypia (3_2), tubulovillous adenoma with middle-severe atypia (3_3), and tubulovillous adenoma with severe atypia (3_4) (**Table RL1, also shown in Table 2 in the revised manuscript**). The G subtype, one subtype of non-surface epithelial components, comprised normal (N-G), hyperplasia (H-G) and adenoma (A-G) (**Table RL2, also shown in Table 3 in the revised manuscript**).

To present the (sub)stages/subtype clearly, we added the related explanation in the “Introduction”, the part I of “Results”, and showed the subclassification information in the “Methods” of the revised manuscript.

Table RL1. Subclassification information and the number of substage samples in the D subtype, also shown in Table 2 in the revised manuscript.

Stages	Substages (n = sample number)
--------	-------------------------------

Non-tumor stage (NT stage)	1 (n = 150), 1_1 (n = 1), 1_2 (n = 3), 1_3 (n = 7), 1_4 (n = 4), 1_5 (n = 1), 1_6 (n = 1)
Low-grade intraepithelial neoplasia stage (LGIN stage)	2_1 (n = 22), 2_2 (n = 11), 2_3 (n = 21), 2_4 (n = 8)
High-grade intraepithelial neoplasia stage (HGIN stage)	3_1 (n = 13), 3_2 (n = 9), 3_3 (n = 24), 3_4 (n = 25)
Minimally invasion adenocarcinoma stage (MIA stage)	4 (n = 5)
Duodenal adenocarcinoma stage (DAC stage)	5 (n = 20)

Table RL2. Subclassification information and the number of substage samples in the G subtype, also shown in Table 3 in the revised manuscript.

Stages	Substages (n = sample number)
Normal stage (N-G stage)	G_1 (n = 47)
Hyperplasia Brunner's stage (H-G stage)	G_2 (n = 19), G_2_1 (n = 7)
Adenoma Brunner's stage (A-G stage)	G_3 (n = 22), G_3_1 (n = 1), G_3_2 (n = 1)

As for the necessary for exploring the (sub)stages of DC.

In DC progression, the morphological characteristics are complicated, including the benign polyp (NT stage), adenoma (e.g., LGIN and HGIN stages) to adenocarcinoma (e.g., MIA and DAC stage)³ (J Clin Pathol, 1994, PMID: 7962621). The tubular adenoma and tubulovillous adenoma are usually present in the LGIN and HGIN^{8,9} (Gut, 2017, PMID: 28450390; Histopathology, 2016, PMID: 26212352). The risks of patients at the different (sub)stage in DC progression are quite diverse. Take the D subtype as an example, of which the carcinogenesis process was classified into 5 stages, including the NT, LGIN, HGIN, MIA, and DAC stages (**Table RL1, also shown in Table 2 in the revised manuscript**). For the patients at the LGIN stage, 5-year cumulative probability of eventually progression to the HGIN or to adenocarcinoma has been reported as no more than 50%^{10,11} (Gut, 2003, PMID: 12865270; Gastroenterology, 2004, PMID: 15168373). Whereas, the HGIN is usually progressed to adenocarcinoma in pathology (>90%), and should be considered for surveillance¹² (Digestive endoscopy, 2014, PMID:24750146). The different stages during the carcinogenesis of DC are correlated with the lesion size and invasion to the corresponding layers¹³ (Gut, 2001, PMID: 11156645) and required distinctive operation strategies. For example, for the lesions at the LGIN and HGIN stages (< 20 mm in size), endoscopic resection (e.g., ESD and EMR) is suggested as an effective treatment approach¹⁴ (Gastroenterology, 2005, PMID: 26752110). The lesions invaded into the submucosal layer (\geq 20 mm in size) are assessed as carcinoma (DAC stage) and are required for surgical resection and

surgical resection¹⁵ (Digestive diseases, 2019, PMID: 30921797). For the lesions at the MIA stage, the clinic management is controversial, further implying the complexity and undefined characteristics of the (sub)stages in DC progression. Thus, it is vital that the revelation of the key events during the carcinogenesis process of DC to explore the pathological mechanism and the corresponding the targeted therapy in DC progression. However, the molecular characterization of the (sub)stage of the lesions in DC progression are yet unclear.

According to issues above, we collaborated with the team of experienced gastrointestinal pathologists in Zhongshan hospital, Fudan university, determining the enrolled DC patients and designing the study including evaluating the samples at different (sub)stages during the carcinogenesis process of DC. According to the 4th and 5th edition WHO Classification of Digestive system⁴ (Histopathology, 2020, PMID: 31433515), we divided the carcinogenesis process of D and G subtypes into 5 stages (NT, LGIN, HGIN, MIA, and DAC) and 3 stages (N-G, H-G, and A-G), respectively. In addition, based on the diverse morphological characteristics, the stages of the D and G subtypes were subclassified into 17 and 6 substages, respectively (**Table RL1, 2, also shown in Table 2, 3 in the revised manuscript**). Therefore, a total of 438 samples from 156 DC patients were collected in this study. The advances of the (sub)classification in a pathological region resolved mode enabled us to portray the molecular profiles of DC in a stage resolved mode, and to investigate the occurrences of the mutations/key events and the related effects in during the carcinogenesis of DC, providing a new insight into molecular profiles and corresponding precise medication in DC progression.

The reviewer is correct that sometimes with only 1 or 2 cases for each substage, such as mild hyperplasia, and hyperplastic polyp. In consideration of this issue, we integrated the substages into stages, and used the stage-based classification for the proteogenomic analysis in this study (e.g., **Fig. 2, 3, 4**, etc.), including the NT, LGIN, HGIN, MIA, DAC stages in the D subtype, and the N-G, H-G, and A-G stages in the G subtype (**shown in the “Methods” of the revised manuscript**). Take the D subtype as an example, we found that a staging molecular model drove carcinogenesis as digestion pathway (NT stage) – AMPK signaling/ insulin resistance (LGIN stage) – cell cycle/ DNA repair (tubular adenoma in the HGIN stage) – apoptosis/ aminoacyl-tRNA biosynthesis (tubular adenoma in the HGIN stage) – mTOR signaling/ EGFR signaling (MIA and DAC stages) (**shown in Extended Data Fig. 5g**). In addition, proteogenomics revealed that the gain of chromosome 8q functioned in the transmit from the intraepithelial neoplasia (IEN) phase (LGIN stage) to the infiltration tumor (IFT) phase (HGIN stage last till to DAC stage) via MAPK signaling (**shown in Fig. 3 and Extended Data Fig. 4**), and the *DST* mutation improved mTOR signaling activity in the DAC stage of the D subtype (**shown in Fig. 4 and Extended Data Fig. 5**). To provide the resource for the clinic guidance, we presented the precise substages during the carcinogenesis of the D and G subtypes in the supplementary materials (**shown in Extended Data Fig. 2d and “Methods” of the revised manuscript**), which would be investigated for further analysis in the future.

To present the necessary of the (sub)stage in DC progression clearly, we added this section in the **“Introduction”** and **“Discussion”** in the revised manuscript with red text.

As for the findings of the stages of DC, contributing to a comprehensive proteogenomic characterization in DC progression.

i) The WES-based SCNAs analyses represented the chr8q functioned in the transmit process from the IEN phase (LGIN stage) to the IFT phase (ranging from HGIN stage to DAC stage) in the D subtype, in which the amplification of *LYN* enhanced the protein-level of *LYN* and enhanced phosphorylation. KSEA revealed MAPK signaling (e.g., MAPK1/3/7, RPS6KA3, etc.) related kinases were overrepresented in the *LYN* amplification group, in which MAPK1 showed positive association with *LYN*. In addition, the substrate of MAPK1 Y187 was overrepresented in the *LYN* amplification group, which could activate the substrates of RPS6KA3 (S369, S375, S715, and T365) and TOP2A (S1213). Furthermore, *LYN* positively correlated with cell proliferation, evidenced by MCM4, MCM6, PCNA, TOP2A, etc. Together, the chr8q gain functioned in the transmit process from the IEN phase to the IFT phase, in which the amplification of *LYN* enhanced the protein-level of *LYN*, which activated MAPK signaling and promoted the DC cell proliferation, hinting the potential medical actions of saracatinib¹⁶ (J Clin Invest, 2014, PMID: 24316974) in the DC clinic (**Fig. 3 and Extended Data Fig. 4**).

ii) We identified and summarized a carcinogenetic lineage with 5 dynamic pathways in the D subtype progression at multi-omics level: digestion pathway – AMPK signaling/ insulin resistance – cell cycle/ DNA repair – apoptosis/ aminoacyl-tRNA biosynthesis – mTOR signaling/ EGFR signaling. In addition, we displayed a carcinogenesis path with 3 kinetic pathways in the G subtype progression for the first time: digestion pathway – ECM signaling/ focal adhesion – complement cascade/ platelet activation. The DAC stage prominent mutation of *DST*, plays a key role in calcium (Ca^{2+}) signal and increases Ca^{2+} level¹⁷ (Autoimmunity, 2002, PMID: 12482196), enhanced the protein level of *DST* which regulated the levels of the kinases (PDK1 and PRKDC), and thus activated mTOR signaling at the protein and phosphoprotein levels. Overall, the strong correlation between proteome and genome further validated the influences of carcinogenesis lineages-pathways of diverse subtypes of DC, and provided a referred proteogenomic dataset of DC (**Fig. 4 and Extended Data Fig. 5**).

In addition, we discovered the malignant potential of DC subtypes, and illustrated the diverse characteristics between the D and G subtypes.

(2) About clarifying the data presentation and the major findings of the manuscript.

As for clarifying the data presentation and text.

In the revised version, to present the data and manuscript logically and readably, we streamlined the manuscript and re-arranged the figures and extended data figures in the revised version. In addition, we removed the redundant and unrelated information. The details were shown as follows.

Fig.1 and Extended Data Fig. 1 and 2 showed the overview of the proteogenomic landscape and of the early-stage DC and the quality control in this study. In the second-round revision, to present

the figures clarify, we moved the **Fig. 1d** to the **Extended Data Fig. 2b** in the second-round revision, and removed the redundant **Extended Data Fig. 1a, 2b, and 2e** in the first-round revision.

Fig.2 and Extended Data Fig. 3 presented the enrichment of the Tobacco signature and DNA repair panel in DC. To show the figures clear and readable, we moved the **Fig. 2a, 2d** to the **Extended Data Fig. 3c, 3e**, respectively, in the second-round revision, and removed the repeated information of **Extended Data Fig. 3b, 3e** in the first-round revision.

Fig.3 and Extended Data Fig. 4 illustrated the involvement of chr8q gain in the transmit process from the IEN phase to the IFT phase in the D subtype. To be more logical and clarified, the **Fig. 3a** was moved to the **Extended Data Fig. 4a** in the second-round revision, and the redundant **Extended Data Fig. 4a, 4d** in the first-round revision were removed.

Fig.4 and Extended Data Fig. 5 presented the diverse characteristic origins and carcinogenesis tracks of the D and G subtypes, revealed the staging and driver pathway waves in the progression of the D and G subtypes, and the impacts of *DST* mutation on mTOR signaling in the DAC stage of DC. To clarify the data presentation, we firstly removed the unrelated information of the **Extended Data Fig. 5g, 5h** from the first-round revision, which was added according to the reviewer's comments to display the proteomic characterization of hereditary nonpolyposis colorectal cancer (HNPCC) in the first-round revision. In addition, we moved the **Fig. 4e, 4h** to the **Extended Data Fig. 4g, 4k**, respectively, in the second-round revision, and removed the redundant **Extended Data Fig. 5i, 5j** in the first-round revision.

Fig.5 and Extended Data Fig. 6 elucidated the characterization of immune-based clusters of DC. To exhibit the figures concisely and readably, we moved the **Fig. 5e, 5f** to the **Extended Data Fig. 6f, 6l**, respectively, in the second-round revision. In addition, we removed the unrelated information of the **Extended Data Fig. 6k** from the first-round revision, which was added according to the reviewer's comments to showing the immune-based characterization of HNPCC in the first-round revision.

Fig. 6, 7 and Extended Data Fig. 7, 8 depicted the validation of AARS1, which was demonstrated to promote DC cell proliferation and invasion through elevating lysine-alanylation (K-Ala) modification on PARP1. To show the figures clarifying, we moved the **Fig. 6j** to the **Extended Data Fig. 7j** in the second-round revision. Moreover, the **Fig. 7c, 7d**, showing the results of **Fig. 7b**, were moved to the **Extended Data Fig. 8a, 8b**.

Furthermore, to make our statement more accurate and clarified, we streamlined our manuscript thoroughly and tuned down our statement, which was updated in the revised manuscript with red text.

As for the “**Introduction**” section, we streamlined the information of background of this study, and emphasized the description and necessary of the (sub)stage/subtype in this study. The updated

information was highlighted with red text in the revised manuscript. In addition, the details of the (sub)stage/subtype were added in the “**Methods**” of the revised manuscript.

As for the “**Result**” section, we deleted the description of the removed figures and extended data figures. In addition, we streamlined the complicated text of data presentation, including the list of low/high expressed protein of the Fudan DC cohort, the redundant background, the list and number of the DEPs, etc. Besides, the methods related text were moved to the “**Methods**” of the revised manuscript with red text.

As for the “**Discussion**” section, we deleted the description of proteomic characterization related to the familial adenomatous polyposis (FAP) and HNPCC, which was added in the first-round revision according to the reviewer’s comments. In addition, we streamlined the description of the results presentation.

In addition, we have carefully proofread and rephased the manuscript to assure the clarity of the statement and text. For example, the “upregulated proteins in” was rephased as “enhanced/overrepresented/elevated proteins in”. Moreover, the revised manuscript was also edited by professional native speakers. All changes were highlighted with red text in the revised manuscript. Please see the details in the revised manuscript.

As for the major findings of the revised manuscript.

We streamlined and updated the manuscript with red text, focusing on the most important findings and shortening the description of the general observations. We performed a comprehensive proteogenomic analysis of 438 trace tumor samples from 156 DC patients, covering 2 major and 5 rare subtypes (**Fig. 1 and Extended Data Fig. 1, 2**). There are **5 major parts** in this study as follows: (1) COSMIC-based signature analysis revealed that more patients with the habit of smoking were observed in the Tobacco signature, which was prominent in the D subtype and might induce NNK-O-glucuronic acid secretion via the overrepresentation of glycosaminoglycans (GAGs). DNA repair panel was featured with high tumor mutation burden (TMB) and Microsatellite instability (MSI), in which ARS-PARPs signaling was overactivated (**Fig. 2 and Extended Data Fig. 3**); (2) The chr8q gain might function in the transmit from the IEN phase to the infiltration tumor (IFT) phase, in which MAPK signaling was activated (**Fig. 3 and Extended Data Fig. 4**); (3) Proteome-based analysis disclosed that divergency of DC subtypes was associated with their origin normal tissues, and elucidated the stage-specific molecular characterization (**Fig. 4 and Extended Data Fig. 5**); (4) Stage-based proteomic analysis illustrated the cancer-driving waves along with the mutation accumulation in the D and G subtypes, in which the mutation of *DST* improved mTOR signaling in the DAC stage (**Fig. 4 and Extended Data Fig. 5**); (5) Immune-based cluster delineated the unique profiles of four immune cluster of DC, and illuminated that the *Hot immune cluster* was featured with high TMB and overrepresentation of AARS1 (**Fig. 5 and Extended Data Fig. 6**), which was gradually enhanced in DC progression and catalyzed the lysine-alanylation of PARP1, eventually promoting cell proliferation and tumorigenesis (**Fig. 6, 7, and Extended Data Fig. 7, 8**).

In summary, our study presented a comprehensive multi-omics landscape of DC for the first time. We delineated the characteristic origins and diverse carcinogenesis tracks, and discovered the kinetic waves of the dominant cancer pathways of the D and G subtypes via integrative proteogenomic analysis. Proteogenomics elucidated the positive impacts of *LYN* amplification at the chr8q gain on MAPK signaling in the IFT phase, and revealed the functions of *DST* mutation on mTOR signaling in the DAC stage. We illuminated the drug-targetable AARS1 in the TMB/immune infiltration in DC, which was demonstrated the value of this multi-omics mapping strategy to suppress tumor growth (**Figure RL3, also shown in Extended Data Fig. 9**). We believe this study provides insights into understanding DC and enables new advances in understanding its mechanism and diagnostics, delivering a useful resource for potential therapeutic approaches and personalized medicine for DC.

Figure RL3 | The model of the key events in the progression of DC, also shown in Extended Data Fig. 9 in the revised version.

Q2: Lastly, while the discussion seems to have been expanded, much of it appears to be additional data reporting or reiteration of previously stated parts from their results. This is not the purpose of a discussion. Rather.

Response: Thanks for the constructive comments and valuable suggestion, and we agree with the reviewer's point. According to the comments, we rewrote the discussion part of the manuscript in this round revision.

Firstly, we removed the figures and discussion text about the proteomic characterization of familial adenomatous polyposis (FAP) and HNPCC. FAP and HNPCC are two rare and autosomally inherited disorders¹⁸ (Nature genetics, 1995, PMID: 7550326), and are characterized by the development premalignant and malignant lesions throughout the gastrointestinal tract, such as adenomatous polyps of the stomach, small bowel, etc. In the first-round revision, according to the reviewer's comments, we re-follow up DC patients and found that 2 (1.3%) FAP patients and 4 (2.6%) HNPCC patients were enrolled in our study. Integration of the links of published studies and our cohort, revealed that HNPCC presented diverse molecular characterization on the basis of MSI status, and specifically, MSI-high group (like D subtype in HNPCC) was featured by DNA repair and cell cycle, and the ECM signaling was predominant in the MSI-low group (like G subtype in HNPCC) with lower immune infiltration. The limited samples of FAP and HNPCC exhibited a barrier for presenting comprehensive features of HNPCC, more samples are needed

for further analysis in the future. Therefore, to show the discussion clarifying, we added the discussion of the association between FAP/HNPCC and DC, and deleted the data presentation of the FAP and HNPCC in the discussion of the revised manuscript.

Secondly, we streamlined the results parts presentation and added discussion about the novelty and advantages of the study, including the importance of the (sub)stages in DC progression and the methods for protein extraction of tiny samples. In addition, we added the discussion about the DC subtypes (e.g., potential malignancy) and risks for DC (e.g., smoking, drinking, familial inheritance, etc.). Furthermore, we expanded the discussion of AARS1, which was demonstrated to function in the DC development, providing a useful resource for potential therapeutic approaches and personalized medicine for DC.

Thirdly, we discussed the possible bias for proteome, and highlighted the short-comings of the study in the “**Discussion**” of the revised manuscript with red text, especially concentrated on the limited samples for transcriptome, the insufficient cases for rare subtypes of DC to present comprehensive proteogenomic characterization of DC, more evidence for validating the association between tobacco usage and DC, and so on. Please see the details in the revised manuscript.

References:

- 1 Barat, M. *et al.* Mass-forming lesions of the duodenum: A pictorial review. *Diagn Interv Imaging* **98**, 663-675, doi:10.1016/j.diii.2017.01.004 (2017).
- 2 Hanzelmann, S., Castelo, R. & Guinney, J. GSEA: gene set variation analysis for microarray and RNA-Seq data. *Bmc Bioinformatics* **14**, doi:Artn 710.1186/1471-2105-14-7 (2013).
- 3 Spigelman, A. D. *et al.* Evidence for Adenoma-Carcinoma Sequence in the Duodenum of Patients with Familial Adenomatous Polyposis. *J Clin Pathol* **47**, 709-710, doi:DOI 10.1136/jcp.47.8.709 (1994).
- 4 Nagtegaal, I. D. *et al.* The 2019 WHO classification of tumours of the digestive system. *Histopathology* **76**, 182-188, doi:10.1111/his.13975 (2020).
- 5 Levine, J. A., Burgart, L. J., Batts, K. P. & Wang, K. K. Brunner's gland hamartomas: clinical presentation and pathological features of 27 cases. *Am J Gastroenterol* **90**, 290-294 (1995).
- 6 Grossman, M. I. The Glands of Brunner. *Physiol Rev* **38**, 675-690 (1958).
- 7 Erb, W. H. & Johnson, T. A. Hyperplasia of Brunner's glands simulating duodenal polyposis. *Gastroenterology* **11**, 740-745 (1948).
- 8 East, J. E. *et al.* British Society of Gastroenterology position statement on serrated polyps in the colon and rectum. *Gut* **66**, 1181-1196, doi:10.1136/gutjnl-2017-314005 (2017).
- 9 Bettington, M. *et al.* Serrated tubulovillous adenoma of the large intestine. *Histopathology* **68**, 578-587, doi:10.1111/his.12788 (2016).
- 10 Lim, C. H. *et al.* Ten year follow up of ulcerative colitis patients with and without low grade dysplasia. *Gut* **52**, 1127-1132, doi:DOI 10.1136/gut.52.8.1127 (2003).
- 11 Itzkowitz, S. H. & Harpaz, N. Diagnosis and management of dysplasia in patients with inflammatory bowel diseases. *Gastroenterology* **126**, 1634-1648, doi:10.1053/j.gastro.2004.03.025 (2004).
- 12 Kakushima, N., Ono, H., Takao, T., Kanemoto, H. & Sasaki, K. Method and timing of resection

- of superficial non-ampullary duodenal epithelial tumors. *Dig Endosc* **26 Suppl 2**, 35-40, doi:10.1111/den.12259 (2014).
- 13 Ono, H. *et al.* Endoscopic mucosal resection for treatment of early gastric cancer. *Gut* **48**, 225-229, doi:DOI 10.1136/gut.48.2.225 (2001).
- 14 Klein, A., Tutticci, N., Singh, R. & Bourke, M. J. Expanding the Boundaries of Endoscopic Resection: Circumferential Laterally Spreading Lesions of the Duodenum. *Gastroenterology* **150**, 560-563, doi:10.1053/j.gastro.2015.12.031 (2016).
- 15 Pavlovic-Markovic, A., Dragasevic, S., Krstic, M., Stojkovic Lalosevic, M. & Milosavljevic, T. Assessment of Duodenal Adenomas and Strategies for Curative Therapy. *Dig Dis* **37**, 374-380, doi:10.1159/000496697 (2019).
- 16 El Touny, L. H. *et al.* Combined SFK/MEK inhibition prevents metastatic outgrowth of dormant tumor cells. *J Clin Invest* **124**, 156-168, doi:10.1172/JCI70259 (2014).
- 17 Suzuki, M., Murata, S., Yaoita, H. & Nakagawa, H. An antibody to BP 180 kDa antigen is able to induce an increase of intracellular Ca²⁺ concentration in DJM-1 (human squamous cell carcinoma) cells. *Autoimmunity* **35**, 271-276, doi:10.1080/0891693021000010721 (2002).
- 18 Papadopoulos, N., Leach, F. S., Kinzler, K. W. & Vogelstein, B. Monoallelic Mutation Analysis (Mama) for Identifying Germline Mutations. *Nat Genet* **11**, 99-102, doi:DOI 10.1038/ng0995-99 (1995).

REVIEWERS' COMMENTS

Reviewer #2 (Remarks to the Author):

The authors addressed all my points. I am now supportive.

The point-to-point response of the referees are as follows:

Reviewer #2: The authors addressed all my points. I am now supportive.

Response: We deeply appreciate the reviewer for the kind support and the positive evaluation.